# Understanding Task Vectors in In-Context Learning: Emergence, Functionality, and Limitations

**Yuxin Dong, Jiachen Jiang, Zhihui Zhu & Xia Ning** [*]
Department of Computer Science and Engineering, The Ohio State University
`{dong.1357, jiang.2880, zhu.3440, ning.104}@osu.edu`

## Abstract

Task vector is a compelling mechanism for accelerating inference in in-context learning (ICL) by distilling task-specific information into a single, reusable representation. Despite their empirical success, the underlying principles governing their emergence and functionality remain unclear. This work proposes the *Task Vectors as Representative Demonstrations* conjecture, positing that task vectors encode single in-context demonstrations distilled from the original ones. We provide both theoretical and empirical support for this conjecture. First, we show that task vectors naturally emerge in linear transformers trained on triplet-formatted prompts through loss landscape analysis. Next, we predict the failure of task vectors in representing high-rank mappings and confirm this on practical LLMs. Our findings are further validated through saliency analyses and parameter visualization, suggesting an enhancement of task vectors by injecting multiple ones into few-shot prompts. Together, our results advance the understanding of task vectors and shed light on the mechanisms underlying ICL in transformer-based models.

## 1 Introduction

In-context learning (ICL) is a core capability of large language models (LLMs), allowing them to perform new tasks without parameter updates by conditioning on a few input-output examples in the prompt (Brown et al., 2020). Unlike traditional training, ICL relies on attention-based mechanisms to infer task structure directly from context. This surprising generalization ability has led to growing interest in uncovering the principles of learning purely from contextual examples (Xie et al., 2022; Chan et al., 2022; Dai et al., 2023; Shen et al., 2024; Deutch et al., 2024).

A recent work investigates the task vector method (Hendel et al., 2023) (concurrent works include function vectors (Todd et al., 2024) and in-context vectors (Liu et al., 2024)), a technique that distills underlying task information from ICL demonstrations into a single vector. Typically, ICL prompts are structured as sequences of triplets, each encoding a semantic mapping, in addition to a query at the end (e.g., *"hot → cold, up → down, dark →"*). Task vectors are then extracted from the hidden states of the last (→) token. Once obtained, these vectors can be injected into new zero-shot prompts (e.g., *"big →"*), enabling the model to generalize to unseen inputs in a zero-shot fashion.

Task vectors naturally emerge even in small transformer models trained from scratch (Yang et al., 2025), suggesting that their formation is a general property of attention-based architectures. Recent studies further demonstrate that task vectors can be enhanced by aggregating hidden states across multiple layers and arrow tokens (Li et al., 2024). Beyond language models, task vectors are also effective in large-scale visual (Hojel et al., 2024) and multi-modal (Huang et al., 2024) models.

Despite their empirical effectiveness, the underlying mechanism of task vectors, especially how they emerge, function, and encode task information, remains poorly understood. This paper takes a step toward unveiling the principles behind it by introducing the following conjecture:

---

[*]Corresponding author.

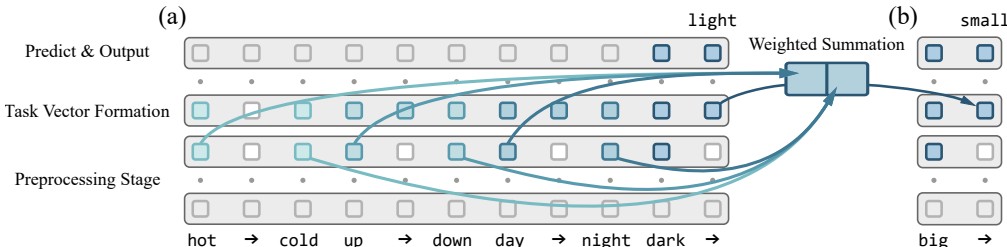

Figure 1: Overview of task vector and our main conjecture. (a) Task vector emerges during ICL by distilling from the preceding in-context demonstrations. (b) It can then be injected into zero-shot prompts and functions as a single, representative demonstration, facilitating efficient prediction.

---

**Conjecture (Task Vectors as Representative Demonstrations)**

*The injected task vector facilitates zero-shot inference by encoding a single representative demonstration, distilled from the original in-context examples.*

---

An intuitive illustration is provided in Figure 1. In the following sections, we validate this conjecture through various empirical and theoretical perspectives. These analyses comprehensively explain how task vectors naturally emerge within attention-based model architectures, effectively encode task-related information, and facilitate inference in zero-shot prompts. Our work advances the understanding of the underlying mechanisms behind ICL, clarifying both the efficacy and limitations of task vectors in transformer-based LLMs. The highlights of this paper are as follows:

- **Theoretical Justification in Linear-Attention Models:** We theoretically characterize the critical points of linear-attention models and demonstrate how they solve random linear regression tasks through embedding concatenation and gradient descent. With a triplet-formatted input prompt structure, task vectors naturally emerge at arrow tokens as weighted summations of the in-context demonstrations, potentially enhancing robustness under representational perturbations by redundantly encoding task information. Empirically, the learned linear model parameters closely align with the predicted structure and successfully replicate the task vector mechanism.

- **Empirical Verification in Practical LLMs:** We visualize the information flow in LLMs with saliency analysis and observe patterns consistent with linear models, suggesting they share similar underlying mechanisms. According to our conjecture, inference with task vectors is analogous to 1-shot ICL, which is inherently limited to rank-one meta-predictors under the gradient descent perspective. To validate this, we introduce a series of bijection tasks that are provably unsolvable by rank-one predictors, and empirically confirm this failure in real-world transformers. Building on these insights, we enhance the standard task vector method by injecting multiple vectors into few-shot prompts, resulting in consistent performance gains across a range of ICL tasks.

## 1.1 RELATED WORKS

**Theory of ICL.** Recent analyses have shown that attention layers can simulate gradient-descent algorithms for regression tasks (Garg et al., 2022; Von Oswald et al., 2023a; Ahn et al., 2023; Wu et al., 2024). Other works study generalization and sample complexity (Xie et al., 2022; Chan et al., 2022; Shen et al., 2024; Von Oswald et al., 2023b; Deutch et al., 2024). These works reveal the inductive bias of attention but leave open how abstract task representations are formed or encoded.

**Task Vector Mechanism.** Multiple recent works identified the mechanism of task vectors during ICL inference (Hendel et al., 2023; Todd et al., 2024; Liu et al., 2024). These vectors emerge in the pretraining stage of LLMs (Yang et al., 2025) and extend beyond text to vision (Hojel et al., 2024) and multimodal (Huang et al., 2024) models. Despite the effectiveness, their underlying mechanism remains poorly understood. A concurrent work (Bu et al., 2025) interprets them via a word2vec-like additive scheme, but is limited to simple additive tasks, single-token prompts, and 1-layer models. In contrast, our analysis extends to pairwise or triplet prompts and multi-layer attention.

A more comprehensive discussion of the related works can be found in Appendix A.2.

## 2 SETTING: LINEAR REGRESSION WITH LINEAR-ATTENTION MODELS

**Notations:** We write $[n] = \{1, \cdots, n\}$. The Hadamard product is denoted by $\circ$, and the Kronecker product by $\otimes$. The identity matrix of dimension $n$ is denoted by $I_n$, while $0_n$ and $0_{m \times n}$ represent zero vectors or matrices of the corresponding dimensions. Subscripts are omitted when the dimensions are clear from context. We define $\mathcal{M}(M) = \{ \Lambda \in \mathbb{R}^{\dim(M)} \mid \Lambda = M \circ A, \ A \in \mathbb{R}^{\dim(M)} \}$ as the set of masked matrices induced by mask $M$. For a general matrix $A$, the element at the $i$-th row and $j$-th column is denoted by $A_{i,j}$, and the sub-block from rows $i$ to $k$ and columns $j$ to $l$ is denoted by $A_{i:k,j:l}$. $\mathrm{diag}(A_1, \cdots, A_n)$ represents the block-diagonal matrix constructed by $\{A_i\}_{i=1}^n$.

**Random Linear Regression:** Following works (Garg et al., 2022; Von Oswald et al., 2023a; Ahn et al., 2023; Wu et al., 2024), we consider training linear transformers on random instances of linear regression. Let $\{x_i\}_{i=1}^{n+1}$, where $x_i \in \mathbb{R}^d$, denote covariates drawn i.i.d. from distribution $P_x$, and let $\{w_i\}_{i=1}^d$, where $w_i \in \mathbb{R}^d$, denote coefficients drawn i.i.d. from distribution $P_w$. Define the coefficient matrix $W = [w_1 \ \cdots \ w_d]^\top \in \mathbb{R}^{d \times d}$. The responses are then generated as $y_i = W x_i$ for $i \in [n+1]$. We denote by $X, Y \in \mathbb{R}^{d \times n}$ the matrices whose columns are $x_i$ and $y_i$, respectively. The query covariate and response are denoted by $x_{\text{test}} = x_{n+1}$ and $y_{\text{test}} = y_{n+1}$ respectively.

**Linear Self-Attention Model:** Following prior works (Von Oswald et al., 2023a; Ahn et al., 2023; Wu et al., 2024), we consider transformers composed of linear self-attention layers. Let $Z_0 \in \mathbb{R}^{2d \times d_p}$ denote the input matrix constructed from $X$, $Y$ and $x_{\text{test}}$ but excluding $y_{\text{test}}$, where $d_p$ denotes the number of tokens and varies across prompt structures. The model is defined by stacking $L$ attention blocks with skip connections, where the $l$-th layer is expressed as:

$$Z_l = Z_{l-1} + \frac{1}{n} \mathrm{Attn}_{V_l, Q_l}(Z_{l-1}), \qquad \mathrm{Attn}_{V,Q}(Z) = V Z M (Z^\top Q Z). \tag{1}$$

Here, the trainable parameters are $\{V_l, Q_l\}_{l=1}^L$, where $V_l \in \mathbb{R}^{2d \times 2d}$ denotes the projection and value matrices, and $Q_l \in \mathbb{R}^{2d \times 2d}$ denotes the query and key matrices. Following the work (Ahn et al., 2023), we adopt a masking matrix $M = \mathrm{diag}(I_{d_p-1}, 0)$ to prevent attention from earlier tokens to the final one. The output of the model is defined as $\mathsf{TF}\big(Z_0; \{V_l, Q_l\}_{l=1}^L\big) = (Z_L)_{(d+1:2d), d_p}$ (i.e., the latter half of the last column). This definition aligns with the structure of the input $Z_0$, which will be further discussed in subsequent sections. During training, the parameters are optimized to minimize the expected ICL risk over random linear regression instances:

$$\mathcal{L}\big(\{V_l, Q_l\}_{l=1}^L\big) = \mathbb{E}_{Z_0, W} \big\| \mathsf{TF}\big(Z_0; \{V_l, Q_l\}_{l=1}^L\big) + W x_{\text{test}} \big\|_2^2. \tag{2}$$

## 3 EMERGENCE OF TASK VECTORS IN LINEAR-ATTENTION MODELS

Firstly, we present theoretical evidence that task vectors naturally arise in simple linear transformers. Specifically, we analyze the loss landscape of the in-context risk, focusing on the properties of its critical points. As a startup, recall the standard linear regression setup (Ahn et al., 2023; Wu et al., 2024), where the $(x_i, y_i)$ pairs for each demonstration are concatenated to form the input prompt:

$$Z_0 = \begin{bmatrix} X & x_{\text{test}} \\ Y & 0 \end{bmatrix} = \begin{bmatrix} x_1 & x_2 & \cdots & x_n & x_{\text{test}} \\ y_1 & y_2 & \cdots & y_n & 0 \end{bmatrix} \in \mathbb{R}^{2d \times d_p}, \quad d_p = n+1. \tag{3}$$

According to existing analyses (Ahn et al., 2023; Zhang et al., 2024; Mahankali et al., 2024), each attention layer in this setting performs one step of gradient descent on the learned coefficient matrix. Specifically, the theoretically optimal single-layer (possibly nonlinear) attention (Katharopoulos et al., 2020) implements the following predictive function (Ahn et al., 2023) when the covariates are drawn from $P_x = \mathcal{N}(0, I_d)$, by selecting $V_1 \propto \mathrm{diag}(0_{d \times d}, I_d)$ and $Q_1 \propto \mathrm{diag}(I_d, 0_{d \times d})$:

$$\mathsf{TF}(Z_0; (V_1, Q_1)) = -\frac{1}{n} Y \sigma(X)^\top \sigma(x_{\text{test}}), \quad \text{where } \sigma : \mathbb{R}^d \mapsto \mathbb{R}^r \text{ is a kernel function.} \tag{4}$$

Here, we abbreviate $[\sigma(x_1) \ \cdots \ \sigma(x_n)]$ as $\sigma(X)$. This model employs $W' \propto Y \sigma(X)^\top$ as an estimate of $W$, yielding prediction $\hat{y}_{\text{test}} = W' \sigma(x_{\text{test}})$. This paper considers alternative settings more reflective of practical scenarios, where $x_i$ and $y_i$ are separated as distinct tokens. As noted (Zuo et al., 2025), such separation necessitates the usage of position encodings for bi-directional attention. Following prior analysis (Kazemnejad et al., 2023), we assume that position encodings are appended to the input tokens, and reformulate the layer-wise update rule of self-attention as:

$$\mathrm{Attn}_{V,Q}(Z) = V Z M \begin{bmatrix} Z^\top & P^\top \end{bmatrix} Q \begin{bmatrix} Z \\ P \end{bmatrix}, \quad \text{where } P \in \mathbb{R}^{d_p \times d_p}. \tag{5}$$

For analytical tractability, we take $P = I_{d_p}$ as one-hot position encodings. Following previous work (Ahn et al., 2023) (see Appendix A.3 for more explanation), we further impose that:

$$V_l = \text{diag}(A_l, B_l), \quad Q_l = \text{diag}(C_l, 0_{d \times d}, D_l), \quad \text{where } A_l, B_l, C_l \in \mathbb{R}^{d \times d}, \ D_l \in \mathbb{R}^{d_p \times d_p}. \quad (6)$$

These parameterizations ensure that the projection and attention operations act independently on the covariate, response, and positional components of the input. This structural decoupling is essential for understanding how the transformer identifies the dependency between each $(x_i, y_i)$ pair and revealing the actual optimization algorithm being executed by the model. The proofs for the main theoretical results in this paper are available in Appendix D.

## 3.1 WARM-UP: LEARNING WITH PAIRWISE DEMONSTRATIONS

We begin by analyzing the optimization of linear transformers on pairwise demonstrations. Following previous approach (Garg et al., 2022; Wibisono & Wang, 2023; Xing et al., 2024), we decompose each demonstration in eq. (3) into a pair of tokens $Z_0^i = \begin{bmatrix} x_i & 0 \\ 0 & y_i \end{bmatrix} \in \mathbb{R}^{2d \times 2}$ to better reflect the practical ICL prompt structure:

$$Z_0 = \begin{bmatrix} Z_0^1 & \cdots & Z_0^n & Z_0^{\text{test}} \end{bmatrix} = \begin{bmatrix} x_1 & 0 & \cdots & x_n & 0 & x_{\text{test}} & 0 \\ 0 & y_1 & \cdots & 0 & y_n & 0 & 0 \end{bmatrix}, \quad d_p = 2n + 2. \quad (7)$$

The following theorem suggests that certain critical points of the in-context risk effectively solve the regression problem by first concatenating each pair of $(x_i, y_i)$ into the same tokens, and then executing a variant of the gradient descent algorithm to compute the prediction. To simplify notation, we denote $A = \{A_l\}_{l=1}^L$ (similarly for $B$, $C$, and $D$) and present:

**Theorem 1** (**Critical Points; Pairwise Demonstrations**). *Assume that $P_x = \mathcal{N}(0, \Sigma)$ and $P_w = \mathcal{N}(0, \Sigma^{-1})$ with $\Sigma \in \mathbb{R}^{d \times d}$ satisfying $\Sigma \succ 0$. Define $\mathcal{S}_I, \mathcal{S}_\Sigma \subset \mathbb{R}^{d \times d}$ and $\mathcal{S}_P \subset \mathbb{R}^{d_p \times d_p}$ as*

$$\mathcal{S}_I = \{\lambda I_d \mid \lambda \in \mathbb{R}\}, \quad \mathcal{S}_\Sigma = \{\lambda \Sigma^{-1} \mid \lambda \in \mathbb{R}\}, \quad \mathcal{S}_P = \{\text{diag}(I_n \otimes \Lambda_1, \Lambda_2) \mid \Lambda_1, \Lambda_2 \in \mathbb{R}^{2 \times 2}\}.$$

*Consider optimizing an $L$-layer transformer under parameter configuration in eq. (6), we have*

$$\inf_{A, B \in \mathcal{S}_I^L, \ C \in \mathcal{S}_\Sigma^L, \ D \in \mathcal{S}_P^L} \sum_{H \in A \cup B \cup C \cup D} \left\| \nabla_H \mathcal{L}\left(\{V_l, Q_l\}_{l=1}^L\right) \right\|_F^2 = 0.$$

To understand the behavior of these critical points within a self-attention layer, we fix $\Sigma = I_d$ and take $A_l, B_l = I_d$, $C_l = -\lambda I_d$, and $D_l = \text{diag}(I_n \otimes \Lambda_1, \Lambda_2)$. Let the first and last $d$ rows of $Z_l$ be denoted by $X_l$ and $Y_l$, respectively. Under these settings, the update rule of each layer becomes:

$$Z_l = Z_{l-1} - \lambda Z_{l-1} M X_{l-1}^\top X_{l-1} + \begin{bmatrix} Z_{l-1}^1 \Lambda_1 & \cdots & Z_{l-1}^n \Lambda_1 & Z_{l-1}^{\text{test}} \text{diag}(1, 0) \Lambda_2 \end{bmatrix}. \quad (8)$$

The above update can be decomposed into the following two distinct components:

- **Gradient Descent:** The first component, $Z_l \leftarrow Z_{l-1} - \lambda Z_{l-1} M X_{l-1}^\top X_{l-1}$, implements the GD++ algorithm (Von Oswald et al., 2023a). This variant enhances convergence speed over standard gradient descent by improving the condition number of $X_{l-1}^\top X_{l-1}$. Notably, this operation modifies only $X_l$ but not $Y_l$ for the first layer, as implied by the structure of $Q_l$ (eq. (6)).

- **Embedding Concatenation:** The second component, $Z_l^i \leftarrow Z_{l-1}^i + Z_{l-1}^i \Lambda_1$ for $i \in [n]$, mixes each pair of $(x_i, y_i)$ tokens. Given that $x_i$ and $y_i$ tokens are initially linearly separable as in our formulation, this operation concatenates each $(x_i, y_i)$ pair, thereby *transforming pairwise demonstrations into the original single-token format*. For the query token $Z_l^{\text{test}}$, this operation copies $x_{\text{test}}$ into the final token, reconstructing the structure in eq. (3), where each non-final token directly concatenates $(x_i, y_i)$ of a demonstration, and the final token contains only $x_{\text{test}}$.

In summary, our analysis reveals that for pairwise demonstrations, the first attention layer leverages position encodings to distinguish between covariate and response tokens, subsequently concatenating them to form a single-token prompt structure. The remaining layers then apply the GD++ algorithm, mirroring the learning dynamics on single-token demonstrations. As a result, **an $L$-layer linear transformer allocates one layer for embedding concatenation and utilizes the remaining $L-1$ layers to perform gradient descent**. In Figure 2a, we visualize the learned $D_l$ weights under the setting of Theorem 1, and observe that they closely match the critical point structure of $\mathcal{S}_P$.

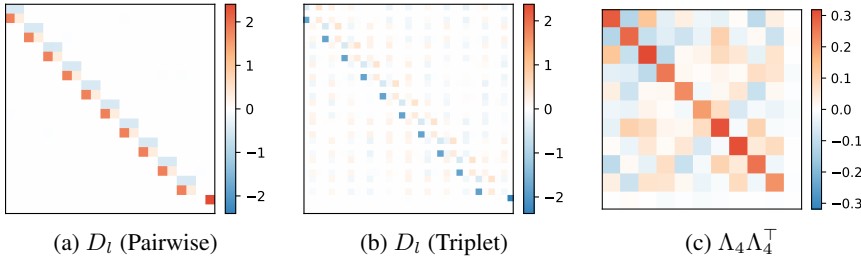

(a) $D_l$ (Pairwise)      (b) $D_l$ (Triplet)      (c) $\Lambda_4\Lambda_4^\top$

Figure 2: Visualization of learned $D_l$ weights. (a) Pairwise demonstrations yield a block-diagonal structure aligned with Theorem 1. (b) Triplet demonstrations yield a richer structure aligned with Theorem 2. (c) The learned matrix $\Lambda_4$ has nearly orthonormal rows as suggested by Proposition 3.

## 3.2 EMERGENCE OF TASK VECTORS WITH TRIPLET DEMONSTRATIONS

Next, to better reflect the prompt structure of practical ICL, we insert additional zero tokens between each pair of $(x_i, y_i)$ to simulate the arrow ($\rightarrow$) tokens. This reformulates each demonstration as a triplet $(x_i, \rightarrow, y_i)$, enabling us to analyze the critical points with these triplet demonstrations:

$$Z_0 = \begin{bmatrix} x_1 & 0 & 0 & \cdots & x_n & 0 & 0 & x_{\text{test}} & 0 & 0 \\ 0 & 0 & y_1 & \cdots & 0 & 0 & y_n & 0 & 0 & 0 \end{bmatrix}, \quad d_p = 3n + 3. \tag{9}$$

**Theorem 2** (**Critical Points; Triplet Demonstrations**). *Assume that $P_x = \mathcal{N}(0, \Sigma)$ and $P_w = \mathcal{N}(0, \Sigma^{-1})$ with $\Sigma \in \mathbb{R}^{d \times d}$ satisfying $\Sigma \succ 0$. Define $\mathcal{S}_I, \mathcal{S}_\Sigma \subset \mathbb{R}^{d \times d}$ and $\mathcal{S}_P \subset \mathbb{R}^{d_p \times d_p}$ as*

$$\mathcal{S}_I = \{\lambda I_d \mid \lambda \in \mathbb{R}\}, \quad \mathcal{S}_\Sigma = \{\lambda \Sigma^{-1} \mid \lambda \in \mathbb{R}\},$$

$$\mathcal{S}_P = \Big\{ \text{diag}(I_n \otimes \Lambda_1, \Lambda_2) + I_{n+1} \otimes \Lambda_3 + \Lambda_4 \otimes \Lambda_5 \Big|$$

$$\Lambda_1, \Lambda_2 \in \mathcal{M}\left(\begin{smallmatrix} 1 & 0 & 1 \\ 0 & 0 & 0 \\ 1 & 0 & 1 \end{smallmatrix}\right), \Lambda_3 \in \mathcal{M}\left(\begin{smallmatrix} 0 & 0 & 0 \\ 0 & 1 & 0 \\ 0 & 0 & 0 \end{smallmatrix}\right), \Lambda_4 \in \mathbb{R}^{(n+1) \times (n+1)}, \Lambda_5 \in \mathcal{M}\left(\begin{smallmatrix} 0 & 1 & 0 \\ 0 & 0 & 0 \\ 0 & 1 & 0 \end{smallmatrix}\right) \Big\}.$$

*Consider optimizing an $L$-layer transformer under parameter configuration in eq.* (6)*, we have*

$$\inf_{A,B \in \mathcal{S}_I^L,\ C \in \mathcal{S}_\Sigma^L,\ D \in \mathcal{S}_P^L} \sum_{H \in A \cup B \cup C \cup D} \left\| \nabla_H \mathcal{L}\left(\{V_l, Q_l\}_{l=1}^L\right) \right\|_F^2 = 0.$$

To analyze the behavior of each attention layer, we note that the critical points for the matrices $A_l$, $B_l$, and $C_l$ remain consistent with Theorem 1, thereby implementing the GD++ algorithm. For the matrix $D_l$, we decompose its structure into three distinct components:

- **Embedding Concatenation:** The first component, $\text{diag}(I_n \otimes \Lambda_1, \Lambda_2)$, mixes each pair of $(x_i, y_i)$ tokens, effectively concatenating them — analogous to the operation analyzed in the previous section. This converts all non-arrow tokens into single-token demonstrations.

- **Self Magnification:** The second component, $I_{n+1} \otimes \Lambda_3$, scales the embeddings corresponding to each arrow ($\rightarrow$) token by a fixed constant and adds them back to themselves.

- **Task Vector Formation:** The third component, $\Lambda_4 \otimes \Lambda_5$, performs a weighted summation across all demonstrations in the prompt. This operation is central to the emergence of task vectors. Let $[\beta_1 \ \cdots \ \beta_{n+1}] \in \mathbb{R}^{n \times (n+1)}$ denote the first $n$ rows of $\Lambda_4$ (we will soon show that the last row of $\Lambda_4$ converges to zero), *the first self-attention layer then outputs $n + 1$ linear combinations of the demonstrations as the hidden states for the arrow tokens*, expressed as $z_{\text{tv}}^i = \left[\begin{smallmatrix} \alpha_1 X \beta_i \\ \alpha_2 Y \beta_i \end{smallmatrix}\right]$ for $i \in [n + 1]$, where $\alpha_1, \alpha_2 \in \mathbb{R}$ are the two non-zero entries of $\Lambda_5$. These vectors can then be injected into zero-shot prompts and function as single-token demonstrations.

This mechanism provides strong theoretical evidence for our main conjecture, demonstrating that **task vectors naturally emerge from the pretraining stage of linear-attention transformers on triplet-formatted prompts**. Notably, the structure of $\mathcal{S}_P$ closely aligns with our visualization of $D_l$ in Figure 2b, confirming our theoretical analysis. We now further investigate the structure of the weight matrix $\Lambda_4$, and present the following result:

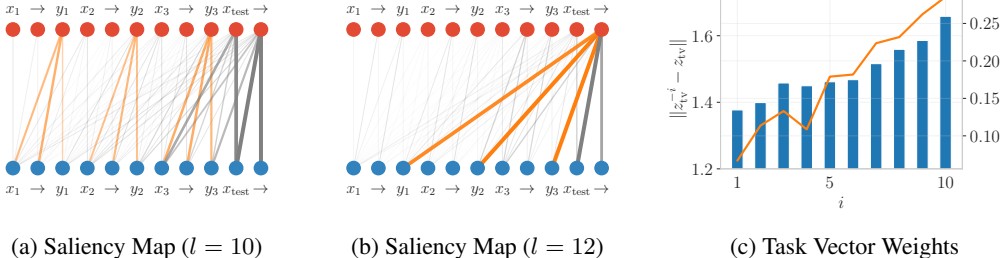

(a) Saliency Map ($l = 10$)  (b) Saliency Map ($l = 12$)  (c) Task Vector Weights

Figure 3: Visualizations on Llama-7B: (1) saliency matrices as bipartite graphs between layer $l$ (●) and $l + 1$ (●), edge widths indicate saliency magnitude; (2) variations in the extracted task vector after perturbing the $i$-th demonstration (▮) and the optimal task vector weights (—) obtained by optimizing Proposition 5. (a) Each $y_i$ token attends to its corresponding $(x_i, y_i)$ pair, reflecting embedding concatenation. (b) The final ($\rightarrow$) token attends broadly to all $y_i$ tokens, indicating task vector formation. This occurs just before the optimal injection layer ($l = 13$). (c) The predicted task vector weights closely match the trend of empirical results, validating our theoretical model.

**Proposition 3** (**Optimal Task Vector Weights**). *Assume $P_x, P_w = \mathcal{N}(0, I_d)$. Consider optimizing a $2$-layer linear-attention transformer with triplet demonstrations and parameter configuration given in eq. (6), and assume $C_1 = 0_{d \times d}$. Let*

$$D_1 = \mathrm{diag}(I_n \otimes \Lambda_1, \Lambda_2) + I_{n+1} \otimes \Lambda_3 + \Lambda_4 \otimes \Lambda_5 \in \mathcal{S}_P$$

*be any minimizer of the in-context risk $\mathcal{L}\big(\{V_l, Q_l\}_{l=1}^L\big)$, we then have $\Lambda_4 \in \mathcal{S}_U$, where*

$$\mathcal{S}_U = \big\{\Lambda \mid \Lambda\Lambda^\top = \lambda \, \mathrm{diag}(I_n, 0), \lambda \in \mathbb{R}\big\}.$$

This result suggests that the optimal $\Lambda_4$ weight matrix satisfies two key properties: (1) the last row is zero, and (2) the first $n$ rows are mutually orthonormal. These conditions imply that the learned weight vectors $\beta_1, \cdots, \beta_{n+1}$ are likely to be distinct. Therefore, the $n + 1$ task vectors produce diverse linear combinations of the demonstrations, thereby enriching the representation within the input prompt. This implication is verified in Figure 2c. While task vectors are typically extracted from the final arrow ($\rightarrow$) token in standard usage, here we consider all arrow tokens as task vectors as bi-directional attention allows each to aggregate information from the full prompt.

## 4    PREDICTED FAILURE OF TASK VECTORS ON BIJECTION TASKS

We then present an empirical observation that supports our conjecture. Consider the setting where task vectors are injected into zero-shot prompts. Based on our prior analysis, the injected task vector $z_{\mathrm{tv}}$ is formed as a weighted summation of the original demonstrations. As a result, we show that the injected prompt reconstructs the single-token structure in eq. (3) with only 1 demonstration:

$$Z_0 = [z_{\mathrm{test}} \quad z_{\mathrm{tv}} \quad 0] = \begin{bmatrix} x_{\mathrm{test}} & x_{\mathrm{tv}} & 0 \\ 0 & y_{\mathrm{tv}} & 0 \end{bmatrix} = \begin{bmatrix} x_{\mathrm{test}} & \alpha_1 X \beta & 0 \\ 0 & \alpha_2 Y \beta & 0 \end{bmatrix} \in \mathbb{R}^{2d \times 3}, \qquad (10)$$

where the weight vector $\beta \in \mathbb{R}^n$ comes from the last column of $\Lambda_4$, and the weights $\alpha_1, \alpha_2$ come from $\Lambda_5$ (see our discussion after Theorem 2). After the first layer, the $\Lambda_2$ matrix of $\mathcal{S}_P$ moves $x_{\mathrm{test}}$ to the last token, reducing the prompt to a single-shot, single-token demonstration. According to the optimal single-layer transformer (eq. (4)), the estimated coefficient matrix is now $W' = \alpha_1 \alpha_2 Y \beta (X\beta)^\top$, which is rank-one. Therefore, task vectors are inherently limited in their expressiveness: *they can only replicate $1$-shot ICL, which is restricted to rank-one coefficient matrices*. This implication also naturally extends to multi-layer transformers.

While our analysis is conducted on linear-attention transformers, we demonstrate that similar learning patterns also emerge within practical LLMs. Specifically, we visualize the layer-wise information flow between tokens using saliency maps (Wang et al., 2023), where the saliency score for each attention matrix is computed as $S(A_l) = \sum_h |A_{l,h} \cdot \partial\mathcal{L}/\partial A_{l,h}|$, $A_{l,h}$ denotes the attention matrix of the $h$-th head at layer $l$, and $\mathcal{L}$ is the ICL loss (i.e., the cross-entropy loss for predicting $y_{\mathrm{test}}$).

Table 1: Comparison of the accuracies of **many-shot** ICL and task vector on bijection tasks (Llama-7B, $n = 10$). We use gray text to indicate accuracies lower than 60%.

| Task | Domain $\mathcal{X}$ | Domain $\mathcal{Y}$ | Example | $\mathcal{X} \to \mathcal{Y}$ | | $\mathcal{Y} \to \mathcal{X}$ | | $\mathcal{X} \leftrightarrow \mathcal{Y}$ | |
|---|---|---|---|---|---|---|---|---|---|
| | | | | ICL | TV | ICL | TV | ICL | TV |
| To Upper | $\{a, \cdots, z\}$ | $\{A, \cdots, Z\}$ | a $\to$ A | 1.00 | 0.91 | 1.00 | 0.99 | 1.00 | 0.55 |
| Translation | English | French | hello $\to$ bonjour | 0.83 | 0.84 | 0.82 | 0.70 | 0.54 | 0.35 |
| | English | Italian | hello $\to$ ciao | 0.84 | 0.78 | 0.82 | 0.74 | 0.70 | 0.47 |
| | English | Spanish | hello $\to$ hola | 0.92 | 0.88 | 0.89 | 0.75 | 0.64 | 0.43 |
| Linguistic | Present | Gerund | go $\to$ going | 0.99 | 0.95 | 1.00 | 0.97 | 0.80 | 0.41 |
| | Present | Past | go $\to$ went | 0.98 | 0.91 | 0.99 | 0.96 | 0.52 | 0.33 |
| | Present | Past Perfect | go $\to$ gone | 0.82 | 0.82 | 0.94 | 0.65 | 0.55 | 0.33 |
| | Singular | Plural | dog $\to$ dogs | 0.88 | 0.78 | 0.94 | 0.89 | 0.76 | 0.51 |
| Copy | $\{a, \cdots, z, A, \cdots, Z\}$ | | A $\to$ A | - | | - | | 1.00 | 0.98 |
| Antonym | Adjectives | | happy $\to$ sad | 0.89 | 0.83 | - | | 0.83 | 0.73 |

As demonstrated in Figures 3a and 3b, the saliency maps reveal certain patterns matching the ones of embedding concatenation and weighted summation. This suggests that real-world transformers implement a similar algorithm to solve ICL tasks and, consequently, inherit the same expressiveness limitation. The full saliency score maps are given in Appendix B.5.

To verify this, we construct a specialized class of ICL tasks, named bijection tasks. Specifically, given a bijective mapping from domain $\mathcal{X}$ to codomain $\mathcal{Y}$, one can combine it with its inverse mapping to form a new task that maps $\mathcal{X} \cup \mathcal{Y}$ onto itself. For instance, combining the "to uppercase" task with its inverse "to lowercase" yields a bijection task that maps each letter to its opposite case, and a valid ICL prompt takes the form: *"a $\to$ A, B $\to$ b, c $\to$ C, D $\to$"*. Note that this differs from task superposition (Xiong et al., 2024), as each input corresponds to a unique, well-defined output. We then establish a key limitation of rank-one coefficient matrices in addressing such tasks:

**Proposition 4.** *Let $x, y \in \mathbb{R}^d$ be non-zero vectors. Then the following are equivalent: (1) There exists a rank-one matrix $W \in \mathbb{R}^{d \times d}$ such that $y = Wx$ and $x = Wy$; (2) $x = y$ or $x = -y$.*

This result highlights that *rank-one coefficient matrices cannot solve general bijection tasks*, and are restricted to two special cases: the identity mapping ($x = y$), or the negation mapping ($x = -y$). We further verify this implication in real-world LLMs: in Table 1, both ICL and task vectors perform well on the original tasks and their inverses. But for bijection tasks, while ICL preserves performance in many cases, the task vector method consistently fails, confusing examples from the two domains and yielding near-random predictions (50%) (e.g., in "To Upper", task vectors predict the correct letter but fail to distinguish between uppercase and lowercase. See Appendix B.4 for further results). The only exceptions are Copy and Antonym, the special cases in Proposition 4.

Together, these findings empirically validate our main conjecture: **the task vector approach, which is restricted to one-shot ICL, is limited to rank-one mappings and cannot solve general ICL tasks (e.g., bijection tasks)**. While a variety of ICL tasks have been explored to assess the capabilities of task vectors (Hendel et al., 2023; Todd et al., 2024; Li et al., 2024), the fundamental limitation of task vectors in addressing these bijection tasks has not been previously identified.

## 5 Further Discussions

**Effect of Causal Attention and Dropout.** While task vectors naturally emerge in linear attention, their embeddings do not directly help minimize the ICL risk, as evidenced by the identical performance between pairwise and triplet formatted prompts (Figures 4a and 4b). Instead, we show that task vectors do contribute to optimization under token-wise dropout, acting as redundancies for in-context demonstrations that may be randomly dropped during training. This redundancy ensures that essential task information is preserved to facilitate inference despite partial context loss.

**Proposition 5.** *Under the same settings as Proposition 3, consider adding token-wise dropouts $O_l$:*

$$Z_l = Z_{l-1}O_l + \frac{1}{n} \text{Attn}_{V_l, Q_l}(Z_{l-1})O_l, \quad \text{where } O_l = \text{diag}(o_l^1, \cdots, o_l^{d_p}), \ o_l^i \overset{i.i.d.}{\sim} \text{Bern}(p).$$

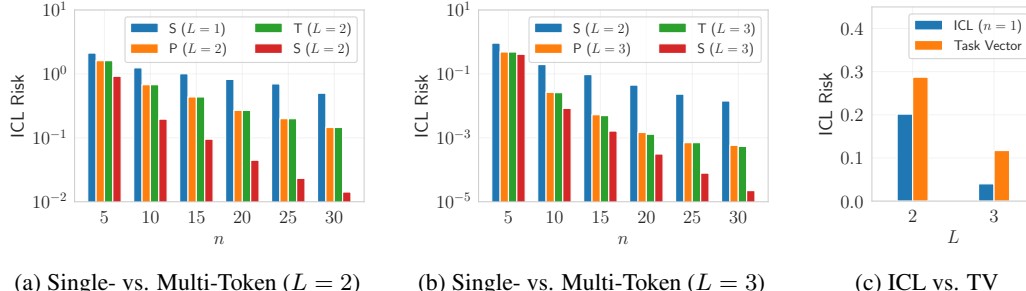

(a) Single- vs. Multi-Token ($L = 2$)   (b) Single- vs. Multi-Token ($L = 3$)   (c) ICL vs. TV

Figure 4: (a, b) Comparison of the best ICL risk achieved using single (S), pairwise (P), and triplet (T) formatted prompts. (c) Performance comparison between 1-shot ICL and task vector.

*Then any minimizer $\Lambda_4$ of the in-context risk $\mathcal{L}\big(\{V_l, Q_l\}_{l=1}^L\big)$ satisfies $(\Lambda_4)_{n+1,:} = 0$ and:*

$$(\Lambda_4)_{1:n,:} \propto \arg\min_{\Lambda} \; c_1 \|\Lambda\|_4^4 + c_2 \sum_{i=1}^n \|\Lambda_{i,:}\|_2^4 + c_3 \sum_{j=1}^{n+1} \|\Lambda_{:,j}\|_2^4 + c_4 \big\|\Lambda\Lambda^\top\big\|_F^2, \;\; s.t. \; \|\Lambda\|_F^2 = 1.$$

*where $c_1, \cdots, c_4$ are non-negative constants depending on $V_l$, $Q_l$, and $p$.*

This result suggests that dropout introduces additional higher-order regularization on the task vector weights, encouraging them to distribute more uniformly across demonstrations. Furthermore, when considering causal attention (i.e., enforcing $\Lambda_4$ to be upper-triangular), it induces a decaying weight pattern from later to earlier demonstrations, which exactly matches the practical behavior observed in practical transformer models (as evidenced in Figure 3c).

**Decoding the Vocabulary of Task Vectors.** Multiple prior works (Hendel et al., 2023; Todd et al., 2024) have observed an interesting phenomenon that, when task vectors are directly decoded through the final classification layer, the top tokens often belong to the output space of the current task (see Table 4 in the Appendix). Our theoretical analysis provides a natural explanation for this: assuming a $2d$-dimensional hidden state space partitioned into input ($x_i$) and output ($y_i$) halves, the output half of task vectors then encodes weighted summations of $y_i$. Since the final prediction relies on the output half, decoding a task vector yields a combination of $y_i$, which is likely lying in the output space. This observation suggests that practical LLMs adopt a similar hidden-state partition.

**Extra EOS Tokens.** In our previous analysis, we consistently imposed an additional zero token at the end of the input prompt. While this token can be interpreted as an EOS token in practical models, such a design choice is uncommon in standard ICL tasks. We justify this modeling decision with:

**Proposition 6** (Informal). *Given any $L$-layer, 1-head, $d$-dimensional linear-attention model with EOS, there exists an equivalent $L$-layer, 2-head, $2d$-dimensional model operating without EOS.*

This equivalence suggests that the same learning dynamics can be realized through multi-head architectures without relying on explicit EOS tokens. Specifically, the first head is dedicated to task vector formation, while the other handles ICL prediction. This separation allows the model to retain the functional role of the EOS token implicitly within its hidden states.

## 6 EXPERIMENTAL STUDIES

### 6.1 SYNTHETIC RESULTS WITH RANDOM LINEAR REGRESSION

In this section, we validate our critical points analysis with synthetic linear regression tasks. Specifically, we examine the achievable ICL risk of linear-attention models with single-token (eq. (3)), pairwise (eq. (7)), and triplet (eq. (9)) demonstrations. We set the input dimension to $d = 4$ and $P_x = P_w = \mathcal{N}(0, I_d)$. For each setting, we train multiple models with different random seeds and report the minimum ICL risk achieved as a proxy for the global optimum. The comparative results across different numbers of layers $L$ and demonstration formats are shown in Figures 4a and 4b.

These results support our theoretical analysis: when trained with pairwise or triplet demonstrations, the model recovers the GD++ algorithm similar to the single-token case. Notably, the performance

Table 2: Accuracy comparison between **few-shot** ICL (Baseline), the task vector method (TaskV), and our strategy (TaskV-M). The experiment is conducted on Llama-13B with $n = 10$.

| Method | | Knowledge | Algorithmic | Translation | Linguistic | Bijection | Average |
|---|---|---|---|---|---|---|---|
| 0-shot | Baseline | $6.90 \pm 2.08$ | $15.60 \pm 1.72$ | $7.00 \pm 1.65$ | $12.44 \pm 1.74$ | $8.27 \pm 1.33$ | $10.28 \pm 0.98$ |
| | TaskV | $\mathbf{68.80} \pm 2.66$ | $\mathbf{86.20} \pm 1.61$ | $\mathbf{73.53} \pm 0.91$ | $\mathbf{85.24} \pm 1.80$ | $\mathbf{50.67} \pm 2.32$ | $\mathbf{72.26} \pm 1.01$ |
| 1-shot | Baseline | $69.50 \pm 3.86$ | $73.67 \pm 1.56$ | $57.80 \pm 2.01$ | $56.22 \pm 1.57$ | $44.76 \pm 2.44$ | $58.11 \pm 0.63$ |
| | TaskV | $79.50 \pm 2.35$ | $88.47 \pm 0.75$ | $\mathbf{80.67} \pm 2.56$ | $\mathbf{89.11} \pm 0.84$ | $60.44 \pm 2.07$ | $78.79 \pm 0.77$ |
| | TaskV-M | $\mathbf{81.30} \pm 2.80$ | $\mathbf{89.53} \pm 0.65$ | $80.13 \pm 2.14$ | $88.71 \pm 0.62$ | $\mathbf{61.78} \pm 0.96$ | $\mathbf{79.34} \pm 0.37$ |
| 2-shot | Baseline | $78.80 \pm 3.30$ | $85.07 \pm 1.37$ | $75.67 \pm 2.64$ | $76.80 \pm 1.18$ | $56.49 \pm 2.87$ | $72.92 \pm 0.59$ |
| | TaskV | $84.60 \pm 2.11$ | $88.40 \pm 0.68$ | $\mathbf{84.33} \pm 0.92$ | $\mathbf{90.13} \pm 0.92$ | $62.44 \pm 2.16$ | $80.82 \pm 0.42$ |
| | TaskV-M | $\mathbf{85.70} \pm 1.63$ | $\mathbf{89.27} \pm 1.10$ | $84.13 \pm 1.15$ | $89.64 \pm 0.86$ | $\mathbf{64.49} \pm 2.02$ | $\mathbf{81.48} \pm 0.37$ |
| 3-shot | Baseline | $86.20 \pm 2.69$ | $88.07 \pm 1.06$ | $80.00 \pm 1.67$ | $84.04 \pm 1.19$ | $62.18 \pm 1.52$ | $78.51 \pm 0.42$ |
| | TaskV | $90.20 \pm 2.23$ | $88.67 \pm 0.89$ | $\mathbf{86.27} \pm 2.31$ | $92.31 \pm 0.48$ | $66.53 \pm 0.94$ | $83.53 \pm 0.41$ |
| | TaskV-M | $\mathbf{90.30} \pm 1.50$ | $\mathbf{89.87} \pm 0.83$ | $86.07 \pm 2.17$ | $\mathbf{92.36} \pm 0.72$ | $\mathbf{68.13} \pm 0.76$ | $\mathbf{84.15} \pm 0.52$ |
| 4-shot | Baseline | $84.80 \pm 2.06$ | $88.07 \pm 0.61$ | $83.27 \pm 1.82$ | $88.89 \pm 1.91$ | $67.16 \pm 1.47$ | $81.52 \pm 0.66$ |
| | TaskV | $88.70 \pm 1.69$ | $89.53 \pm 1.34$ | $86.27 \pm 1.08$ | $\mathbf{92.76} \pm 0.54$ | $70.44 \pm 1.35$ | $84.66 \pm 0.39$ |
| | TaskV-M | $\mathbf{89.60} \pm 1.43$ | $\mathbf{91.00} \pm 1.01$ | $\mathbf{87.20} \pm 0.62$ | $92.36 \pm 1.44$ | $\mathbf{72.53} \pm 0.94$ | $\mathbf{85.64} \pm 0.29$ |

of $L$-layer models with pairwise (P) and triplet (T) demonstrations closely aligns, indicating a shared underlying learning pattern. Moreover, their performance consistently lies between that of single-token (S) case $L$-layer and $(L-1)$-layer models. The observed improvement over the $(L-1)$-layer single-token baselines comes from the additional GD++ performed solely on $x_i$ tokens in the first layer, effectively acting as a "half-step" of gradient descent.

We then reproduce the task vector method in linear models. Specifically, we extract the hidden state of the final ($\rightarrow$) token from triplet demonstrations after the first layer, and inject this vector into zero-shot prompts consisting of $x_{\text{test}}$ only. To simulate the effect of layer normalization, we normalize the task vectors before inference and the output vectors before ICL risk evaluation. As shown in Figure 4c, the performance of task vectors is highly related to that of standard 1-shot ICL. This validates our conjecture that the injected task vector effectively acts as a single demonstration.

## 6.2 Enhancing the Task Vector Method

We further explore an enhancement to the original task vector method. According to our previous analysis, a single injected task vector may not provide sufficient information for inference on complex tasks (e.g., bijection tasks). Moreover, in linear-attention models, each ($\rightarrow$) token functions as an individual in-context demonstration during the gradient descent phase and thus contributes equally to the ICL risk. Motivated by this, we extend the standard task vector method, which modifies only the final arrow token, and propose a multi-vector variant that injects into every single arrow token in few-shot prompts. This enriched injection scheme enables the model to leverage multiple new demonstrations, thereby providing a more informative and distributed context for prediction.

We compare our multi-vector injection strategy (TaskV-M) against standard $N$-shot ICL (Baseline) and the original task vector method (TaskV). Note that Baseline uses few-shot ICL and TaskV is injecting into few-shot prompts, which are different from the settings in Table 1 which uses many-shot prompts for ICL and zero-shot prompts for task vectors. For each $N$-shot prompt, we generate $N+1$ distinct ICL prompts to produce $N+1$ task vectors, which are then used to replace the embeddings of all arrow tokens in the input. For each task, performance is evaluated over 50 randomly sampled prompts, with mean accuracy and standard deviation reported across 5 independent trials. The final results, summarized in Table 2, span a diverse set of ICL task types, showing that TaskV-M consistently outperforms TaskV, especially the challenging bijection tasks. While the improvement is not dramatic, we believe that the current results sufficiently demonstrate the potential of multi-vector injection, thereby providing insights for the design of future ICL or task vector methods.

## 7 CONCLUSION, LIMITATIONS, AND FUTURE WORKS

This paper proposes a plausible explanation for the emergence and functionality of task vectors in ICL. We support this conjecture with both empirical observations and theoretical analysis, demonstrating how task vectors naturally arise under ICL-style training prompts, and why this method inherently fails on general ICL tasks beyond rank-one mappings. Our work provides a new perspective on the underlying mechanisms and offers a promising direction for interpreting intermediate hidden states in modern transformer-based language models.

While our analysis provides new insights into the emergence and functionality of task vectors, it is primarily conducted on simplified linear-attention transformers and synthetic tasks, which may not fully capture the complexity of real-world LLMs. Moreover, our theoretical framework focuses solely on critical point analysis, and there is still a lack of convergence guarantee or sample complexity analysis to fully understand the learning dynamics during model pretraining.

Future directions of this work may include: (1) extending the current theoretical framework to causal and multimodal settings; (2) exploring how richer architectures (e.g., non-linear attention) or training objectives (e.g., auto-regressive loss) influence the behavior of task vectors; (3) synthesizing orthogonal enhancements of the task vector method (e.g., function vectors (Todd et al., 2024) and in-context vectors (Liu et al., 2024)), and extending to more complex reasoning tasks.

### ACKNOWLEDGMENTS

This work was supported by the Natural Science Foundation under grants IIS-2312840 and IIS-2402952. We would like to thank the anonymous reviewers for their valuable suggestions.

### ETHICS STATEMENT

This work advances the theoretical understanding of in-context learning and task vector mechanisms, which can lead to more efficient and interpretable language models. By enabling faster inference through task vectors, it may reduce the computational cost and energy consumption of large-scale deployment, thereby making AI systems more accessible and environmentally sustainable. Improved interpretability could also enhance trust and transparency in AI applications across education, healthcare, and other socially beneficial domains.

As task vector methods improve efficiency and transferability, they may also be misused to replicate or extract functionality from proprietary models without authorization, raising concerns around model intellectual property. Additionally, while interpretability is often framed as a benefit, deeper insights into model internals could be exploited to engineer adversarial inputs or extract sensitive training data. Careful consideration and mitigation strategies are essential to ensure that such work aligns with the broader goals of safe and beneficial AI.

### REPRODUCIBILITY STATEMENT

We provide complete proofs for our main theoretical results in Appendices C and D, experimental details about the dataset and implementation in Appendix B, and full source codes to reproduce our experimental results at `https://github.com/Yuxin-Dong/ICL-TaskVector`.

### USAGE OF LLMS

We used LLMs only to improve grammar and polish academic writing. All technical ideas, proofs, experiments, and conclusions were entirely conceived and verified by the authors.

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

# A  ADDITIONAL DISCUSSIONS

## A.1  SUMMARY OF MATHEMATICAL NOTATIONS

Table 3: Summary of key mathematical notations used throughout the paper.

| Notation | Description |
|---|---|
| $n$ | Number of demonstrations in the input prompt |
| $L$ | Number of transformer layers |
| $d$ | Dimension of covariate and response embeddings |
| $d_p$ | Prompt length (depends on demonstration structure) |
| $\text{Attn}_{V,Q}$ | Linear-attention layer with parameter $V$, $Q$ |
| TF | Linear-attention model by stacking linear-attention layers |
| $x_i \in \mathbb{R}^d$ | Covariate (input) of the $i$-th demonstration |
| $y_i \in \mathbb{R}^d$ | Response (output) of the $i$-th demonstration |
| $X, Y \in \mathbb{R}^{d \times n}$ | Matrices of covariates and responses for $n$ demonstrations |
| $x_{\text{test}}, y_{\text{test}}$ | Query covariate and ground-truth response |
| $w_j \in \mathbb{R}^d$ | $j$-th regression coefficient vector |
| $W \in \mathbb{R}^{d \times d}$ | Coefficient matrix, $W = [w_1, \cdots, w_d]^\top$ |
| $Z_0 \in \mathbb{R}^{2d \times d_p}$ | Input prompt embeddings before the transformer |
| $Z_l \in \mathbb{R}^{2d \times d_p}$ | Hidden states after the $l$-th layer |
| $P \in \mathbb{R}^{d_p \times d_p}$ | Positional encoding matrix |
| $V_l, Q_l$ | Value and key-query matrices of the $l$-th attention layer |
| $A_l, B_l, C_l, D_l$ | Block components of $V_l, Q_l$ in layer $l$ |
| $\Lambda_k$ | Sub-block matrices of $D_l$ used in critical point analysis |
| $\mathcal{L}$ | In-context learning loss (ICL risk) |
| $\mathcal{M}(M)$ | Set of masked matrices with binary mask $M$ |
| $\mathcal{S}_I, \mathcal{S}_\Sigma, \mathcal{S}_P$ | Structured sets of matrices defining critical points |
| $z_{\text{tv}}$ | Task vector extracted from an arrow ($\rightarrow$) token |
| $\beta \in \mathbb{R}^n$ | Weight vector for task vector formation |

## A.2  ADDITIONAL RELATED WORKS

**In-Context Learning in Attention-based LLMs.** The ability of LLMs to learn from examples provided in the input prompt, without updating parameters, has attracted wide attention since the discovery of ICL in GPT-3 (Brown et al., 2020). A growing body of theoretical work has sought to explain this phenomenon. Early analyses show that transformer attention layers can implement gradient descent–like algorithms over linear regression objectives (Garg et al., 2022; Akyürek et al., 2023; Von Oswald et al., 2023a; Ahn et al., 2023; Wu et al., 2024), while others investigate sample complexity and generalization behavior (Xie et al., 2022; Chan et al., 2022; Shen et al., 2024; Von Oswald et al., 2023b; Deutch et al., 2024). These works collectively suggest that ICL is closely tied to the inductive biases of the attention mechanism, but do not fully explain how higher-level abstractions of tasks are formed or encoded in LLMs.

**The Task Vector Method in ICL.** Task vectors have recently been proposed as an abstraction of ICL demonstrations into compact hidden-state representations. Hendel et al. (2023) introduced task vectors as hidden states extracted from the last arrow token in triplet prompts, enabling zero-shot transfer by injecting them into new contexts. Concurrent works developed similar notions, such as function vectors (Todd et al., 2024) and in-context vectors (Liu et al., 2024). These studies show that task vectors accelerate inference and sometimes match the effectiveness of ICL with fewer demonstrations. However, they remain largely empirical, without a clear theoretical explanation of how or why such vectors encode task information.

Subsequent research has expanded the scope and utility of task vectors. Yang et al. (2025) demonstrates that task vectors naturally emerge even in small transformers trained from scratch with synthetic data, suggesting that their formation is an inherent property of attention-based architectures.

Table 4: Top 20 tokens with the highest output probability by decoding the task vector, results from (Hendel et al., 2023). We underline the tokens in the output space of the current task.

| Model | Task | Tokens |
|-------|------|--------|
| GPT-J 6B | Prev Letter | b, c, v, g, s, name, i, ro, n, j, d, t, A, ai, com, m, ust, test, active, k |
| | French to English | other, name, the, true, is, social, s, active, time, car, type, money, F, force, a, public, heart, one, ms, life |
| | Present to Gerund | getting, storing, working, moving, playing, doing, making, driving, shooting, picking, being, sending, putting, selling, watching, changing, taking, collecting, feeding, reading |
| | Country to Capital | London, Paris, New, West, Berlin, South, Tokyo, San, Chicago, City, Moscow, Jerusalem, Amsterdam, Philadelphia, East, Madrid, Vienna, Beijing, Mexico, Germany |

Li et al. (2024) shows that aggregating hidden states across layers and multiple arrow tokens leads to stronger task representations. Kang et al. (2025) proposes to generate task vectors conditioned on each input query. Beyond text, task vectors have also been applied in vision (Hojel et al., 2024; Peng et al., 2024) and multimodal models (Huang et al., 2024; Luo et al., 2025), where they enable flexible transfer across modalities. Han et al. (2025) connects the performance of task vectors by task decodability, defined by the similarity between task vectors from different ICL tasks. These works highlight the empirical utility of task vectors but stop short of explaining their inner mechanisms.

**Explaining the Task Vector Method.** Task vectors were initially conjectured to encapsulate the complete knowledge of the current task (Hendel et al., 2023). However, this view fails to account for their inconsistent performance across tasks of varying complexity. Empirical observations further suggest that directly decoding task vectors typically yields tokens from the task output space (Todd et al., 2024), rather than explicit task descriptions (Merullo et al., 2024). Concurrent work by Bu et al. (2025) analyzes the learning dynamics of 1-layer transformers with ICL-style prompts, explaining the utility of task vectors through a word2vec-like scheme (i.e., the existence of a vector $z_t$ for task $t$ such that $y \approx z_t + x$ for all input-output pairs $(x, y)$). While insightful, this characterization is restricted to additive translation tasks, single-token prompts, and single-layer architectures, limiting its generality. By contrast, our analysis encompasses richer prompt structures, including pairwise and triplet formats that better reflect practical ICL settings. Moreover, our critical point characterization extends beyond 1-layer models, and our linear regression formulation captures a broader spectrum of ICL tasks. Complementing our findings, Tikhonov et al. (2025) independently shows that standard task vectors lack sufficient expressiveness for complex ICL tasks, reinforcing our conclusion that task vectors are fundamentally constrained by rank-one mappings.

A.3 JUSTIFICATION OF THE BLOCK-DIAGONAL ASSUMPTION

In our main analysis, we impose an assumption on the trainable parameters of linear-attention layers, such that the $V_l$ and $Q_l$ matrices are block-diagonal in eq. (6). This block-diagonal formulation is a widely adopted assumption in theoretical studies of ICL for transformer models, as it facilitates tractable analysis (Ahn et al., 2023; Mahankali et al., 2024; Wu et al., 2024; Zhang et al., 2024). Prior work by Ahn et al. (2023) demonstrates that the global minimizer of single-layer linear-attention transformers indeed exhibits such a block-diagonal structure. Although finding exact solutions for multi-layer transformers is more involved, it is reasonable to conjecture that similar structural patterns hold. Empirically, we observe that when optimizing the full matrices, gradient-based training also tends to converge to block-diagonal solutions.

Intuitively, given the high dimensionality of hidden states in modern LLMs, it is plausible to assume that the $x_i$ and $y_i$ components can be projected into orthogonal or nearly orthogonal subspaces when mixed in the hidden state space. This motivates a decomposition of the projection matrices $V_l$ and $Q_l$ into two separate parts that operate independently on $x_i$ and $y_i$, which can be equivalently formulated as the block-diagonal structures.

## A.4 Inseparable Covariates and Responses

In our main analysis, we assume that $x_i$ and $y_i$ embeddings are linearly separable, allowing the addition $x_i + y_i$ to act a concatenation operation. However, recognizing that this assumption does not generally hold for real-world transformers, we extend our analysis to the following setting, where $x_i$ and $y_i$ are no longer linearly separable. While this still imposes a $2d$-dimensional requirement on the hidden space, such a constraint is easily satisfied in practical transformers, given the high dimensionality of their internal representations.

$$Z_0 = \begin{bmatrix} 0 & 0 & \cdots & 0 & 0 & 0 & 0 \\ x_1 & y_1 & \cdots & x_n & y_n & x_{\text{test}} & 0 \end{bmatrix} \in \mathbb{R}^{(2d) \times (2n+2)}. \tag{11}$$

We slightly modify the sparsity constraints for the first layer, and require $(D_0)_{2i,:} = 0$ for $i \in [n+1]$:

$$V_0 = \begin{bmatrix} 0 & A_0 \\ 0_{d \times d} & 0 \end{bmatrix}, \quad Q_0 = \begin{bmatrix} 0_{2d \times 2d} & 0 \\ 0 & D_0 \end{bmatrix}, \quad \text{where } A_0 \in \mathbb{R}^{d \times d}, \ D_0 \in \mathbb{R}^{d_p \times d_p}. \tag{12}$$

With these conditions, we are ready to establish the critical points for inseparable demonstrations. Note that $V_0$ and $Q_0$ do not involve $B_0$ and $C_0$, so the sequences $B$ and $C$ have size $L - 1$.

**Theorem 7.** *Under the same settings as Theorem 1, define $\mathcal{S}_I, \mathcal{S}_\Sigma \subset \mathbb{R}^{d \times d}$ and $\mathcal{S}_P \subset \mathbb{R}^{d_p \times d_p}$ as*

$$\mathcal{S}_I = \{\lambda I_d \mid \lambda \in \mathbb{R}\}, \quad \mathcal{S}_\Sigma = \{\lambda \Sigma^{-1} \mid \lambda \in \mathbb{R}\}, \quad \mathcal{S}_P = \{\text{diag}(I_n \otimes \Lambda_1, \Lambda_2) \mid \Lambda_1, \Lambda_2 \in \mathbb{R}^{2 \times 2}\}.$$

*Consider optimizing an L-layer linear transformer with inseparable pairwise demonstrations and parameter configuration given in eq. (12) for the first layer and eq. (6) for the remaining layers, then*

$$\inf_{A \in \mathcal{S}_I^L, \ B \in \mathcal{S}_I^{L-1}, \ C \in \mathcal{S}_\Sigma^{L-1}, \ D \in \mathcal{S}_P^L} \sum_{H \in A \cup B \cup C \cup D} \left\| \nabla_H \mathcal{L}\big(\{V_l, Q_l\}_{l=1}^L\big) \right\|_F^2 = 0.$$

This result suggests that for inseparable demonstrations, the first layer performs a functionally similar concatenation operation by "moving" the embedding of each $x_i$ to the corresponding $y_i$ position. This enables the model to reconstruct the single-token structure without linear separability.

## A.5 Last Task Vector Weights the Most

While our analysis of linear-attention models suggests that each formed task vector (i.e., the hidden state at each arrow token) contributes equally to the final prediction, this assumption does not fully hold in practical LLMs. As demonstrated by the conflicting tasks experiment in (Hendel et al., 2023), injecting a task vector from task $B$ into an ICL prompt designed for task $A$ causes the model to predominantly perform task $B$. This behavior indicates that LLMs largely rely on the last arrow token to determine the task identity. We attribute this to the causal attention mechanism used in practical LLMs, which is not captured by our current theoretical analysis. In causal attention, only the final arrow token can aggregate information from the entire preceding context, making it the most informative and influential for prediction. This explains why our multi-vector strategy offers modest, though consistent, performance gains. The improvement suggests that intermediate arrow tokens do participate in the inference process, albeit less effectively. Enhancing how LLMs utilize information from all arrow tokens remains a promising direction for improving task vector accuracy and robustness.

## B Experiment Details and Additional Results

In this section, we present experiment details and additional results not included in the main text due to space limitations. Our experiments are conducted on an A100 40G GPU. It takes around 30 GPU hours to fully reproduce our results.

### B.1 SYNTHETIC EXPERIMENTS ON LINEAR-ATTENTION MODELS

We consider training linear-attention models on random linear regression instances. We take embedding dimension $d = 4$, and the distributions for generating $x_i$ and $w_i$ are both $P_x = P_w = \mathcal{N}(0, I_d)$. We optimize the ICL risk for $L$-layer linear-attention models with $n$ in-context demonstrations using AdamW, where $L \in [3]$ and $n \in [5, 30]$. Each gradient step is computed from a batch size of 1000. We additionally apply $\ell_1$ regularization to simplify the found solutions. For training efficiency and stability, we restrict the $A_l$, $B_l$, and $C_l$ matrices to $\mathcal{S}_I$ during training, and initialize $D_l \in \mathbb{R}^{d_p \times d_p}$ with i.i.d. Gaussian matrices. For each case, we train 40 models with different random seeds, and report the minimum achieved ICL risk to approximate the global minimum.

To reproduce the task vector mechanism, we focus on models trained with triplet-formatted prompts. The training procedure is identical to the above. For inference, we restrict $P_w$ to rank-one coefficient matrices, by letting $W = w_1 w_2^\top$, where $w_1, w_2 \sim \mathcal{N}(0, I_d)$. We first generate normal ICL prompts to generate task vectors as the hidden states of the last arrow token after the first attention layer, and then inject them into zero-shot prompts after normalization. The final outputs $\hat{y}_{\text{test}}$ are taken as the output of these injected zero-shot prompts after being processed with the same transformer model. We compute the final risk as $\mathbb{E} \left\| \frac{\hat{y}_{\text{test}}}{\|\hat{y}_{\text{test}}\|} + \frac{y_{\text{test}}}{\|y_{\text{test}}\|} \right\|$ to simulate the layer normalization blocks in practical LLMs. The reported scores are averaged for $n \in [5, 30]$.

### B.2 EXPERIMENTS ON PRACTICAL LLMS

**Datasets.** Following the settings of the original task vector method (Hendel et al., 2023), our study covers 33 tasks in 5 categories. The detailed description for each task is provided in Table 5.

**Prompt Template.** The template used to construct ICL demonstrations is "Example:$\{x_i\} \rightarrow \{y_i\}$, where $x_i$ and $y_i$ are subsequently replaced by the input and output of the semantic mapping. For the query part, $y_i$ is omitted from the prompt. After concatenating each demonstration with "\n", an example of the full input prompt is:

$$\text{Example:}\{x_1\} \rightarrow \{y_1\} \backslash \text{n} \cdots \text{Example:}\{x_n\} \rightarrow \{y_n\} \backslash \text{nExample:}\{x_{\text{test}}\} \rightarrow \quad (13)$$

**Evaluation.** To evaluate the $N$-shot performance, we generate $50 \times (N + 1)$ i.i.d. prompts for each task with number of demonstrations $n = 10$ for task vector extraction. The hidden states of the last $\rightarrow$ token, which is also literally the last token in the prompt, are recorded for every layer in the transformer. Thereafter, we generate another 50 i.i.d. prompts with $N$ demonstrations, where $x_{\text{test}}$ is selected to be distinct from the previous chosen ones. The final accuracy is measured by whether the next word predicted matches the expected answer. The performance of the standard ICL method (Baseline) is acquired by inferring without interference. For the task vector method (TaskV) and our multi-vector variant (TaskV-M), the extracted task vectors are injected to replace the hidden states of the arrow $\rightarrow$ tokens at a specified layer $l$. For TaskV, only the last arrow token is injected, while for TaskV-M, each of the $N + 1$ arrow tokens is injected with the $N + 1$ extracted task vectors for the same task. The performance is reported for the layer $l \in L$ achieving the highest accuracy. For each case, the mean and standard deviation are evaluated through 5 independent trials.

**Additional Results.** Besides Llama-13B, we also observe consistent accuracy improvement of our TaskV-M method on the Pythia-12B model, as reported in Table 6.

While the performance gains of TaskV-M over TaskV are not dramatic across all ICL tasks, the goal of TaskV-M is not to surpass state-of-the-art ICL techniques but to demonstrate that the task vector framework can be systematically extended by injecting multiple vectors simultaneously. This is especially valuable for complex tasks that inherently require higher-rank representations. Our results on bijection tasks clearly validate this motivation: TaskV-M yields notable improvements over the standard TaskV method. For other simpler tasks, the marginal gains from TaskV-M suggest that the expressiveness of $W$ may not be the primary performance bottleneck. We believe these insights facilitate the design of future ICL and task vector methods.

### B.3 ANOTHER MULTI-VECTOR INJECTION VARIANT

In our main experiments, we implement TaskV-M by extracting $N + 1$ task vectors from the same number of different prompts. Another possible implementation for TaskV-M is to extract multiple

Table 5: Descriptions of the tasks used in our empirical studies.

| Category | Task | Example | Description |
|---|---|---|---|
| Knowledge | Contry to Capital | France → Paris | Output the capital city of the given country. |
| | Person to Language | Macron → French | Output the native language of the given person. |
| | Location to Continent | Paris → Europe | Output the corresponding continent of the given location. |
| | Religion | Saladin → Muslim | Output the associated religion of the given location or person. |
| Algorithmic | List First | [a,b,c] → a | Output the first item in the given list. |
| | List Last | [a,b,c] → c | Output the last item in the given list. |
| | Next Letter | a → b | Output the next letter of the given letter in the alphabet. |
| | Prev Letter | b → a | Output the previous letter of the given letter in the alphabet. |
| | To Upper | a → A | Output the corresponding uppercase letter of the given lowercase letter. |
| | To Lower | A → a | Output the corresponding lowercase letter of the given uppercase letter. |
| Translation | English to French | hello → bonjour | Translate the given word in English to French. |
| | English to Italian | hello → ciao | Translate the given word in English to Italian. |
| | English to Spanish | hello → hola | Translate the given word in English to Spanish. |
| | French to English | bonjour → hello | Translate the given word in French to English. |
| | Italian to English | ciao → hello | Translate the given word in Italian to English. |
| | Spanish to English | hola → hello | Translate the given word in Spanish to English. |
| Linguistic | Present to Gerund | go → going | Output the corresponding gerund form of the given verb in present simple tense. |
| | Present to Past | go → went | Output the corresponding past simple form of the given verb in present simple tense. |
| | Present to Past Perfect | go → gone | Output the corresponding past perfect form of the given verb in present simple tense. |
| | Gerund to Present | going → go | Output the corresponding present simple form of the given verb in gerund form. |
| | Past to Present | went → go | Output the corresponding present simple form of the given verb in past simple tense. |
| | Past Perfect to Present | gone → go | Output the corresponding present simple form of the given verb in past perfect tense. |
| | Singular to Plural | dog → dogs | Output the corresponding plural form of the given noun in singular form. |
| | Plural to Singular | dogs → dog | Output the corresponding singular form of the given noun in plural form. |
| | Antonym | happy → sad | Output the antonym of the given adjective. |
| Bijection | To Upper & Lower | a ↔ A | Output the given letter in uppercase if it is in lowercase, and vice versa. |
| | English & French | hello ↔ bonjour | Translate the given word to French if it is in English, and vice versa. |
| | English & Italian | hello ↔ ciao | Translate the given word to Italian if it is in English, and vice versa. |
| | English & Spanish | hello ↔ hola | Translate the given word to Spanish if it is in English, and vice versa. |
| | Present & Gerund | go ↔ going | Output the given verb in gerund form if it is in present simple tense, and vice versa. |
| | Present & Past | go ↔ went | Output the given verb in past simple form if it is in present simple tense, and vice versa. |
| | Present & Past Perfect | go ↔ gone | Output the given verb in past perfect form if it is in present simple tense, and vice versa. |
| | Singular & Plural | dog ↔ dogs | Output the given noun in plural form if it is in singular form, and vice versa. |

Table 6: Accuracy comparison between standard ICL (Baseline), the task vector method (TaskV), and our strategy (TaskV-M). The experiment is conducted on Pythia-12B with $n = 10$.

| | Method | Knowledge | Algorithmic | Translation | Linguistic | Bijection | Average |
|---|---|---|---|---|---|---|---|
| 0-shot | Baseline | $6.60 \pm 1.59$ | $14.07 \pm 1.45$ | $8.60 \pm 0.68$ | $12.53 \pm 1.57$ | $10.31 \pm 0.70$ | $10.82 \pm 0.48$ |
| | TaskV | $\mathbf{63.30} \pm 2.62$ | $\mathbf{84.73} \pm 1.22$ | $\mathbf{62.07} \pm 0.98$ | $\mathbf{82.58} \pm 1.22$ | $\mathbf{42.27} \pm 0.92$ | $\mathbf{66.40} \pm 0.96$ |
| 1-shot | Baseline | $61.80 \pm 5.45$ | $72.80 \pm 1.15$ | $43.27 \pm 2.92$ | $57.07 \pm 1.15$ | $41.91 \pm 2.83$ | $53.95 \pm 1.02$ |
| | TaskV | $76.40 \pm 2.40$ | $\mathbf{84.20} \pm 1.05$ | $\mathbf{71.47} \pm 1.41$ | $\mathbf{87.16} \pm 2.04$ | $53.11 \pm 2.37$ | $73.59 \pm 0.79$ |
| | TaskV-M | $\mathbf{77.70} \pm 2.52$ | $83.73 \pm 1.37$ | $71.00 \pm 1.48$ | $86.80 \pm 1.59$ | $\mathbf{53.87} \pm 2.90$ | $\mathbf{73.68} \pm 0.90$ |
| 2-shot | Baseline | $70.30 \pm 3.71$ | $82.13 \pm 0.54$ | $60.80 \pm 1.81$ | $81.16 \pm 1.57$ | $50.76 \pm 2.17$ | $68.41 \pm 0.64$ |
| | TaskV | $80.30 \pm 2.46$ | $\mathbf{87.00} \pm 1.63$ | $76.13 \pm 3.77$ | $89.33 \pm 0.70$ | $58.67 \pm 2.44$ | $77.41 \pm 0.50$ |
| | TaskV-M | $\mathbf{81.60} \pm 1.56$ | $86.47 \pm 0.40$ | $\mathbf{77.27} \pm 2.53$ | $\mathbf{89.51} \pm 0.88$ | $\mathbf{59.24} \pm 2.48$ | $\mathbf{77.87} \pm 0.76$ |
| 3-shot | Baseline | $77.60 \pm 2.40$ | $81.87 \pm 0.81$ | $68.13 \pm 2.02$ | $86.31 \pm 1.93$ | $55.73 \pm 1.60$ | $73.20 \pm 0.31$ |
| | TaskV | $84.00 \pm 2.76$ | $86.33 \pm 1.17$ | $\mathbf{79.53} \pm 2.27$ | $92.00 \pm 0.67$ | $58.76 \pm 1.53$ | $79.06 \pm 0.67$ |
| | TaskV-M | $\mathbf{85.40} \pm 2.31$ | $\mathbf{87.07} \pm 1.18$ | $78.13 \pm 1.86$ | $\mathbf{92.84} \pm 0.68$ | $\mathbf{59.56} \pm 1.27$ | $\mathbf{79.54} \pm 0.35$ |
| 4-shot | Baseline | $78.40 \pm 1.83$ | $82.73 \pm 0.44$ | $72.40 \pm 1.24$ | $88.89 \pm 1.25$ | $57.91 \pm 1.46$ | $75.46 \pm 0.64$ |
| | TaskV | $83.80 \pm 1.12$ | $87.60 \pm 1.81$ | $\mathbf{80.20} \pm 2.39$ | $\mathbf{92.18} \pm 0.96$ | $59.38 \pm 0.47$ | $79.59 \pm 0.62$ |
| | TaskV-M | $\mathbf{84.30} \pm 1.50$ | $\mathbf{88.13} \pm 0.81$ | $80.00 \pm 2.67$ | $91.87 \pm 1.25$ | $\mathbf{60.31} \pm 0.86$ | $\mathbf{79.87} \pm 0.51$ |

Table 7: Accuracy comparison between few-shot ICL (Baseline), the task vector method (TaskV), the multi-vector method (TaskV-M), and the single-prompt variant (TaskV-MS). The experiment is conducted on Llama-13B with $n = 10$.

| | Method | Knowledge | Algorithmic | Translation | Linguistic | Bijection | Average |
|---|---|---|---|---|---|---|---|
| 0-shot | Baseline | $6.90 \pm 2.08$ | $15.60 \pm 1.72$ | $7.00 \pm 1.65$ | $12.44 \pm 1.74$ | $8.27 \pm 1.33$ | $10.28 \pm 0.98$ |
| | TaskV | $\mathbf{68.80} \pm 2.66$ | $\mathbf{86.20} \pm 1.61$ | $\mathbf{73.53} \pm 0.91$ | $\mathbf{85.24} \pm 1.80$ | $\mathbf{50.67} \pm 2.32$ | $\mathbf{72.26} \pm 1.01$ |
| 1-shot | Baseline | $69.50 \pm 3.86$ | $73.67 \pm 1.56$ | $57.80 \pm 2.01$ | $56.22 \pm 1.57$ | $44.76 \pm 2.44$ | $58.11 \pm 0.63$ |
| | TaskV | $79.50 \pm 2.35$ | $88.47 \pm 0.75$ | $\mathbf{80.67} \pm 2.56$ | $\mathbf{89.11} \pm 0.84$ | $60.44 \pm 2.07$ | $78.79 \pm 0.77$ |
| | TaskV-M | $\mathbf{81.30} \pm 2.80$ | $\mathbf{89.53} \pm 0.65$ | $80.13 \pm 2.14$ | $88.71 \pm 0.62$ | $\mathbf{61.78} \pm 0.96$ | $\mathbf{79.34} \pm 0.37$ |
| | TaskV-MS | $80.90 \pm 3.10$ | $88.40 \pm 0.93$ | $80.13 \pm 2.54$ | $88.89 \pm 0.73$ | $61.11 \pm 1.31$ | $78.96 \pm 0.43$ |
| 2-shot | Baseline | $78.80 \pm 3.30$ | $85.07 \pm 1.37$ | $75.67 \pm 2.64$ | $76.80 \pm 1.18$ | $56.49 \pm 2.87$ | $72.92 \pm 0.59$ |
| | TaskV | $84.60 \pm 2.11$ | $88.40 \pm 0.68$ | $84.33 \pm 0.92$ | $90.13 \pm 0.92$ | $62.44 \pm 2.16$ | $80.82 \pm 0.42$ |
| | TaskV-M | $\mathbf{85.70} \pm 1.63$ | $89.27 \pm 1.10$ | $84.13 \pm 1.15$ | $89.64 \pm 0.86$ | $\mathbf{64.49} \pm 2.02$ | $81.48 \pm 0.37$ |
| | TaskV-MS | $84.40 \pm 2.13$ | $\mathbf{89.53} \pm 0.98$ | $\mathbf{84.67} \pm 1.73$ | $\mathbf{90.18} \pm 1.39$ | $\mathbf{64.49} \pm 2.30$ | $\mathbf{81.61} \pm 0.80$ |
| 3-shot | Baseline | $86.20 \pm 2.69$ | $88.07 \pm 1.06$ | $80.00 \pm 1.67$ | $84.04 \pm 1.19$ | $62.18 \pm 1.52$ | $78.51 \pm 0.42$ |
| | TaskV | $90.20 \pm 2.23$ | $88.67 \pm 0.89$ | $\mathbf{86.27} \pm 2.31$ | $92.31 \pm 0.48$ | $66.53 \pm 0.94$ | $83.53 \pm 0.41$ |
| | TaskV-M | $90.30 \pm 1.50$ | $\mathbf{89.87} \pm 0.83$ | $86.07 \pm 2.17$ | $\mathbf{92.36} \pm 0.72$ | $\mathbf{68.13} \pm 0.76$ | $\mathbf{84.15} \pm 0.52$ |
| | TaskV-MS | $\mathbf{90.60} \pm 2.20$ | $89.47 \pm 0.78$ | $86.20 \pm 1.89$ | $91.91 \pm 0.87$ | $67.69 \pm 1.40$ | $83.91 \pm 0.45$ |
| 4-shot | Baseline | $84.80 \pm 2.06$ | $88.07 \pm 0.61$ | $83.27 \pm 1.82$ | $88.89 \pm 1.91$ | $67.16 \pm 1.47$ | $81.52 \pm 0.66$ |
| | TaskV | $88.70 \pm 1.69$ | $89.53 \pm 1.34$ | $86.27 \pm 1.08$ | $\mathbf{92.76} \pm 0.54$ | $70.44 \pm 1.35$ | $84.66 \pm 0.39$ |
| | TaskV-M | $89.60 \pm 1.43$ | $\mathbf{91.00} \pm 1.01$ | $\mathbf{87.20} \pm 0.62$ | $92.36 \pm 1.44$ | $\mathbf{72.53} \pm 0.94$ | $\mathbf{85.64} \pm 0.29$ |
| | TaskV-MS | $\mathbf{90.10} \pm 1.39$ | $90.67 \pm 1.10$ | $87.00 \pm 1.17$ | $92.22 \pm 0.92$ | $72.09 \pm 1.46$ | $85.45 \pm 0.26$ |

task vectors from each arrow token in a single few-shot prompt simultaneously. We name this alternative approach as TaskV-MS. As discussed in Proposition 3, the task vector weights that emerge at each arrow token are approximately orthonormal, suggesting they encode distinct information subsets and can be simultaneously injected to enhance model performance (e.g., by increasing the rank of the induced coefficient matrix $W$). Table 7 shows a comparison between the current multi-vector method (TaskV-M) and this single-prompt variant (TaskV-MS).

While TaskV-MS also delivers strong performance, it slightly underperforms TaskV-M. We believe this is due to the causal attention mechanism in real LLMs, where earlier arrow tokens can only aggregate information from a subset of demonstrations. Nonetheless, TaskV-MS is a promising alternative for accelerating inference.

Table 8: Comparison of the accuracies of $n$-shot ICL and task vector on bijection tasks ($n = 10$). We use gray text to indicate accuracies lower than $60\%$.

| Task | GPT-J | | Pythia-6.9B | | Pythia-12B | | Llama-7B | | Llama-13B | | Qwen3-8B | | Llama3-8B | |
|---|---|---|---|---|---|---|---|---|---|---|---|---|---|---|
| | ICL | TV | ICL | TV | ICL | TV | ICL | TV | ICL | TV | ICL | TV | ICL | TV |
| Lower ↔ Upper | 1.00 | 0.08 | 0.90 | 0.28 | 0.96 | 0.24 | 1.00 | 0.55 | 1.00 | 0.58 | 1.00 | 0.56 | 1.00 | 0.38 |
| English ↔ French | 0.64 | 0.50 | 0.38 | 0.28 | 0.52 | 0.28 | 0.54 | 0.35 | 0.64 | 0.32 | 0.84 | 0.48 | 0.66 | 0.42 |
| English ↔ Italian | 0.68 | 0.56 | 0.62 | 0.48 | 0.60 | 0.56 | 0.70 | 0.47 | 0.72 | 0.44 | 0.68 | 0.36 | 0.70 | 0.36 |
| English ↔ Spanish | 0.70 | 0.52 | 0.62 | 0.56 | 0.66 | 0.56 | 0.64 | 0.43 | 0.84 | 0.56 | 0.70 | 0.32 | 0.72 | 0.32 |
| Present ↔ Gerund | 0.64 | 0.36 | 0.44 | 0.32 | 0.40 | 0.22 | 0.80 | 0.41 | 0.74 | 0.26 | 0.72 | 0.34 | 0.94 | 0.52 |
| Present ↔ Past | 0.60 | 0.38 | 0.48 | 0.36 | 0.54 | 0.16 | 0.52 | 0.33 | 0.68 | 0.44 | 0.78 | 0.42 | 0.90 | 0.58 |
| Present ↔ Perfect | 0.46 | 0.14 | 0.38 | 0.24 | 0.46 | 0.28 | 0.55 | 0.33 | 0.54 | 0.42 | 0.66 | 0.42 | 0.78 | 0.50 |
| Singular ↔ Plural | 0.66 | 0.50 | 0.56 | 0.28 | 0.44 | 0.28 | 0.76 | 0.51 | 0.80 | 0.52 | 0.84 | 0.58 | 0.88 | 0.58 |
| Antonym | 0.86 | 0.78 | 0.76 | 0.66 | 0.76 | 0.70 | 0.83 | 0.73 | 0.78 | 0.72 | 0.82 | 0.74 | 0.82 | 0.76 |

Table 9: Comparison of the accuracies of $n$-shot ICL and task vector on bijection tasks ($n = 20$). We use gray text to indicate accuracies lower than $60\%$.

| Task | GPT-J | | Pythia-6.9B | | Pythia-12B | | Llama-7B | | Llama-13B | | Qwen3-8B | | Llama3-8B | |
|---|---|---|---|---|---|---|---|---|---|---|---|---|---|---|
| | ICL | TV | ICL | TV | ICL | TV | ICL | TV | ICL | TV | ICL | TV | ICL | TV |
| Lower ↔ Upper | 1.00 | 0.12 | 1.00 | 0.32 | 0.94 | 0.38 | 1.00 | 0.48 | 1.00 | 0.60 | 1.00 | 0.58 | 1.00 | 0.36 |
| English ↔ French | 0.74 | 0.54 | 0.44 | 0.40 | 0.52 | 0.40 | 0.52 | 0.34 | 0.58 | 0.34 | 0.58 | 0.30 | 0.74 | 0.28 |
| English ↔ Italian | 0.62 | 0.54 | 0.66 | 0.46 | 0.68 | 0.48 | 0.78 | 0.50 | 0.74 | 0.48 | 0.76 | 0.38 | 0.76 | 0.32 |
| English ↔ Spanish | 0.80 | 0.58 | 0.54 | 0.38 | 0.56 | 0.40 | 0.78 | 0.58 | 0.84 | 0.58 | 0.66 | 0.32 | 0.86 | 0.40 |
| Present ↔ Gerund | 0.54 | 0.26 | 0.54 | 0.22 | 0.46 | 0.14 | 0.84 | 0.44 | 0.94 | 0.38 | 0.88 | 0.28 | 0.98 | 0.52 |
| Present ↔ Past | 0.66 | 0.26 | 0.54 | 0.30 | 0.58 | 0.28 | 0.72 | 0.30 | 0.76 | 0.44 | 0.74 | 0.40 | 1.00 | 0.48 |
| Present ↔ Perfect | 0.42 | 0.18 | 0.44 | 0.20 | 0.46 | 0.24 | 0.48 | 0.30 | 0.52 | 0.48 | 0.80 | 0.44 | 0.90 | 0.48 |
| Singular ↔ Plural | 0.64 | 0.40 | 0.62 | 0.36 | 0.52 | 0.28 | 0.80 | 0.52 | 0.94 | 0.42 | 0.86 | 0.60 | 0.92 | 0.60 |
| Antonym | 0.84 | 0.76 | 0.84 | 0.70 | 0.90 | 0.82 | 0.90 | 0.84 | 0.90 | 0.84 | 0.84 | 0.74 | 0.84 | 0.76 |

### B.4 FURTHER RESULTS ON BIJECTION TASKS

Here, we extend the results from Table 1 that illustrate the failure of task vectors on bijection tasks across a broader range of LLMs and varying numbers of input demonstrations. We keep the same experimental settings as Table 1 while increasing the number of demonstrations to $n \in \{10, 20\}$, and report the results for 7 distinct LLMs: GPT-J, Pythia-6.9B, Pythia-12B, Llama-7B, Llama-13B, Qwen3-8B and Llama3-8B. As shown in Tables 8 and 9, the task vector method results in a significant performance drop compared to the standard ICL on bijection tasks. These results further support our claims that:

- Task vectors systematically fail on bijection tasks, even when further increasing the number of demonstrations in the prompt.

- The failure is consistent across multiple model architectures, validating that the issue stems from a fundamental expressiveness limitation rather than model-specific artifacts.

### B.5 FULL SALIENCY ANALYSIS RESULTS

In the main text, we reported a simplified version of the saliency map due to space limitations, focusing only on the demonstration tokens $x_i, \rightarrow, y_i$. In Figure 5, we report the full saliency map covering every token in the prompt. Here, "B" stands for the [BOS] token, and "E" stands for the word "Example". Please refer to eq. (13) for further details about the structure of the input prompt. As can be seen, the highlighted saliency weights exhibit clear patterns of embedding concatenation and weighted summation. It can also be observed that latter demonstrations weigh more for task vector formation (i.e., saliency magnitudes for latter $y_i$ tokens are larger in Figure 5b).

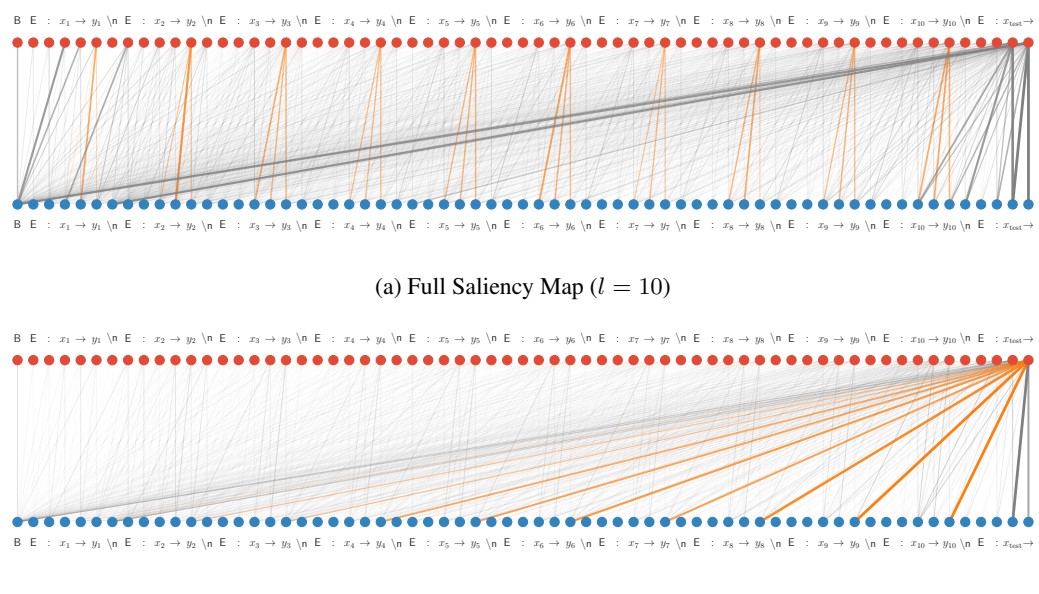

(a) Full Saliency Map ($l = 10$)

(b) Full Saliency Map ($l = 12$)

Figure 5: Visualization of full saliency matrices as bipartite graphs between layer $l$ (●) and $l + 1$ (●), edge widths indicate saliency magnitude (Llama-7B, $n = 10$). (a) Each $y_i$ token attends to its corresponding $(x_i, y_i)$ pair, reflecting embedding concatenation. (b) The final ($\rightarrow$) token attends broadly to all $y_i$ tokens, indicating task vector formation.

## C  AUXILIARY LEMMAS

**Lemma 8** (Proposed in (Ahn et al., 2023)). *Given positive objective function $f(A)$ taking parameters $A = \{A_i\}_{i=1}^n$, where $A_i \in \mathbb{R}^{d_i \times d_i}$. Let $\mathcal{S} = \Pi_{i=1}^n \mathcal{S}_i \subset \Pi_{i=1}^n \mathbb{R}^{d_i \times d_i}$ be a predefined parameter subspace. Define $\widetilde{A}(t, R_i) = \{A_1, \cdots, A_i + tR_i, \cdots, A_n\}$ given $i \in [1, n]$, $R_i \in \mathbb{R}^{d_i \times d_i}$ and $t \in \mathbb{R}$. If for any $A \in \mathcal{S}$ and $R_i \in \mathbb{R}^{d_i \times d_i}$, there exists $\widetilde{R}_i \in \mathcal{S}_i$ such that*

$$\frac{\mathrm{d}}{\mathrm{d}t} f\left(\widetilde{A}(t, \widetilde{R}_i)\right)\bigg|_{t=0} \leq \frac{\mathrm{d}}{\mathrm{d}t} f\left(\widetilde{A}(t, R_i)\right)\bigg|_{t=0},$$

*then we have*

$$\inf_{A \in \mathcal{S}} \sum_{i=1}^n \|\nabla_{A_i} f(A)\|_F^2 = 0.$$

*Proof.* This lemma is proved as part of the main theorems in (Ahn et al., 2023). We rearrange the proof here to accommodate arbitrary function of matrices. Firstly, notice that for any $R = \{R_i\}_{i=1}^n \in \Pi_{i=1}^n \mathbb{R}^{d_i \times d_i}$,

$$\sum_{i=1}^n \frac{\mathrm{d}}{\mathrm{d}t} f\left(\widetilde{A}(t, R_i)\right)\bigg|_{t=0} = \frac{\mathrm{d}}{\mathrm{d}t} f(A + tR)\bigg|_{t=0}.$$

Therefore, the provided precondition is equivalent to stating that for any $A \in \mathcal{S}$ and $R \in \Pi_{i=1}^n \mathbb{R}^{d_i \times d_i}$, there exists $\widetilde{R} \in \mathcal{S}$ such that:

$$\frac{\mathrm{d}}{\mathrm{d}t} f\left(A + t\widetilde{R}\right)\bigg|_{t=0} \leq \frac{\mathrm{d}}{\mathrm{d}t} f(A + tR)\bigg|_{t=0}.$$

Let $R = -\nabla_A f(A)$, we then have

$$\frac{\mathrm{d}}{\mathrm{d}t} f(A + tR)\bigg|_{t=0} = \left\langle \frac{\mathrm{d}f(A - t\nabla_A f(A))}{\mathrm{d}(A - t\nabla_A f(A))}, \frac{\mathrm{d}(A - t\nabla_A f(A))}{t} \right\rangle\bigg|_{t=0}$$

$$= \langle \nabla_A f(A), -\nabla_A f(A) \rangle = -\|\nabla_A f(A)\|_F^2.$$

If the infimum of $\|\nabla_A f(A)\|_F^2$ is not zero but some positive value $p$, then the $\mathcal{S}$-constrained gradient flow induced by $\widetilde{R}$ will lead to unbounded descent:

$$\frac{\mathrm{d}}{\mathrm{d}t} f\Big(A + t\widetilde{R}\Big)\bigg|_{t=0} \leq -p.$$

This contradicts the fact that $f(A) \geq 0$ and concludes the proof. $\qquad\square$

The following lemma is an extension of Lemma 5 in (Ahn et al., 2023) by accommodating multivariate $y$ samples as well as enabling a wider range of demonstration and transformer parameter configurations.

**Lemma 9.** *Let $x_1, \cdots, x_{n+1}$ be i.i.d. samples from an input distribution, and let $W$ be sampled independently of $\{x_i\}_{i=1}^{n+1}$. Let $Z_0 \in \mathbb{R}^{(2d) \times N}$, where $N \in \mathbb{Z}$, be constructed of form*

$$Z_0 = \begin{bmatrix} * & \cdots & * & * \\ * & \cdots & * & 0_d \end{bmatrix} \in \mathbb{R}^{(2d) \times N},$$

*where the $*$ parts can be arbitrarily constructed from $\{x_i\}_{i=1}^{n+1}$ and $W$. Let $\widetilde{Z}_0$ be defined as replacing the zero part of $Z_0$ by $y_{n+1}$:*

$$\widetilde{Z}_0 = \begin{bmatrix} * & \cdots & * & * \\ * & \cdots & * & y_{n+1} \end{bmatrix} \in \mathbb{R}^{(2d) \times N}.$$

*Let $\widetilde{Z}_l$ be the output of the $l$-th layer of the linear transformer, and let $\widetilde{X}_l, \widetilde{Y}_l \in \mathbb{R}^{d \times N}$ be the first and last $d$ rows of $\widetilde{Z}_l$, respectively. Suppose that the $\{Q_l\}_{l=1}^L$ matrices are of form*

$$Q_l = \begin{bmatrix} \underbrace{*}_{d\ columns} & 0_{(2d+d_p) \times d} & \underbrace{*}_{d_p\ columns} \end{bmatrix},$$

*Then the in-context risk of this $L$-layer linear transformer is equivalent to*

$$\mathcal{L}\big(\{V_l, Q_l\}_{l=1}^L\big) = \mathbb{E}_{\widetilde{Z}_0, W}\Big[\mathrm{tr}\Big((I_N - M)\widetilde{Y}_L^\top \widetilde{Y}_L (I_N - M)\Big)\Big]. \tag{14}$$

*Proof.* Let the $V_l$ and $Q_l$ matrices be represented as:

$$V_l = \begin{bmatrix} V_l^1 \\ V_l^2 \end{bmatrix}, \quad Q_l = \begin{bmatrix} Q_l^1 & 0 & Q_l^2 \end{bmatrix},$$

where $V_l^1, V_l^2 \in \mathbb{R}^{d \times 2d}, Q_l^1 \in \mathbb{R}^{(2d+d_p) \times d}, Q_l^2 \in \mathbb{R}^{(2d+d_p) \times d_p}$. Then the update rule in eq. (5) can be rephrased as

$$X_l = X_{l-1} + \frac{1}{n} V_l^1 Z_{l-1} M \big[Z_{l-1}^\top, P\big]\big(Q_l^1 X_{l-1} + Q_l^2 P\big),$$

$$Y_l = Y_{l-1} + \frac{1}{n} V_l^2 Z_{l-1} M \big[Z_{l-1}^\top, P\big]\big(Q_l^1 X_{l-1} + Q_l^2 P\big).$$

Let $\Delta_Z = \widetilde{Z}_0 - Z_0$, i.e. an all-zero matrix except that the last half of the last column is $y_{n+1}$. Let $\Delta_X$ and $\Delta_Y$ be its first and last $d$ rows respectively, then $\Delta_X = 0$ and $\Delta_Y = \begin{bmatrix} 0 & \cdots & 0 & y_{n+1} \end{bmatrix}$. Note that $\widetilde{Z}_l = Z_l + \Delta_Z$ holds for $l = 0$ trivially. Now suppose it holds for some $l = k - 1$, then

$$\widetilde{X}_k = \widetilde{X}_{k-1} + \frac{1}{n} V_k^1 \widetilde{Z}_{k-1} M\big[\widetilde{Z}_{k-1}^\top, P\big]\Big(Q_k^1 \widetilde{X}_{k-1} + Q_k^2 P\Big)$$

$$= X_{k-1} + \frac{1}{n} V_k^1 Z_{k-1} M\big[Z_{k-1}^\top, P\big]\big(Q_k^1 X_{k-1} + Q_k^2 P\big)$$

$$+ \frac{1}{n} V_k^1 \Delta_Z M\big[Z_{k-1}^\top, P\big]\big(Q_k^1 X_{k-1} + Q_k^2 P\big)$$

$$+ \frac{1}{n} V_k^1 Z_{k-1} M\big[\Delta_Z^\top, 0_{d_p \times d_p}\big]\big(Q_k^1 X_{k-1} + Q_k^2 P\big)$$

$$+ \frac{1}{n} V_k^1 \Delta_Z M \left[ \Delta_Z^\top, 0_{d_p \times d_p} \right] \left( Q_k^1 X_{k-1} + Q_k^2 P \right)$$

$$= X_{k-1} + \frac{1}{n} V_k^1 Z_{k-1} M \left[ Z_{k-1}^\top, P \right] \left( Q_k^1 X_{k-1} + Q_k^2 P \right) = X_k,$$

where the last step holds by noticing that $\Delta_Z M = 0$. Similarly, one can prove that

$$\widetilde{Y}_k = Y_{k-1} + \Delta_Y + \frac{1}{n} V_k^2 Z_{k-1} M \left[ Z_{k-1}^\top, P \right] \left( Q_k^1 X_{k-1} + Q_k^2 P \right) = Y_k + \Delta_Y.$$

Therefore, it holds that for any $l \in [1, L]$, $\widetilde{Z}_l = Z_l + \Delta_Z$. Recall the in-context risk in eq. (2):

$$\mathcal{L}\big(\{V_l, Q_l\}_{l=1}^L\big) = \mathbb{E}_{Z_0, W} \left\| (Z_L)_{(d+1:2d), N} + y_{n+1} \right\|_2^2$$

$$= \mathbb{E}_{Z_0, W} \left\| (Y_L + \Delta_Y)(I_N - M) \right\|_2^2$$

$$= \mathbb{E}_{\widetilde{Z}_0, W} \left[ \text{tr}\Big( (I_N - M) \widetilde{Y}_L^\top \widetilde{Y}_L (I_N - M) \Big) \right].$$

The proof is complete. □

## D  PROOF OF THEORETICAL RESULTS

### D.1  PROOF OF PROPOSITION 4

*Proof.* We will first prove sufficiency. Let $W = ab^\top$ be a rank-one matrix, where $a, b \in \mathbb{R}^d$. The given conditions imply that $x = Wy = WWx = ab^\top ab^\top x$, we then have $b^\top x = b^\top ab^\top ab^\top x = (b^\top a)^2 b^\top x$. Since $b^\top x \neq 0$, we can conclude that $b^\top a = \pm 1$. Then, $x = ab^\top ab^\top x = \pm ab^\top x = \pm y$.

To prove the necessity, it suffices to show that selecting $W = xx^\top / \|x\|_2^2$ when $x = y$ satisfies the given conditions (alternatively, select $W = -xx^\top / \|x\|_2^2$ when $x = -y$). □

### D.2  PROOF OF THEOREM 1

*Proof.* To enhance the readability of the notations in this proof, we will drop the constant $\frac{1}{n}$ factor in linear attention. Furthermore, we will simplify $\widetilde{Z}_0$, $\widetilde{X}_0$ and $\widetilde{Y}_0$ in Lemma 9 as $Z_0$, $X_0$ and $Y_0$ respectively. This results in different definitions compared to the original ones, but we will not refer to the original definitions in the remainder of this proof.

$$Z_0 = \begin{bmatrix} X_0 \\ Y_0 \end{bmatrix} = \begin{bmatrix} x_1 & 0 & \cdots & x_n & 0 & x_{\text{test}} & 0 \\ 0 & y_1 & \cdots & 0 & y_n & 0 & y_{\text{test}} \end{bmatrix} \in \mathbb{R}^{(2d) \times (2n+2)}.$$

Let $Z_l$ be the output of the $l$-th layer of the transformer, and let $X_l, Y_l \in \mathbb{R}^{d \times (2n+2)}$ denote the first and last $d$ rows of $Z_l$, respectively. Under the constraint in eq. (6), we can verify that

$$\begin{aligned} X_l &= X_{l-1} + A_l X_{l-1} M(X_{l-1}^\top C_l X_{l-1} + D_l), \\ Y_l &= Y_{l-1} + B_l Y_{l-1} M(X_{l-1}^\top C_l X_{l-1} + D_l). \end{aligned} \tag{15}$$

In the following analysis, we will use $f(A \leftarrow B)$ to denote the result of the function $f$ of $A$ when replacing the value of $A$ with $B$. Additionally, we denote $f(A \leftarrow B * A)$ as $f(A \overset{*}{\leftarrow} B)$ for any operator $*$. Therefore, $f(A \overset{+}{\leftarrow} B) = f(A \leftarrow A + B)$. We also denote $f(A \overset{\times}{\leftarrow} B) = f(A \leftarrow BA)$ and $f(A \overset{\diamond}{\leftarrow} B) = f(A \leftarrow AB)$ for convenience.

Our goal is proving that, for any $E \in A \cup B \cup C \cup D$ and an arbitrary matrix $R \in \mathbb{R}^{d \times d}$ ($\mathbb{R}^{d_p \times d_p}$ for $D$), there exists $\widetilde{R} \in \mathcal{S}_I$ ($\mathcal{S}_\Sigma$ for $C$, $\mathcal{S}_P$ for $D$) such that

$$\left. \frac{\mathrm{d}}{\mathrm{d}t} \mathcal{L}(E \overset{+}{\leftarrow} t\widetilde{R}) \right|_{t=0} \leq \left. \frac{\mathrm{d}}{\mathrm{d}t} \mathcal{L}(E \overset{+}{\leftarrow} tR) \right|_{t=0}. \tag{16}$$

Let $\overline{X}_0 = [0, x_1, \cdots, 0, x_{\text{test}}]$ be a function of $X_0$, we then have $Y_0 = W \overline{X}_0$. Let $U_\perp \in \mathbb{R}^{d \times d}$ be a uniformly sampled random orthonormal matrix, and let $U_\Sigma = \Sigma^{1/2} U_\perp \Sigma^{-1/2}$. One can verify that

$U_{\Sigma}^{-1} = \Sigma^{1/2} U_{\perp}^{\top} \Sigma^{-1/2}$. By applying Lemma 9 and the fact that $X_0 \overset{d}{=} U_{\Sigma} X_0$, we have that for any given matrix $R$,

$$\frac{\mathrm{d}}{\mathrm{d}t} \mathcal{L}(E \overset{\pm}{\leftarrow} tR) \Big|_{t=0}$$

$$= \frac{\mathrm{d}}{\mathrm{d}t} \mathbb{E}_{X_0, W} \left[ \mathrm{tr}\left( (I-M) Y_L^{\top}(E \overset{\pm}{\leftarrow} tR) Y_L(E \overset{\pm}{\leftarrow} tR)(I-M) \right) \right] \Big|_{t=0}$$

$$= 2 \mathbb{E}_{X_0, W} \left[ \mathrm{tr}\left( (I-M) Y_L^{\top} \frac{\mathrm{d}}{\mathrm{d}t} Y_L(E \overset{\pm}{\leftarrow} tR) \Big|_{t=0} (I-M) \right) \right]$$

$$= 2 \mathbb{E}_{X_0, W, U_{\perp}} \left[ \mathrm{tr}\left( (I-M) Y_L^{\top}(X_0 \overset{\times}{\leftarrow} U_{\Sigma}) \frac{\mathrm{d}}{\mathrm{d}t} Y_L(X_0 \overset{\times}{\leftarrow} U_{\Sigma}, E \overset{\pm}{\leftarrow} tR) \Big|_{t=0} (I-M) \right) \right].$$

Next, we will show that eq. (16) holds for each one of $A_i, B_i, C_i, D_i$ for any $i \in [1, L]$.

**1. Equation (16) holds for $A_i$.**

We first show that for any $l \in [1, L]$, the following equations hold:

$$X_l(X_0 \overset{\times}{\leftarrow} U_{\Sigma}) = U_{\Sigma} X_l, \tag{17}$$

$$\frac{\mathrm{d}}{\mathrm{d}t} X_l(X_0 \overset{\times}{\leftarrow} U_{\Sigma}, A_i \overset{\pm}{\leftarrow} tR) \Big|_{t=0} = U_{\Sigma} \frac{\mathrm{d}}{\mathrm{d}t} X_l(A_i \overset{\pm}{\leftarrow} t U_{\Sigma}^{-1} R U_{\Sigma}) \Big|_{t=0}. \tag{18}$$

It is straightforward to verify that eq. (17) holds for $l = 0$. Now suppose that eq. (17) holds for some $l = k - 1$, we then have

$$X_k(X_0 \overset{\times}{\leftarrow} U_{\Sigma})$$

$$= X_{k-1}(X_0 \overset{\times}{\leftarrow} U_{\Sigma}) + A_l X_{k-1}(X_0 \overset{\times}{\leftarrow} U_{\Sigma}) M \left( X_{k-1}^{\top}(X_0 \overset{\times}{\leftarrow} U_{\Sigma}) C_l X_{k-1}(X_0 \overset{\times}{\leftarrow} U_{\Sigma}) + D_l \right)$$

$$= U_{\Sigma} X_{k-1} + A_l U_{\Sigma} X_{k-1} M \left( X_{k-1}^{\top} U_{\Sigma}^{\top} C_l U_{\Sigma} X_{k-1} + D_l \right)$$

$$= U_{\Sigma} \left( X_{k-1} + A_l X_{k-1} M \left( X_{k-1}^{\top} C_l X_{k-1} + D_l \right) \right) = U_{\Sigma} X_k,$$

where the third equality follows by noticing that when $A_l = a_l I_d$ and $C_l = c_l \Sigma^{-1}$, we have $A_l U_{\Sigma} = U_{\Sigma} A_l$ and $U_{\Sigma}^{\top} C_l U_{\Sigma} = C_l$. This concludes the proof of eq. (17).

We now turn to the proof of eq. (18). Notice that when $l < i$, we naturally have

$$\frac{\mathrm{d}}{\mathrm{d}t} X_l(X_0 \overset{\times}{\leftarrow} U_{\Sigma}, A_i \overset{\pm}{\leftarrow} tR) \Big|_{t=0} = U_{\Sigma} \frac{\mathrm{d}}{\mathrm{d}t} X_l(A_i \overset{\pm}{\leftarrow} t U_{\Sigma}^{-1} R U_{\Sigma}) \Big|_{t=0} = 0.$$

When $l = i$, it is easy to verify that

$$\frac{\mathrm{d}}{\mathrm{d}t} X_l(X_0 \overset{\times}{\leftarrow} U_{\Sigma}, A_i \overset{\pm}{\leftarrow} tR) \Big|_{t=0} = R U_{\Sigma} X_{l-1} M (X_{l-1}^{\top} U_{\Sigma}^{\top} C_l U_{\Sigma} X_{l-1} + D_l)$$

$$= U_{\Sigma} \cdot U_{\Sigma}^{-1} R U_{\Sigma} M (X_{l-1}^{\top} C_l X_{l-1} + D_l)$$

$$= U_{\Sigma} \frac{\mathrm{d}}{\mathrm{d}t} X_l(A_i \overset{\pm}{\leftarrow} t U_{\Sigma}^{-1} R U_{\Sigma}) \Big|_{t=0}.$$

Now suppose that eq. (18) holds for some $l = k - 1 \geq i$, one can verify that:

$$\frac{\mathrm{d}}{\mathrm{d}t} X_k(X_0 \overset{\times}{\leftarrow} U_{\Sigma}, A_i \overset{\pm}{\leftarrow} tR) \Big|_{t=0}$$

$$= \frac{\mathrm{d}}{\mathrm{d}t} X_{k-1}(X_0 \overset{\times}{\leftarrow} U_{\Sigma}, A_i \overset{\pm}{\leftarrow} tR) \Big|_{t=0} + \frac{\mathrm{d}}{\mathrm{d}t} A_k X_{k-1}(X_0 \overset{\times}{\leftarrow} U_{\Sigma}, A_i \overset{\pm}{\leftarrow} tR) M$$

$$\cdot \left( X_{k-1}^{\top}(X_0 \overset{\times}{\leftarrow} U_{\Sigma}, A_i \overset{\pm}{\leftarrow} tR) C_k X_{k-1}(X_0 \overset{\times}{\leftarrow} U_{\Sigma}, A_i \overset{\pm}{\leftarrow} tR) + D_k \right) \Big|_{t=0}$$

$$= \frac{\mathrm{d}}{\mathrm{d}t} X_{k-1}(X_0 \overset{\times}{\leftarrow} U_{\Sigma}, A_i \overset{\pm}{\leftarrow} tR) \Big|_{t=0}$$

$$+ A_k \frac{\mathrm{d}}{\mathrm{d}t} X_{k-1}(X_0 \overset{\times}{\leftarrow} U_\Sigma, A_i \overset{+}{\leftarrow} tR)\Big|_{t=0} M\left(X_{k-1}^\top(X_0 \overset{\times}{\leftarrow} U_\Sigma)C_k X_{k-1}(X_0 \overset{\times}{\leftarrow} U_\Sigma) + D_k\right)$$

$$+ A_k X_{k-1}(X_0 \overset{\times}{\leftarrow} U_\Sigma)M \frac{\mathrm{d}}{\mathrm{d}t} X_{k-1}^\top(X_0 \overset{\times}{\leftarrow} U_\Sigma, A_i \overset{+}{\leftarrow} tR)\Big|_{t=0} C_k X_{k-1}(X_0 \overset{\times}{\leftarrow} U_\Sigma)$$

$$+ A_k X_{k-1}(X_0 \overset{\times}{\leftarrow} U_\Sigma)M X_{k-1}^\top(X_0 \overset{\times}{\leftarrow} U_\Sigma)C_k \frac{\mathrm{d}}{\mathrm{d}t} X_{k-1}(X_0 \overset{\times}{\leftarrow} U_\Sigma, A_i \overset{+}{\leftarrow} tR)\Big|_{t=0}$$

$$= U_\Sigma \frac{\mathrm{d}}{\mathrm{d}t} X_{k-1}(A_i \overset{+}{\leftarrow} tU_\Sigma^{-1}RU_\Sigma)\Big|_{t=0}$$

$$+ U_\Sigma A_k \frac{\mathrm{d}}{\mathrm{d}t} X_{k-1}(A_i \overset{+}{\leftarrow} tU_\Sigma^{-1}RU_\Sigma)\Big|_{t=0} M\left(X_{k-1}^\top C_k X_{k-1} + D_k\right)$$

$$+ U_\Sigma A_k X_{k-1} M \frac{\mathrm{d}}{\mathrm{d}t} X_{k-1}^\top(A_i \overset{+}{\leftarrow} tU_\Sigma^{-1}RU_\Sigma)\Big|_{t=0} C_k X_{k-1}$$

$$+ U_\Sigma A_k X_{k-1} M X_{k-1}^\top C_k \frac{\mathrm{d}}{\mathrm{d}t} X_{k-1}(A_i \overset{+}{\leftarrow} tU_\Sigma^{-1}RU_\Sigma)\Big|_{t=0}$$

$$= U_\Sigma \frac{\mathrm{d}}{\mathrm{d}t} X_{k-1}(A_i \overset{+}{\leftarrow} tU_\Sigma^{-1}RU_\Sigma)\Big|_{t=0} + U_\Sigma \frac{\mathrm{d}}{\mathrm{d}t} A_k X_{k-1}(A_i \overset{+}{\leftarrow} tU_\Sigma^{-1}RU_\Sigma)M$$

$$\cdot \left(X_{k-1}^\top(A_i \overset{+}{\leftarrow} tU_\Sigma^{-1}RU_\Sigma)C_k X_{k-1}(A_i \overset{+}{\leftarrow} tU_\Sigma^{-1}RU_\Sigma) + D_k\right)\Big|_{t=0}$$

$$= U_\Sigma \frac{\mathrm{d}}{\mathrm{d}t} X_k(A_i \overset{+}{\leftarrow} tU_\Sigma^{-1}RU_\Sigma)\Big|_{t=0}.$$

This completes the proof of eq. (18).

Under the condition that $B_l = b_l I_d$ for some $b_l \in \mathbb{R}$, we can simplify eq. (15) as

$$Y_l = Y_{l-1} + b_l Y_{l-1} M(X_{l-1}^\top C_l X_{l-1} + D_l)$$
$$= Y_{l-1}\left(I + b_l M(X_{l-1}^\top C_l X_{l-1} + D_l)\right)$$
$$= Y_0 \prod_{j=1}^{l}\left(I + b_j M(X_{j-1}^\top C_j X_{j-1} + D_j)\right).$$

Define $G_l = \overline{X}_0 \prod_{j=1}^{l}\left(I + b_j M(X_{j-1}^\top C_j X_{j-1} + D_j)\right)$, then it satisfies that $Y_l = WG_l$. We are ready to prove that similar results to eqs. (17) and (18) also hold for $G_l, l \in [1, L]$:

$$G_l(X_0 \overset{\times}{\leftarrow} U_\Sigma) = U_\Sigma G_l, \tag{19}$$

$$\frac{\mathrm{d}}{\mathrm{d}t} G_l(X_0 \overset{\times}{\leftarrow} U_\Sigma, A_i \overset{+}{\leftarrow} tR)\Big|_{t=0} = U_\Sigma \frac{\mathrm{d}}{\mathrm{d}t} G_l(A_i \overset{+}{\leftarrow} tU_\Sigma^{-1}RU_\Sigma)\Big|_{t=0}. \tag{20}$$

Notice that eq. (19) holds trivially for $l = 0$ as $G_0 = \overline{X}_0$. Now suppose that eq. (19) holds for some $l = k - 1$, we then have

$$G_k(X_0 \overset{\times}{\leftarrow} U_\Sigma) = G_{k-1}(X_0 \overset{\times}{\leftarrow} U_\Sigma)\left(I + b_k M(X_{k-1}^\top(X_0 \overset{\times}{\leftarrow} U_\Sigma)C_k X_{k-1}(X_0 \overset{\times}{\leftarrow} U_\Sigma) + D_k)\right)$$
$$= U_\Sigma G_{k-1}\left(I + b_k M(X_{k-1}^\top C_k X_{k-1} + D_k)\right) = U_\Sigma G_k.$$

This concludes eq. (19). As for eq. (20), notice that both sides equal 0 when $l \leq i$. Now suppose that eq. (20) holds for some $l = k - 1 \geq i$, we then have:

$$\frac{\mathrm{d}}{\mathrm{d}t} G_k(X_0 \overset{\times}{\leftarrow} U_\Sigma, A_i \overset{+}{\leftarrow} tR)\Big|_{t=0}$$

$$= \frac{\mathrm{d}}{\mathrm{d}t} G_{k-1}(X_0 \overset{\times}{\leftarrow} U_\Sigma, A_i \overset{+}{\leftarrow} tR)\Big|_{t=0} + \frac{\mathrm{d}}{\mathrm{d}t} b_k G_{k-1}(X_0 \overset{\times}{\leftarrow} U_\Sigma, A_i \overset{+}{\leftarrow} tR)M$$

$$\cdot \left(X_{k-1}^\top(X_0 \overset{\times}{\leftarrow} U_\Sigma, A_i \overset{+}{\leftarrow} tR)C_k X_{k-1}(X_0 \overset{\times}{\leftarrow} U_\Sigma, A_i \overset{+}{\leftarrow} tR) + D_k\right)\Big|_{t=0}$$

$$= \frac{\mathrm{d}}{\mathrm{d}t} G_{k-1}(X_0 \overset{\times}{\leftarrow} U_\Sigma, A_i \overset{+}{\leftarrow} tR)\Big|_{t=0}$$

$$+ b_k \frac{\mathrm{d}}{\mathrm{d}t} G_{k-1}(X_0 \overset{\times}{\leftarrow} U_\Sigma, A_i \overset{+}{\leftarrow} tR)\Big|_{t=0} M\Big(X_{k-1}^\top(X_0 \overset{\times}{\leftarrow} U_\Sigma)C_k X_{k-1}(X_0 \overset{\times}{\leftarrow} U_\Sigma) + D_k\Big)$$

$$+ b_k G_{k-1}(X_0 \overset{\times}{\leftarrow} U_\Sigma)M \frac{\mathrm{d}}{\mathrm{d}t} X_{k-1}^\top(X_0 \overset{\times}{\leftarrow} U_\Sigma, A_i \overset{+}{\leftarrow} tR)\Big|_{t=0} C_k X_{k-1}(X_0 \overset{\times}{\leftarrow} U_\Sigma)$$

$$+ b_k G_{k-1}(X_0 \overset{\times}{\leftarrow} U_\Sigma)M X_{k-1}^\top(X_0 \overset{\times}{\leftarrow} U_\Sigma)C_k \frac{\mathrm{d}}{\mathrm{d}t} X_{k-1}(X_0 \overset{\times}{\leftarrow} U_\Sigma, A_i \overset{+}{\leftarrow} tR)\Big|_{t=0}$$

$$= U_\Sigma \frac{\mathrm{d}}{\mathrm{d}t} G_{k-1}(A_i \overset{+}{\leftarrow} tU_\Sigma^{-1}RU_\Sigma)\Big|_{t=0}$$

$$+ b_k U_\Sigma \frac{\mathrm{d}}{\mathrm{d}t} G_{k-1}(A_i \overset{+}{\leftarrow} tU_\Sigma^{-1}RU_\Sigma)\Big|_{t=0} M\big(X_{k-1}^\top C_k X_{k-1} + D_k\big)$$

$$+ b_k U_\Sigma G_{k-1} M \frac{\mathrm{d}}{\mathrm{d}t} X_{k-1}^\top(A_i \overset{+}{\leftarrow} tU_\Sigma^{-1}RU_\Sigma)\Big|_{t=0} C_k X_{k-1}$$

$$+ b_k U_\Sigma G_{k-1} M X_{k-1}^\top C_k \frac{\mathrm{d}}{\mathrm{d}t} X_{k-1}(A_i \overset{+}{\leftarrow} tU_\Sigma^{-1}RU_\Sigma)\Big|_{t=0}$$

$$= U_\Sigma \frac{\mathrm{d}}{\mathrm{d}t} G_k(A_i \overset{+}{\leftarrow} tU_\Sigma^{-1}RU_\Sigma)\Big|_{t=0}.$$

This concludes the proof of eq. (20). Consider the in-context risk:

$$\frac{\mathrm{d}}{\mathrm{d}t} \mathcal{L}(A_i \overset{+}{\leftarrow} tR)\Big|_{t=0}$$

$$= 2\mathbb{E}_{X_0, W, U_\perp}\left[\mathrm{tr}\left((I - M)Y_L^\top(X_0 \overset{\times}{\leftarrow} U_\Sigma) \frac{\mathrm{d}}{\mathrm{d}t} Y_L(X_0 \overset{\times}{\leftarrow} U_\Sigma, A_i \overset{+}{\leftarrow} tR)\Big|_{t=0} (I - M)\right)\right]$$

$$= 2\mathbb{E}_{X_0, W, U_\perp}\left[\mathrm{tr}\left((I - M)G_L^\top U_\Sigma^\top W^\top W U_\Sigma \frac{\mathrm{d}}{\mathrm{d}t} G_L(A_i \overset{+}{\leftarrow} tU_\Sigma^{-1}RU_\Sigma)\Big|_{t=0} (I - M)\right)\right]$$

$$= 2d\,\mathbb{E}_{X_0}\left[\mathrm{tr}\left((I - M)G_L^\top \Sigma^{-1} \frac{\mathrm{d}}{\mathrm{d}t} \mathbb{E}_{U_\perp}\left[G_L(A_i \overset{+}{\leftarrow} tU_\Sigma^{-1}RU_\Sigma)\right]\Big|_{t=0} (I - M)\right)\right]$$

$$= 2d\,\mathbb{E}_{X_0}\left[\mathrm{tr}\left((I - M)G_L^\top \Sigma^{-1} \frac{\mathrm{d}}{\mathrm{d}t} G_L(A_i \overset{+}{\leftarrow} \mathbb{E}_{U_\perp}[tU_\Sigma^{-1}RU_\Sigma])\Big|_{t=0} (I - M)\right)\right]$$

$$= 2d\,\mathbb{E}_{X_0}\left[\mathrm{tr}\left((I - M)G_L^\top \Sigma^{-1} \frac{\mathrm{d}}{\mathrm{d}t} G_L(A_i \overset{+}{\leftarrow} trI_d)\Big|_{t=0} (I - M)\right)\right]$$

$$= \frac{\mathrm{d}}{\mathrm{d}t} \mathbb{E}_{X_0, W}\left[\mathrm{tr}\left((I - M)Y_L^\top(A_i \overset{+}{\leftarrow} trI_d)Y_L(A_i \overset{+}{\leftarrow} trI_d)(I - M)\right)\right]\Big|_{t=0}$$

$$= \frac{\mathrm{d}}{\mathrm{d}t} \mathcal{L}(A_i \overset{+}{\leftarrow} trI_d)\Big|_{t=0},$$

where $r = \mathbb{E}_{U_\perp}[U_\Sigma^{-1}RU_\Sigma] = \frac{1}{d}\mathrm{tr}(\Sigma^{-1/2}R\Sigma^{1/2})$, and we used the fact that $U_\Sigma^\top \Sigma^{-1} U_\Sigma = \Sigma^{-1}$, and $\frac{\mathrm{d}}{\mathrm{d}t} G_L(A_i \overset{+}{\leftarrow} tR)\Big|_{t=0}$ is affine in $R$. This concludes that eq. (16) holds for $A_i$, $i \in [1, L]$.

### 2. Equation (16) holds for $B_i$.

From the recursive expressions in eq. (15), we can conclude that the values of $X_l$ do not depend on $B_i$. Therefore, we naturally have

$$X_l(B_i \overset{+}{\leftarrow} tR) = X_l. \tag{21}$$

Next, we would like to show that for any $l \in [1, L]$,

$$\mathbb{E}_W\left[W^\top \frac{\mathrm{d}}{\mathrm{d}t} Y_l(B_i \overset{+}{\leftarrow} tR)\Big|_{t=0}\right] = \Sigma^{-1} \frac{\mathrm{d}}{\mathrm{d}t} G_l(b_i \overset{+}{\leftarrow} t\,\mathrm{tr}(R))\Big|_{t=0}. \tag{22}$$

When $l < i$, we can easily verify eq. (22) since both sides equal $0$. When $l = i$, we can get

$$
\begin{aligned}
\mathbb{E}_W\left[W^\top \frac{\mathrm{d}}{\mathrm{d}t}Y_l(B_i \xleftarrow{+} tR)\Big|_{t=0}\right] &= \mathbb{E}_W\left[W^\top RY_{l-1}M\left(X_{l-1}^\top C_l X_{l-1} + D_l\right)\right] \\
&= \mathbb{E}_W\left[W^\top RW\right]G_{l-1}M\left(X_{l-1}^\top C_l X_{l-1} + D_l\right) \\
&= \mathrm{tr}(R)\Sigma^{-1}G_{l-1}M\left(X_{l-1}^\top C_l X_{l-1} + D_l\right) \\
&= \Sigma^{-1}\frac{\mathrm{d}}{\mathrm{d}t}G_l(b_i \xleftarrow{+} t\,\mathrm{tr}(R))\Big|_{t=0}.
\end{aligned}
$$

Suppose that eq. (22) holds for some $l = k - 1 \ge i$. One can then verify

$$
\begin{aligned}
&\mathbb{E}_W\left[W^\top \frac{\mathrm{d}}{\mathrm{d}t}Y_k(B_i \xleftarrow{+} tR)\Big|_{t=0}\right] \\
&= \mathbb{E}_W\left[W^\top \frac{\mathrm{d}}{\mathrm{d}t}Y_{k-1}(B_i \xleftarrow{+} tR)\left(I + b_k M(X_{k-1}^\top C_k X_{k-1} + D_k)\right)\Big|_{t=0}\right] \\
&= \mathbb{E}_W\left[W^\top \frac{\mathrm{d}}{\mathrm{d}t}Y_{k-1}(B_i \xleftarrow{+} tR)\Big|_{t=0}\right]\left(I + b_k M(X_{k-1}^\top C_k X_{k-1} + D_k)\right) \\
&= \Sigma^{-1}\frac{\mathrm{d}}{\mathrm{d}t}G_{k-1}(b_i \xleftarrow{+} t\,\mathrm{tr}(R))\Big|_{t=0}\left(I + b_k M(X_{k-1}^\top C_k X_{k-1} + D_k)\right) \\
&= \Sigma^{-1}\frac{\mathrm{d}}{\mathrm{d}t}G_k(b_i \xleftarrow{+} t\,\mathrm{tr}(R))\Big|_{t=0}.
\end{aligned}
$$

The proof of eq. (22) is complete. Now, look at the in-context risk, we have

$$
\begin{aligned}
\frac{\mathrm{d}}{\mathrm{d}t}\mathcal{L}(B_i \xleftarrow{+} tR)\Big|_{t=0} &= 2\,\mathbb{E}_{X_0,W}\left[\mathrm{tr}\left((I-M)Y_L^\top \frac{\mathrm{d}}{\mathrm{d}t}Y_L(B_i \xleftarrow{+} tR)\Big|_{t=0}(I-M)\right)\right] \\
&= 2\,\mathbb{E}_{X_0}\left[\mathrm{tr}\left((I-M)G_L^\top \mathbb{E}_W\left[W^\top \frac{\mathrm{d}}{\mathrm{d}t}Y_L(B_i \xleftarrow{+} tR)\Big|_{t=0}\right](I-M)\right)\right] \\
&= 2\,\mathbb{E}_{X_0}\left[\mathrm{tr}\left((I-M)G_L^\top \Sigma^{-1}\frac{\mathrm{d}}{\mathrm{d}t}G_L(b_i \xleftarrow{+} t\,\mathrm{tr}(R))\Big|_{t=0}(I-M)\right)\right] \\
&= 2\,\mathbb{E}_{X_0,W}\left[\mathrm{tr}\left((I-M)Y_L^\top \frac{\mathrm{d}}{\mathrm{d}t}Y_L(B_i \xleftarrow{+} t\,\mathrm{tr}(R)I_d)\Big|_{t=0}(I-M)\right)\right] \\
&= \frac{\mathrm{d}}{\mathrm{d}t}\mathcal{L}(B_i \xleftarrow{+} t\,\mathrm{tr}(R)I_d)\Big|_{t=0}.
\end{aligned}
$$

This concludes that eq. (16) holds for $B_i$, $i \in [1, L]$.

### 3. Equation (16) holds for $C_i$.

Similar to the $A_i$ case, we will first prove that for any $l \in [1, L]$,

$$
\frac{\mathrm{d}}{\mathrm{d}t}X_l(X_0 \xleftarrow{\times} U_\Sigma, C_i \xleftarrow{+} tR)\Big|_{t=0} = U_\Sigma \frac{\mathrm{d}}{\mathrm{d}t}X_l(C_i \xleftarrow{+} tU_\Sigma^\top RU_\Sigma)\Big|_{t=0}. \tag{23}
$$

The equation above holds trivially for $l < i$. For the case $l = i$, we have

$$
\begin{aligned}
&\frac{\mathrm{d}}{\mathrm{d}t}X_l(X_0 \xleftarrow{\times} U_\Sigma, C_i \xleftarrow{+} tR)\Big|_{t=0} \\
&= A_j X_{l-1}(X_0 \xleftarrow{\times} U_\Sigma)MX_{l-1}^\top(X_0 \xleftarrow{\times} U_\Sigma)RX_{l-1}(X_0 \xleftarrow{\times} U_\Sigma) \\
&= U_\Sigma A_j X_{l-1}MX_{l-1}^\top U_\Sigma^\top RU_\Sigma X_{l-1} = U_\Sigma \frac{\mathrm{d}}{\mathrm{d}t}X_l(C_i \xleftarrow{+} tU_\Sigma^\top RU_\Sigma)\Big|_{t=0}.
\end{aligned}
$$

One can conclude the proof of eq. (23) through a similar reduction as eq. (18) for $l > i$ layers. Next, we establish the corresponding result for $G_l$:

$$
\frac{\mathrm{d}}{\mathrm{d}t}G_l(X_0 \xleftarrow{\times} U_\Sigma, C_i \xleftarrow{+} tR)\Big|_{t=0} = U_\Sigma \frac{\mathrm{d}}{\mathrm{d}t}G_l(C_i \xleftarrow{+} tU_\Sigma^\top RU_\Sigma)\Big|_{t=0}. \tag{24}
$$

This equation holds trivially for $l < i$. When taking $l = i$, we can verify that

$$\frac{\mathrm{d}}{\mathrm{d}t} G_l(X_0 \overset{\times}{\leftarrow} U_\Sigma, C_i \overset{+}{\leftarrow} tR)\Big|_{t=0} = b_l G_{l-1}(X_0 \overset{\times}{\leftarrow} U_\Sigma) M X_{l-1}^\top (X_0 \overset{\times}{\leftarrow} U_\Sigma) R X_{l-1}(X_0 \overset{\times}{\leftarrow} U_\Sigma)$$

$$= b_l U_\Sigma G_{l-1}(X_0 \overset{\times}{\leftarrow} U_\Sigma) M X_{l-1}^\top U_\Sigma^\top R U_\Sigma X_{l-1}$$

$$= U_\Sigma \frac{\mathrm{d}}{\mathrm{d}t} G_l(C_i \overset{+}{\leftarrow} t U_\Sigma^\top R U_\Sigma)\Big|_{t=0} .$$

For $l > i$ layers, one can follow similar reductions as eq. (20) to finish the proof. We then consider the in-context risk:

$$\frac{\mathrm{d}}{\mathrm{d}t} \mathcal{L}(C_i \overset{+}{\leftarrow} tR)\Big|_{t=0}$$

$$= 2 \mathbb{E}_{X_0, W, U_\perp}\left[ \mathrm{tr}\left( (I - M) Y_L^\top (X_0 \overset{\times}{\leftarrow} U_\Sigma) \frac{\mathrm{d}}{\mathrm{d}t} Y_L(X_0 \overset{\times}{\leftarrow} U_\Sigma, C_i \overset{+}{\leftarrow} tR)\Big|_{t=0} (I - M) \right) \right]$$

$$= 2 \mathbb{E}_{X_0, W, U_\perp}\left[ \mathrm{tr}\left( (I - M) G_L^\top U_\Sigma^\top W^\top W U_\Sigma \frac{\mathrm{d}}{\mathrm{d}t} G_L(C_i \overset{+}{\leftarrow} tR)\Big|_{t=0} (I - M) \right) \right]$$

$$= 2d \, \mathbb{E}_{X_0}\left[ \mathrm{tr}\left( (I - M) G_L^\top \Sigma^{-1} \frac{\mathrm{d}}{\mathrm{d}t} \mathbb{E}_{U_\perp}\left[ G_L(C_i \overset{+}{\leftarrow} t U_\Sigma^\top R U_\Sigma) \right]\Big|_{t=0} (I - M) \right) \right]$$

$$= 2d \, \mathbb{E}_{X_0}\left[ \mathrm{tr}\left( (I - M) G_L^\top \Sigma^{-1} \frac{\mathrm{d}}{\mathrm{d}t} G_L(C_i \overset{+}{\leftarrow} tr\Sigma^{-1})\Big|_{t=0} (I - M) \right) \right]$$

$$= \frac{\mathrm{d}}{\mathrm{d}t} \mathbb{E}_{X_0, W}\left[ \mathrm{tr}\left( (I - M) Y_L^\top (C_i \overset{+}{\leftarrow} tr\Sigma^{-1}) Y_L(C_i \overset{+}{\leftarrow} tr\Sigma^{-1})(I - M) \right) \right]\Big|_{t=0}$$

$$= \frac{\mathrm{d}}{\mathrm{d}t} \mathcal{L}(C_i \overset{+}{\leftarrow} tr\Sigma^{-1})\Big|_{t=0} ,$$

where $r = \mathbb{E}_{U_\perp}[U_\Sigma^\top R U_\Sigma] = \frac{1}{d} \mathrm{tr}\left( \Sigma^{1/2} R \Sigma^{1/2} \right)$. This concludes that eq. (16) holds for $C_i$.

**4. Equation (16) holds for $D_i$.**

Let $U_p \in \mathbb{R}^{n \times n}$ be a uniformly sampled permutation matrix, i.e., a binary matrix that has exactly one 1 entry in each row and column with all other entries 0. Let $U_\circ = \mathrm{diag}(U_p \otimes I_2, I_2) \in \mathbb{R}^{(2n+2) \times (2n+2)}$. One can verify that by multiplying $X_0 U_\circ$, it is equal to shuffling the first $n$ 2-column sub-blocks of $X_0$ and keeping the last 2 columns unchanged.

Then, consider a matrix $U_\xi = \mathrm{diag}(\xi_1, \ldots, \xi_{n+1}) \in \mathbb{R}^{(n+1) \times (n+1)}$ where $\xi_i \overset{\text{i.i.d.}}{\sim} \mathrm{Unif}\{\pm 1\}$, i.e., a diagonal matrix with random $\pm 1$ entries. Let $U_\pm = U_\xi \otimes I_2 \in \mathbb{R}^{(2n+2) \times (2n+2)}$. Thus, $U_\pm = U_\pm^\top$ and $X_0 U_\pm$ is randomly flipping the sign of each 2-column sub-block in $X_0$.

We are going to prove that for any $l \in [1, L]$, recalling that $f(A \overset{\diamond}{\leftarrow} B) = f(A \leftarrow AB)$,

$$X_l(X_0 \overset{\diamond}{\leftarrow} U_\pm U_\circ) = X_l U_\pm U_\circ, \tag{25}$$

$$G_l(X_0 \overset{\diamond}{\leftarrow} U_\pm U_\circ) = G_l U_\pm U_\circ. \tag{26}$$

Equation (25) holds trivially for $l = 0$. When eq. (25) holds for some $l = k - 1$, we can verify that

$$X_k(X_0 \overset{\diamond}{\leftarrow} U_\pm U_\circ)$$

$$= X_{k-1} U_\pm U_\circ + A_k X_{k-1} U_\pm U_\circ M \left( U_\circ^\top U_\pm^\top X_{k-1}^\top C_k X_{k-1} U_\pm U_\circ + D_k \right)$$

$$= X_{k-1} U_\pm U_\circ + A_k X_{k-1} U_\pm U_\circ M U_\circ^\top U_\pm^\top \left( X_{k-1}^\top C_k X_{k-1} + U_\pm U_\circ D_k U_\circ^\top U_\pm^\top \right) U_\pm U_\circ$$

$$= X_{k-1} U_\pm U_\circ + A_k X_{k-1} M \left( X_{k-1}^\top C_k X_{k-1} + D_k \right) U_\pm U_\circ$$

$$= \left( X_{k-1} + A_k X_{k-1} M \left( X_{k-1}^\top C_k X_{k-1} + D_k \right) \right) U_\pm U_\circ = X_k U_\pm U_\circ.$$

It uses the fact that there exists some $D_i^1, D_i^2 \in \mathbb{R}^{2 \times 2}$ such that $D_i = \mathrm{diag}(I_n \otimes D_i^1, D_i^2)$, so shuffling the first $n$ $2 \times 2$ diagonal sub-blocks of $D_i$ does not change the matrix, and we have $U_\circ D_i U_\circ^\top = D_i$. Similarly, we have $U_\pm D_k U_\pm^\top = D_k$. This concludes eq. (25), and eq. (26) could be acquired similarly.

Next, we will establish the following equalities for $X_l$ and $G_l$:

$$\frac{\mathrm{d}}{\mathrm{d}t}X_l(X_0 \overset{\diamond}{\leftarrow} U_\pm U_\circ, D_i \overset{+}{\leftarrow} tR)\Big|_{t=0} = \frac{\mathrm{d}}{\mathrm{d}t}X_l(D_i \overset{+}{\leftarrow} tU_\pm U_\circ RU_\circ^\top U_\pm^\top)\Big|_{t=0} U_\pm U_\circ, \quad (27)$$

$$\frac{\mathrm{d}}{\mathrm{d}t}G_l(X_0 \overset{\diamond}{\leftarrow} U_\pm U_\circ, D_i \overset{+}{\leftarrow} tR)\Big|_{t=0} = \frac{\mathrm{d}}{\mathrm{d}t}G_l(D_i \overset{+}{\leftarrow} tU_\pm U_\circ RU_\circ^\top U_\pm^\top)\Big|_{t=0} U_\pm U_\circ. \quad (28)$$

The proof follows by similar reductions as proving eqs. (18) and (20).

Finally, we consider the in-context risk under the permutation of $U_p$ and $U_\xi$. Since each pair of $(x_i, y_i)$ is equivalently sampled from Gaussian distributions, we have $X_0 \overset{d}{=} X_0 U_\pm U_\circ$. Therefore,

$$\frac{\mathrm{d}}{\mathrm{d}t}\mathcal{L}(D_i \overset{+}{\leftarrow} tR)\Big|_{t=0}$$

$$= 2\,\mathbb{E}_{X_0,W}\left[\mathrm{tr}\left((I-M)Y_L^\top \frac{\mathrm{d}}{\mathrm{d}t}Y_L(D_i \overset{+}{\leftarrow} tR)\Big|_{t=0}(I-M)\right)\right]$$

$$= 2\,\mathbb{E}_{X_0,W,U_p,U_\xi}\left[\mathrm{tr}\left((I-M)Y_L^\top(X_0 \overset{\diamond}{\leftarrow} U_\pm U_\circ)\frac{\mathrm{d}}{\mathrm{d}t}Y_L(X_0 \overset{\diamond}{\leftarrow} U_\pm U_\circ, D_i \overset{+}{\leftarrow} tR)\Big|_{t=0}(I-M)\right)\right]$$

$$= 2d\,\mathbb{E}_{X_0,U_p,U_\xi}\left[\mathrm{tr}\left((I-M)U_\circ^\top U_\pm^\top G_L^\top \Sigma^{-1}\frac{\mathrm{d}}{\mathrm{d}t}G_L(D_i \overset{+}{\leftarrow} tU_\pm U_\circ RU_\circ^\top U_\pm^\top)\Big|_{t=0}U_\pm U_\circ(I-M)\right)\right]$$

$$= 2d\,\mathbb{E}_{X_0}\left[\mathrm{tr}\left((I-M)G_L^\top \Sigma^{-1}\frac{\mathrm{d}}{\mathrm{d}t}\mathbb{E}_{U_p,U_\xi}\left[G_L(D_i \overset{+}{\leftarrow} tU_\pm U_\circ^\top RU_\circ U_\pm)\right]\Big|_{t=0}(I-M)\right)\right]$$

$$= 2d\,\mathbb{E}_{X_0}\left[\mathrm{tr}\left((I-M)G_L^\top \Sigma^{-1}\frac{\mathrm{d}}{\mathrm{d}t}G_L(D_i \overset{+}{\leftarrow} t\widetilde{R})\Big|_{t=0}(I-M)\right)\right] = \frac{\mathrm{d}}{\mathrm{d}t}\mathcal{L}(D_i \overset{+}{\leftarrow} t\widetilde{R})\Big|_{t=0},$$

where $\widetilde{R} = \mathbb{E}_{U_p,U_\xi}[U_\pm U_\circ^\top RU_\circ U_\pm] = \mathrm{diag}(I_n \otimes R^1, R^2)$, $R^1 = \frac{1}{n}\sum_{j=1}^n R_j$, $R^2 = R_{n+1}$, and $R_j$ is the $j$-th $2 \times 2$ diagonal block of $R$. The 4th equality uses the fact that $\mathrm{tr}[(I-M)A(I-M)]$ is extracting the right-bottom element of $A$, so it should be equal to $\mathrm{tr}[(I-M)U_\circ^\top U_\pm^\top AU_\pm U_\circ(I-M)]$ for any matrix $A$. This concludes that eq. (16) holds for $D_i$.

Till now, we have proved that eq. (16) holds for each one of $A_i, B_i, C_i, D_i$. The proof of the whole theorem is then completed by applying Lemma 8. □

### D.3 PROOF OF THEOREM 2

*Proof.* In this proof, we follow the same notations as the proof of Theorem 1, where the constant $\frac{1}{n}$ factor is dropped and $\widetilde{Z}_0, \widetilde{X}_0, \widetilde{Y}_0$ are simplified as $Z_0, X_0, Y_0$ respectively.

$$Z_0 = \begin{bmatrix} x_1 & 0 & 0 & \cdots & x_n & 0 & 0 & x_{\text{test}} & 0 & 0 \\ 0 & 0 & y_1 & \cdots & 0 & 0 & y_n & 0 & 0 & y_{\text{test}} \end{bmatrix} \in \mathbb{R}^{(2d) \times (3n+3)}. \quad (29)$$

Let $Z_l \in \mathbb{R}^{2d \times (3n+3)}$ be the $l$-th layer's output and let $X_l, Y_l \in \mathbb{R}^{d \times (3n+3)}$ be its first and last $d$ rows. Our goal is to prove that, for any $E \in A \cup B \cup C \cup D$ and an arbitrary matrix $R \in \mathbb{R}^{d \times d}$ ($\mathbb{R}^{d_p \times d_p} \, for \, D$), there exists $\widetilde{R} \in \mathcal{S}_I$ ($\mathcal{S}_\Sigma$ for C, $\mathcal{S}_P$ for D) such that

$$\frac{\mathrm{d}}{\mathrm{d}t}\mathcal{L}(E \overset{+}{\leftarrow} t\widetilde{R})\Big|_{t=0} \leq \frac{\mathrm{d}}{\mathrm{d}t}\mathcal{L}(E \overset{+}{\leftarrow} tR)\Big|_{t=0}. \quad (30)$$

The proofs of eq. (30) for $A_i$, $B_i$ and $C_i$ are identical with the proof of Theorem 1 so we omit them. We will be focusing on $D_i$ for the rest of the proof.

Let $U_p^s \in \mathbb{R}^{n \times n}$ and $U_p^t \in \mathbb{R}^{(n+1) \times (n+1)}$ be uniformly sampled permutation matrices. Let $U_\circ^s = \mathrm{diag}(U_p^s, 1) \otimes \mathrm{diag}(1, 0, 1)$ and $U_\circ^t = U_p^t \otimes \mathrm{diag}(0, 1, 0)$. Therefore, $X_0 U_\circ^s$ is shuffling the 1-st and 3-rd columns among each 3-column sub-block of $X_0$ (except for the last 3-column sub-block), and $X_0 U_\circ^s$ is shuffling the 2-nd column among each 3-column sub-block. Next, let $U_\xi^s, U_\xi^t \in \mathbb{R}^{(n+1) \times (n+1)}$ be diagonal matrices with uniformly sampled $\pm 1$ entries. Define $U_\pm^s = U_\xi^s \otimes \mathrm{diag}(1, 0, 1)$ and $U_\pm^t = U_\xi^t \otimes \mathrm{diag}(0, 1, 0)$. It can then be verified that $X_0 U_\pm^s U_\pm^t \overset{d}{=} X_0$.

To simplify the notations, let $U_{\equiv}$ denote $U_{\pm}^s U_{\pm}^t U_{\circ}^s U_{\circ}^t$. We will focus on a subset of $\mathcal{S}_P$:

$$\mathcal{S}_P' = \left\{ \mathrm{diag}(I_n \otimes \Lambda_1, \Lambda_2) + I_{n+1} \otimes \Lambda_3 \ \middle| \ \Lambda_1, \Lambda_2 \in \mathcal{M}\left(\begin{smallmatrix} 1 & 0 & 1 \\ 0 & 0 & 0 \\ 1 & 0 & 1 \end{smallmatrix}\right), \Lambda_3 \in \mathcal{M}\left(\begin{smallmatrix} 0 & 0 & 0 \\ 0 & 1 & 0 \\ 0 & 0 & 0 \end{smallmatrix}\right) \right\}.$$

Assume $D_k = \mathrm{diag}(I_n \otimes \Lambda_1, \Lambda_2) + I_{n+1} \otimes \Lambda_3 \in \mathcal{S}_P'$ as defined above, one can verify that it is a block-diagonal matrix constructed from the same $3 \times 3$ sub-blocks, and thus is invariant under $U_{\equiv} D_k U_{\equiv}^\top$. We will then prove that for any $l \in [1, L]$,

$$X_l(X_0 \overset{\diamond}{\leftarrow} U_{\equiv}) = X_l U_{\equiv}, \tag{31}$$

$$G_l(X_0 \overset{\diamond}{\leftarrow} U_{\equiv}) = G_l U_{\equiv}, \tag{32}$$

$$\frac{\mathrm{d}}{\mathrm{d}t} X_l(X_0 \overset{\diamond}{\leftarrow} U_{\equiv}, D_i \overset{\pm}{\leftarrow} tR)\Big|_{t=0} = \frac{\mathrm{d}}{\mathrm{d}t} X_l(D_i \overset{\pm}{\leftarrow} tU_{\equiv}RU_{\equiv}^\top)\Big|_{t=0} U_{\equiv}, \tag{33}$$

$$\frac{\mathrm{d}}{\mathrm{d}t} G_l(X_0 \overset{\diamond}{\leftarrow} U_{\equiv}, D_i \overset{\pm}{\leftarrow} tR)\Big|_{t=0} = \frac{\mathrm{d}}{\mathrm{d}t} G_l(D_i \overset{\pm}{\leftarrow} tU_{\equiv}RU_{\equiv}^\top)\Big|_{t=0} U_{\equiv}. \tag{34}$$

These results can be acquired by similar proofs as eqs. (25) to (28). We then consider the in-context risk under the permutations of $U_{\equiv}$. Similarly, we have $X_0 \overset{d}{=} X_0 U_{\equiv}$ and

$$\frac{\mathrm{d}}{\mathrm{d}t} \mathcal{L}(D_i \overset{\pm}{\leftarrow} tR)\Big|_{t=0}$$

$$= 2 \mathbb{E}_{X_0, W}\left[\mathrm{tr}\left((I - M)Y_L^\top \frac{\mathrm{d}}{\mathrm{d}t} Y_L(D_i \overset{\pm}{\leftarrow} tR)\Big|_{t=0} (I - M)\right)\right]$$

$$= 2d \,\mathbb{E}_{X_0, U_{\equiv}}\left[\mathrm{tr}\left((I - M)G_L^\top(X_0 \overset{\diamond}{\leftarrow} U_{\equiv})\Sigma^{-1} \frac{\mathrm{d}}{\mathrm{d}t} G_L(X_0 \overset{\diamond}{\leftarrow} U_{\equiv}, D_i \overset{\pm}{\leftarrow} tR)\Big|_{t=0} (I - M)\right)\right]$$

$$= 2d \,\mathbb{E}_{X_0, U_{\equiv}}\left[\mathrm{tr}\left((I - M)U_{\equiv}^\top G_L^\top \Sigma^{-1} \frac{\mathrm{d}}{\mathrm{d}t} G_L(D_i \overset{\pm}{\leftarrow} tU_{\equiv}RU_{\equiv}^\top)\Big|_{t=0} U_{\equiv}(I - M)\right)\right]$$

$$= 2d \,\mathbb{E}_{X_0}\left[\mathrm{tr}\left((I - M)G_L^\top \Sigma^{-1} \frac{\mathrm{d}}{\mathrm{d}t} G_L(D_i \overset{\pm}{\leftarrow} t\,\mathbb{E}_{U_{\equiv}}[U_{\equiv}RU_{\equiv}^\top])\Big|_{t=0} (I - M)\right)\right]$$

$$= \frac{\mathrm{d}}{\mathrm{d}t} \mathcal{L}(D_i \overset{\pm}{\leftarrow} t\widetilde{R})\Big|_{t=0}.$$

Let $R_j$ be the $j$-th $3 \times 3$ diagonal block of $R$, then $R^1 = \frac{1}{n}\sum_{j=1}^n R_j \circ \left(\begin{smallmatrix} 1 & 0 & 1 \\ 0 & 0 & 0 \\ 1 & 0 & 1 \end{smallmatrix}\right)$, $R^2 = R_{n+1} \circ \left(\begin{smallmatrix} 1 & 0 & 1 \\ 0 & 0 & 0 \\ 1 & 0 & 1 \end{smallmatrix}\right)$, $R^3 = \frac{1}{n+1}\sum_{j=1}^{n+1} R_j \circ \left(\begin{smallmatrix} 0 & 0 & 0 \\ 0 & 1 & 0 \\ 0 & 0 & 0 \end{smallmatrix}\right)$ and $\widetilde{R} = \mathbb{E}_{U_{\equiv}}[U_{\equiv}RU_{\equiv}^\top] = \mathrm{diag}(I_n \otimes R^1, R^2) + I_{n+1} \otimes R^3$. This indicates that eq. (30) holds for each $D_i \in \mathcal{S}_P'$, and thus the proof of the whole theorem completes by applying Lemma 8 and noticing that $\mathcal{S}_P' \subset \mathcal{S}_P$. $\qquad\square$

## D.4 PROOF OF THEOREM 7

*Proof.* We keep the same notations as the proof of Theorem 1, dropping the $\frac{1}{n}$ factor and simplifying $\widetilde{X}_0, \widetilde{Y}_0, \widetilde{Z}_0$ as $X_0, Y_0, Z_0$, as follows:

$$Z_0 = \begin{bmatrix} 0 & 0 & \cdots & 0 & 0 & 0 & 0 \\ x_1 & y_1 & \cdots & x_n & y_n & x_{\text{test}} & y_{\text{test}} \end{bmatrix} \in \mathbb{R}^{(2d) \times (2n+2)}. \tag{35}$$

Note that we now have $X_0$ and $Y_0$ containing both $x_i$ and $y_i$. Define

$$X = \begin{bmatrix} x_1 & 0 & \cdots & x_n & 0 & x_{\text{test}} & 0 \end{bmatrix},$$
$$\overline{X} = \begin{bmatrix} 0 & x_1 & \cdots & 0 & x_n & 0 & x_{\text{test}} \end{bmatrix},$$
$$Y = \begin{bmatrix} 0 & y_1 & \cdots & 0 & y_n & 0 & y_{\text{test}} \end{bmatrix}.$$

we then have $Y_0 = X + Y = X + W\overline{X}$. From the parameter configuration in eq. (12), the update rule of the first attention layer is

$$X_1 = A_1 Y_0 M D_1 = A_1 X M D_1, \quad Y_1 = Y_0 = X + W\overline{X}. \tag{36}$$

The update rule for the following layers is the same as eq. (15). We are going to prove that, for any $E \in A \cup B \cup C \cup D$ and an arbitrary matrix $R \in \mathbb{R}^{d \times d}$ ($\mathbb{R}^{d_p \times d_p}$ for $D$), there exists $\widetilde{R} \in \mathcal{S}_I$ ($\mathcal{S}_\Sigma$ for $C$, $\mathcal{S}_P$ for $D$) such that

$$\frac{\mathrm{d}}{\mathrm{d}t}\mathcal{L}(E \xleftarrow{\pm} t\widetilde{R})\Big|_{t=0} \leq \frac{\mathrm{d}}{\mathrm{d}t}\mathcal{L}(E \xleftarrow{\pm} tR)\Big|_{t=0}. \tag{37}$$

Similarly to Theorem 1, we uniformly sample $U_\perp \in \mathbb{R}^{d \times d}$ as an orthonormal random matrix, and let $U_\Sigma = \Sigma^{1/2} U_\perp \Sigma^{-1/2}$. Under the condition that $B_l = b_l I_d$ for some $b_l \in \mathbb{R}$, we have

$$Y_l = Y_1 \prod_{j=2}^{l} \big(I + b_j M\big(X_{j-1}^\top C_j X_{j-1} + D_j\big)\big).$$

Let $F_l = X \prod_{j=2}^{l}\big(I + b_j M\big(X_{j-1}^\top C_j X_{j-1} + D_j\big)\big)$, $G_l = \overline{X} \prod_{j=2}^{l}\big(I + b_j M\big(X_{j-1}^\top C_j X_{j-1} + D_j\big)\big)$, we then have $Y_l = F_l + WG_l$. According to Lemma 9,

$$\frac{\mathrm{d}}{\mathrm{d}t}\mathcal{L}(E \xleftarrow{\pm} tR)\Big|_{t=0}$$

$$= \frac{\mathrm{d}}{\mathrm{d}t}\mathbb{E}_{X_0,W}\Big[\mathrm{tr}\Big((I-M)Y_L^\top (E \xleftarrow{\pm} tR)Y_L(E \xleftarrow{\pm} tR)(I-M)\Big)\Big]\Big|_{t=0}$$

$$= \frac{\mathrm{d}}{\mathrm{d}t}\mathbb{E}_{X_0,W}\Big[\mathrm{tr}\Big((I-M)F_L^\top (E \xleftarrow{\pm} tR)F_L(E \xleftarrow{\pm} tR)(I-M)\Big)\Big]\Big|_{t=0}$$

$$+ \frac{\mathrm{d}}{\mathrm{d}t}\mathbb{E}_{X_0,W}\Big[\mathrm{tr}\Big((I-M)G_L^\top (E \xleftarrow{\pm} tR)W^\top WG_L(E \xleftarrow{\pm} tR)(I-M)\Big)\Big]\Big|_{t=0}$$

$$= 2\mathbb{E}_{X_0}\Big[\mathrm{tr}\Big((I-M)F_L^\top \frac{\mathrm{d}}{\mathrm{d}t}F_L(E \xleftarrow{\pm} tR)\Big|_{t=0}(I-M)\Big)\Big]$$

$$+ 2d\,\mathbb{E}_{X_0}\Big[\mathrm{tr}\Big((I-M)G_L^\top \Sigma^{-1}\frac{\mathrm{d}}{\mathrm{d}t}G_L(E \xleftarrow{\pm} tR)\Big|_{t=0}(I-M)\Big)\Big].$$

Next, we will show that eq. (37) holds for each one of $A_i, B_i, C_i, D_i$ for any $i \in [1, L]$.

### 1. Equation (37) holds for $A_i$.

One can easily verify that eqs. (17) and (18) still hold. Furthermore, eqs. (19) and (20) hold for both $F_l$ and $G_l$. With these observations, we can then verify

$$\frac{\mathrm{d}}{\mathrm{d}t}\mathcal{L}(A_i \xleftarrow{\pm} tR)\Big|_{t=0}$$

$$= 2\mathbb{E}_{X_0,U_\perp}\Big[\mathrm{tr}\Big((I-M)F_L^\top (X \xleftarrow{\times} U_\Sigma)\frac{\mathrm{d}}{\mathrm{d}t}F_L(X \xleftarrow{\times} U_\Sigma, A_i \xleftarrow{\pm} tR)\Big|_{t=0}(I-M)\Big)\Big]$$

$$+ 2d\,\mathbb{E}_{X_0,U_\perp}\Big[\mathrm{tr}\Big((I-M)G_L^\top (X \xleftarrow{\times} U_\Sigma)\Sigma^{-1}\frac{\mathrm{d}}{\mathrm{d}t}G_L(X \xleftarrow{\times} U_\Sigma, A_i \xleftarrow{\pm} tR)\Big|_{t=0}(I-M)\Big)\Big]$$

$$= 2\mathbb{E}_{X_0,U_\perp}\Big[\mathrm{tr}\Big((I-M)F_L^\top U_\Sigma^\top U_\Sigma \frac{\mathrm{d}}{\mathrm{d}t}F_L(A_i \xleftarrow{\pm} tU_\Sigma^{-1}RU_\Sigma)\Big|_{t=0}(I-M)\Big)\Big]$$

$$+ 2d\,\mathbb{E}_{X_0,U_\perp}\Big[\mathrm{tr}\Big((I-M)G_L^\top U_\Sigma^\top \Sigma^{-1}U_\Sigma \frac{\mathrm{d}}{\mathrm{d}t}G_L(A_i \xleftarrow{\pm} tU_\Sigma^{-1}RU_\Sigma)\Big|_{t=0}(I-M)\Big)\Big]$$

$$= 2\mathbb{E}_{X_0}\Big[\mathrm{tr}\Big((I-M)F_L^\top \frac{\mathrm{d}}{\mathrm{d}t}F_L(A_i \xleftarrow{\pm} trI_d)\Big|_{t=0}(I-M)\Big)\Big]$$

$$+ 2d\,\mathbb{E}_{X_0}\Big[\mathrm{tr}\Big((I-M)G_L^\top \Sigma^{-1}\frac{\mathrm{d}}{\mathrm{d}t}G_L(A_i \xleftarrow{\pm} trI_d)\Big|_{t=0}(I-M)\Big)\Big]$$

$$= \frac{\mathrm{d}}{\mathrm{d}t}\mathcal{L}(A_i \xleftarrow{\pm} trI_d)\Big|_{t=0},$$

where $r = \mathbb{E}_{U_\perp}[U_\Sigma^{-1}RU_\Sigma] = \frac{1}{d}\mathrm{tr}\big(\Sigma^{-1/2}R\Sigma^{1/2}\big)$.

**2. Equation (37) holds for $B_i$.**

From the definition of $F_l$ and $G_l$, we can verify that

$$\frac{\mathrm{d}}{\mathrm{d}t}Y_l(B_i \overset{\pm}{\leftarrow} tR)\Big|_{t=0}$$

$$= R(F_{i-1} + WG_{i-1})M(X_{i-1}^\top C_i X_{i-1} + D_i) \prod_{j=i+1}^{l} \left(I + b_j M(X_{j-1}^\top C_j X_{j-1} + D_j)\right).$$

Define

$$\overline{F}_l^i = \left(F_{i-1} + B_i F_{i-1} M(X_{i-1}^\top C_i X_{i-1} + D_i)\right) \prod_{j=i+1}^{l} \left(I + b_j M(X_{j-1}^\top C_j X_{j-1} + D_j)\right),$$

$$\overline{G}_l^i = \left(WG_{i-1} + B_i WG_{i-1} M(X_{i-1}^\top C_i X_{i-1} + D_i)\right) \prod_{j=i+1}^{l} \left(I + b_j M(X_{j-1}^\top C_j X_{j-1} + D_j)\right),$$

We then have

$$\frac{\mathrm{d}}{\mathrm{d}t}Y_l(B_i \overset{\pm}{\leftarrow} tR)\Big|_{t=0} = \frac{\mathrm{d}}{\mathrm{d}t}\overline{F}_l^i(B_i \overset{\pm}{\leftarrow} tR)\Big|_{t=0} + \frac{\mathrm{d}}{\mathrm{d}t}\overline{G}_l^i(B_i \overset{\pm}{\leftarrow} tR)\Big|_{t=0}.$$

Similar to eqs. (20) and (22), we can prove that

$$\frac{\mathrm{d}}{\mathrm{d}t}\overline{F}_l^i(X_0 \overset{\times}{\leftarrow} U_\Sigma, B_i \overset{\pm}{\leftarrow} tR)\Big|_{t=0} = U_\Sigma \frac{\mathrm{d}}{\mathrm{d}t}\overline{F}_l^i(B_i \overset{\pm}{\leftarrow} tU_\Sigma^{-1}RU_\Sigma)\Big|_{t=0},$$

$$\mathbb{E}_W\left[W^\top \frac{\mathrm{d}}{\mathrm{d}t}\overline{G}_l^i(B_i \overset{\pm}{\leftarrow} tR)\Big|_{t=0}\right] = \Sigma^{-1} \frac{\mathrm{d}}{\mathrm{d}t}\overline{G}_l^i(B_i \overset{\pm}{\leftarrow} t\operatorname{tr}(R)I_d)\Big|_{t=0}.$$

Without loss of generality, we assume that $r = \frac{1}{d}\operatorname{tr}\left(\Sigma^{-1/2}R\Sigma^{1/2}\right) \leq \frac{1}{d}\operatorname{tr}(R)$, and let $\gamma = rd/\operatorname{tr}(R) \leq 1$. Then, one can verify that

$$\frac{\mathrm{d}}{\mathrm{d}t}\mathcal{L}(B_i \overset{\pm}{\leftarrow} tR)\Big|_{t=0}$$

$$= 2\,\mathbb{E}_{X_0,U_\perp}\left[\operatorname{tr}\left((I - M)F_l^\top(X \overset{\times}{\leftarrow} U_\Sigma)\frac{\mathrm{d}}{\mathrm{d}t}\overline{F}_l^i(X \overset{\times}{\leftarrow} U_\Sigma, B_i \overset{\pm}{\leftarrow} tR)\Big|_{t=0}(I - M)\right)\right]$$

$$+ 2\,\mathbb{E}_{X_0,W}\left[\operatorname{tr}\left((I - M)G_l^\top W^\top \frac{\mathrm{d}}{\mathrm{d}t}\overline{G}_l^i(B_i \overset{\pm}{\leftarrow} tR)\Big|_{t=0}(I - M)\right)\right]$$

$$= 2\,\mathbb{E}_{X_0}\left[\operatorname{tr}\left((I - M)F_l^\top \frac{\mathrm{d}}{\mathrm{d}t}\overline{F}_l^i(B_i \overset{\pm}{\leftarrow} trI_d)\Big|_{t=0}(I - M)\right)\right]$$

$$+ 2\,\mathbb{E}_{X_0}\left[\operatorname{tr}\left((I - M)G_l^\top \Sigma^{-1} \frac{\mathrm{d}}{\mathrm{d}t}\overline{G}_l^i(B_i \overset{\pm}{\leftarrow} t\operatorname{tr}(R)I_d)\Big|_{t=0}(I - M)\right)\right]$$

$$= 2\,\mathbb{E}_{X_0}\left[\operatorname{tr}\left((I - M)F_l^\top \frac{\mathrm{d}}{\mathrm{d}t}F_l(B_i \overset{\pm}{\leftarrow} trI_d)\Big|_{t=0}(I - M)\right)\right]$$

$$+ \frac{1}{\gamma}2d\,\mathbb{E}_{X_0}\left[\operatorname{tr}\left((I - M)G_l^\top \Sigma^{-1} \frac{\mathrm{d}}{\mathrm{d}t}G_l(B_i \overset{\pm}{\leftarrow} trI_d)\Big|_{t=0}(I - M)\right)\right]$$

$$= \left(\frac{1}{\gamma} - 1\right)2d\,\mathbb{E}_{X_0}\left[\operatorname{tr}\left((I - M)G_l^\top \Sigma^{-1} \frac{\mathrm{d}}{\mathrm{d}t}G_l(B_i \overset{\pm}{\leftarrow} trI_d)\Big|_{t=0}(I - M)\right)\right]$$

$$+ \frac{\mathrm{d}}{\mathrm{d}t}\mathcal{L}(B_i \overset{\pm}{\leftarrow} trI_d)\Big|_{t=0} \geq \frac{\mathrm{d}}{\mathrm{d}t}\mathcal{L}(B_i \overset{\pm}{\leftarrow} trI_d)\Big|_{t=0}.$$

The last inequality assumes the positivity of the term involving $G_l$. Otherwise, one can simply flip the numerator and denominator of $\gamma$ and scale the derivative of $F_l$ instead of $G_l$ to yield an additional positive term besides the risk term to finish the proof.

**3. Equation (37) holds for $C_i$, $D_i$.**

Similarly, one can verify that eqs. (23) and (24) still hold (also eqs. (25) to (28)), and finish the proof by following the same reductions as Theorem 1 with $F_l$ and $G_l$. $\qquad\square$

### D.5 PROOF OF PROPOSITION 3

*Proof.* Let $A_l = a_l I_d$, $B_l = b_l I_d$, $C_l = c_l I_d$ and $D_l = \text{diag}(I_n \otimes D_l^1, D_l^2) + I_{n+1} \otimes D_l^3 + D_l^4 \otimes D_l^5$ for $l \in [1, 2]$. Let $Z_l \in \mathbb{R}^{2d \times (3n+3)}$ be the output of the $l$-th attention layer, and let $X_l, Y_l \in \mathbb{R}^{d \times (3n+3)}$ be its first and last $d$ rows respectively. Note that $Y_l$ in this proof does not contain $y_{\text{test}}$.

Let $D_1^1 = \begin{pmatrix} d_x^x & 0 & d_x^y \\ 0 & 0 & 0 \\ d_y^x & 0 & d_y^y \end{pmatrix}$, $D_1^2 = \begin{pmatrix} s_x & 0 & s_y \\ 0 & 0 & 0 \\ 0 & 0 & 0 \end{pmatrix}$ (note that the last row of $D_1^2$ is masked out by $M$, so we simply set it to 0), and $D_1^5 = \begin{pmatrix} 0 & t_x & 0 \\ 0 & 0 & 0 \\ 0 & t_y & 0 \end{pmatrix}$. We use $D$ as an abbreviation for $D_1^4$, and use $d_{i,j}$ to denote the elements in $D$. One can verify that

$$X_1 = X_0 + a_1 X_0 M \big( \text{diag}(I_n \otimes D_1^1, D_1^2) + I_{n+1} \otimes D_1^3 + D_1^4 \otimes D_1^5 \big)$$

$$= \begin{bmatrix} (1 + a_1 d_x^x)x_1 & a_1 t_x \sum_{i=1}^{n+1} d_{i,1}x_i & a_1 d_x^y x_1 \\ & \cdots & \\ (1 + a_1 d_x^x)x_n & a_1 t_x \sum_{i=1}^{n+1} d_{i,n}x_i & a_1 d_x^y x_n \\ (1 + a_1 d_x^x)x_{\text{test}} & a_1 t_x \sum_{i=1}^{n+1} d_{i,n+1}x_i & a_1 d_x^y x_{\text{test}} \end{bmatrix}.$$

Similarly, we have

$$Y_1 = Y_0 + b_1 Y_0 M \big( \text{diag}(I_n \otimes D_1^1, D_1^2) + I_{n+1} \otimes D_1^3 + D_1^4 \otimes D_1^5 \big)$$

$$= \begin{bmatrix} b_1 d_y^x y_1 & b_1 t_y \sum_{i=1}^{n} d_{i,1}y_i & (1 + b_1 d_y^y)y_1 \\ & \cdots & \\ b_1 d_y^x y_n & b_1 t_y \sum_{i=1}^{n} d_{i,n}y_i & (1 + b_1 d_y^y)y_n \\ 0 & b_1 t_y \sum_{i=1}^{n} d_{i,n+1}y_i & 0 \end{bmatrix}.$$

By the definition of linear attention, we can show that

$$\mathsf{TF}(Z_0; \{V_l, Q_l\}_{l=1}^2) = (Y_2)_{3n+3} = b_2 Y_1 M \big( c_2 X_1^\top (X_1)_{3n+3} + (D_2)_{3n+3} \big)$$

$$= b_2 c_2 a_1 d_x^y \bigg( \sum_{i=1}^{3n+2} (Y_1)_i (X_1)_i^\top \bigg) x_{\text{test}}.$$

Define $\Delta X_1 = [0 \quad a_1 t_x d_{n+1,1} x_{\text{test}} \quad 0 \quad \cdots \quad 0 \quad a_1 t_x d_{n+1,n+1} x_{\text{test}} \quad 0]$, and let $\overline{X}_1 = X_1 - \Delta X_1$, then $\mathsf{TF}(Z_0; \{V_l, Q_l\}_{l=1}^2) = \mathsf{TF}(Z_0; \{V_l, Q_l\}_{l=1}^2, X_1 \leftarrow \overline{X}_1) + \mathsf{TF}(Z_0; \{V_l, Q_l\}_{l=1}^2, X_1 \leftarrow \Delta X_1)$. Let $b_1 d_y^x (1 + a_1 d_x^x) + (1 + b_1 d_y^y) a_1 d_x^x = a$, $b_1 t_y a_1 t_x = b$, $b_2 c_2 a_1 d_x^y = c$, we then have

$$\mathsf{TF}(Z_0; \{V_l, Q_l\}_{l=1}^2, X_1 \leftarrow \overline{X}_1) = c \bigg( a \sum_{i=1}^n y_i x_i^\top + b \sum_{i=1}^{n+1} \bigg( \sum_{j=1}^n d_{j,i} y_j \bigg) \bigg( \sum_{j=1}^n d_{j,i} x_j^\top \bigg) \bigg) x_{\text{test}}$$

$$= c \bigg( a \sum_{i=1}^n y_i x_i^\top + b \sum_{j=1}^n \sum_{k=1}^n \bigg( \sum_{i=1}^{n+1} d_{j,i} d_{k,i} \bigg) y_j x_k^\top \bigg) x_{\text{test}}, \quad (38)$$

$$\mathsf{TF}(Z_0; \{V_l, Q_l\}_{l=1}^2, X_1 \leftarrow \Delta X_1) = bc \sum_{i=1}^{n+1} \sum_{j=1}^n d_{j,i} y_j d_{n+1,i} x_{\text{test}}^\top x_{\text{test}}$$

$$= bc \sum_{j=1}^n \bigg( \sum_{i=1}^{n+1} d_{j,i} d_{n+1,i} \bigg) y_j x_{\text{test}}^\top x_{\text{test}}. \quad (39)$$

Now consider the in-context risk,

$$\mathcal{L}(V, Q) = \mathbb{E}_{Z_0, W} \| \mathsf{TF}(Z_0; \{V, Q\}) + W x_{\text{test}} \|_2^2$$

$$= \mathbb{E}_{Z_0, W} \Big[ \big( \mathsf{TF}(Z_0; \{V, Q\}) + W x_{\text{test}} \big)^\top \big( \mathsf{TF}(Z_0; \{V, Q\}) + W x_{\text{test}} \big) \Big]$$

$$= \mathbb{E}_{Z_0, W} \Big[ \big( \mathsf{TF}(Z_0; \{V, Q\}, X_1 \leftarrow \overline{X}_1) + W x_{\text{test}} \big)^\top \big( \mathsf{TF}(Z_0; \{V, Q\}, X_1 \leftarrow \overline{X}_1) + W x_{\text{test}} \big) \Big]$$

$$+ 2 \mathbb{E}_{Z_0, W} \Big[ \mathsf{TF}(Z_0; \{V, Q\}, X_1 \leftarrow \Delta X_1)^\top \big( \mathsf{TF}(Z_0; \{V, Q\}, X_1 \leftarrow \overline{X}_1) + W x_{\text{test}} \big) \Big]$$

$$+ \mathbb{E}_{Z_0, W} \Big[ \mathsf{TF}(Z_0; \{V, Q\}, X_1 \leftarrow \Delta X_1)^\top \mathsf{TF}(Z_0; \{V, Q\}, X_1 \leftarrow \Delta X_1) \Big].$$

In the equation above, the 3-rd part is always positive. We then examine the second part:

$$\mathbb{E}_{Z_0,W}\left[\mathsf{TF}(Z_0;\{V,Q\},X_1 \leftarrow \Delta X_1)^\top\left(\mathsf{TF}(Z_0;\{V,Q\},X_1 \leftarrow \overline{X}_1)+Wx_{\text{test}}\right)\right]$$
$$=\mathbb{E}_{Z_0,W}\left[x_{\text{test}}^\top x_{\text{test}}v_1 x_{\text{test}} + x_{\text{test}}^\top x_{\text{test}}v_2 x_{\text{test}}\right]=0,$$

where $v_1 = bc\sum_{j=1}^{n}\left(\sum_{i=1}^{n+1}d_{j,i}d_{n+1,i}\right)y_j^\top c\left(a\sum_{i=1}^{n}y_i x_i^\top + b\sum_{j=1}^{n}\sum_{k=1}^{n}\left(\sum_{i=1}^{n+1}d_{j,i}d_{k,i}\right)y_j x_k^\top\right)$
and $v_2 = bc\sum_{j=1}^{n}\left(\sum_{i=1}^{n+1}d_{j,i}d_{n+1,i}\right)y_j^\top W$ are independent of $x_{\text{test}}$. Therefore, $\mathcal{L}(V,Q)$ attains
its minimum only if $\mathsf{TF}(Z_0;\{V,Q\},X_1 \leftarrow \Delta X_1)=0$, implying $d_{n+1,i}=0$ for $i \in [1,n+1]$.

In the following analysis, we will assume that the last row of $D$ is 0, and let $M \in \mathbb{R}^{n\times(n+1)}$ be
the first $n$ rows of $D$. Additionally, we will drop the $c$ factor in eq. (38), since its position could be
substituted by $a$ and $b$. We then define $\widetilde{W} = a\sum_{i=1}^{n}y_i x_i^\top + b\sum_{j=1}^{n}\sum_{k=1}^{n}\left(\sum_{i=1}^{n+1}d_{j,i}d_{k,i}\right)y_j x_k^\top$,
$X=\begin{bmatrix}x_1 & \cdots & x_n\end{bmatrix}$ and $Y=\begin{bmatrix}y_1 & \cdots & y_n\end{bmatrix}$. One can verify that

$$\widetilde{W}=aYX^\top + bYMM^\top X^\top = aWXX^\top + bWXMM^\top X^\top. \tag{40}$$

Furthermore, the in-context risk could be expanded as

$$\mathcal{L}(V,Q)=\mathbb{E}_{Z_0,W}\left\|\widetilde{W}x_{\text{test}}+Wx_{\text{test}}\right\|_2^2 = \mathbb{E}_{Z_0,W}\left[x_{\text{test}}^\top(\widetilde{W}+W)^\top(\widetilde{W}+W)x_{\text{test}}\right]$$
$$=\mathbb{E}_{Z_0,W}\left[\text{tr}\left((\widetilde{W}+W)^\top(\widetilde{W}+W)\right)\right]$$
$$=\mathbb{E}_{Z_0,W}\left[\text{tr}\left(\widetilde{W}^\top\widetilde{W}\right)+2\,\text{tr}\left(W^\top\widetilde{W}\right)+\text{tr}\left(W^\top W\right)\right].$$

We will use the identity $\mathbb{E}_X[XAX^\top XBX^\top]=\left(\text{tr}(A)\,\text{tr}(B)+\text{tr}\left(AB^\top\right)+d\,\text{tr}(AB)\right)I_d$ for any
$A, B \in \mathbb{R}^{n\times n}$, which can be acquired by expanding each element and applying Isserlis' theorem.
Let $T_1 = \text{tr}\left(MM^\top\right)$ and $T_2 = \text{tr}\left(MM^\top MM^\top\right)$, then

$$\mathbb{E}_{Z_0,W}\left[\text{tr}\left((aWXX^\top + bWXMM^\top X^\top)^\top(aWXX^\top + bWXMM^\top X^\top)\right)\right]$$
$$=\mathbb{E}_{Z_0,W}\left[a^2\,\text{tr}\left(XX^\top W^\top WXX^\top\right)+2ab\,\text{tr}\left(XX^\top W^\top WXMM^\top X^\top\right)\right]$$
$$\quad + \mathbb{E}_{Z_0,W}\left[b^2\,\text{tr}\left(XMM^\top X^\top W^\top WXMM^\top X^\top\right)\right]$$
$$=d\,\mathbb{E}_{Z_0}\left[a^2\,\text{tr}\left(XX^\top XX^\top\right)+2ab\,\text{tr}\left(XX^\top XMM^\top X^\top\right)+b^2\,\text{tr}\left(XMM^\top X^\top XMM^\top X^\top\right)\right]$$
$$=a^2 d^2 n(n+1+d)+2abd^2(n+1+d)T_1 + b^2 d^2(T_1^2 + (1+d)T_2).$$

Simultaneously, we can verify that $\mathbb{E}_{Z_0,W}\left[\text{tr}\left(W^\top W\right)\right]=d^2$ and

$$\mathbb{E}_{Z_0,W}\left[\text{tr}\left(W^\top \widetilde{W}\right)\right]=\mathbb{E}_{Z_0,W}\left[aW^\top WXX^\top + bW^\top WXMM^\top X^\top\right]=ad^2 n + bd^2 T_1.$$

Combining the results above, we aim to find the optimal $a, b, M$ that minimize

$$\frac{1}{d^2}\mathcal{L}(V,Q)=c_0 + c_1 T_1 + c_2 T_1^2 + c_3 T_2,$$

where

$$c_0 = a^2 n(n+1+d)+1+2an, \quad c_1 = 2ab(n+1+d)+2b,$$
$$c_2 = b^2, \quad c_3 = b^2(1+d).$$

Since $c_3 \geq 0$, to minimize $\mathcal{L}(V,Q)$ we need to minimize $T_2$. Given that $MM^\top$ is symmetric, we
denote its $n$ eigenvalues as $\lambda_i$, $i \in [1,n]$. Then by Cauchy–Schwarz inequality,

$$\text{tr}\left(MM^\top MM^\top\right)=\sum_{i=1}^{n}\lambda_i^2 \geq \frac{1}{n}\left(\sum_{i=1}^{n}\lambda_i\right)^2 = \frac{1}{n}\text{tr}^2(MM^\top).$$

Therefore, $\mathcal{L}(V,Q)$ is minimized only if the inequality above holds with equality, which implies
that $\lambda_i = \lambda_j$ for any $i \neq j$. This concludes the proof by showing that there exists $\lambda \in \mathbb{R}$ such that
$MM^\top = \lambda I_d$, and thus $DD^\top = \text{diag}(\lambda I_d, 0)$. $\qquad\square$

### D.6 PROOF OF PROPOSITION 5

*Proof.* We will continue from eqs. (38) and (39). After applying token-wise dropout, we have

$$\mathsf{TF}(Z_0; \{V_l, Q_l\}_{l=1}^2, X_1 \leftarrow \overline{X}_1) = \sum_{i=1}^n (ao_2^{3i-2} + bo_2^{3i})o_1^{3i-2}o_1^{3i}y_i x_i^\top o_1^{3n+1}o_2^{3n+3}x_{\text{test}}$$

$$+ c\sum_{j=1}^n \sum_{k=1}^n \left( \sum_{i=1}^{n+1} o_2^{3i-1}d_{j,i}d_{k,i} \right) o_1^{3j}o_1^{3k-2}y_j x_k^\top o_1^{3n+1}o_2^{3n+3}x_{\text{test}}, \tag{41}$$

$$\mathsf{TF}(Z_0; \{V_l, Q_l\}_{l=1}^2, X_1 \leftarrow \Delta X_1) = co_2^{3n+3}\sum_{j=1}^n \left( \sum_{i=1}^{n+1} d_{j,i}d_{n+1,i} \right) o_1^{3j}o_1^{3n+1}y_j x_{\text{test}}^\top x_{\text{test}},$$

where $a = b_2 c_2 a_1 d_y^y b_1 d_y^x (1 + a_1 d_x^x)$, $b = b_2 c_2 a_1 d_x^y (1 + b_1 d_y^x)a_1 d_x^x$ and $c = b_2 c_2 a_1 d_x^y b_1 t_y a_1 t_x$. One can verify that our previous analysis about $\mathsf{TF}(Z_0; \{V_l, Q_l\}_{l=1}^2, X_1 \leftarrow \Delta X_1)$ still holds and we thus have $d_{n+1,:} = 0$. We then define:

$$O_l^1 = \text{diag}(o_l^1, \cdots, o_l^{3n-2}) \in \mathbb{R}^{n\times n}, \quad O_l^2 = \text{diag}(o_l^3, \cdots, o_l^{3n}) \in \mathbb{R}^{n\times n}, \quad \text{for } l \in [2],$$

$$O_2^3 = \text{diag}(o_2^2, \cdots, o_2^{3n+2}) \in \mathbb{R}^{(n+1)\times(n+1)}.$$

By defining

$$\widetilde{W} = \sum_{i=1}^n (ao_2^{3i-2} + bo_2^{3i})o_1^{3i-2}o_1^{3i}y_i x_i^\top + c\sum_{j=1}^n \sum_{k=1}^n \left( \sum_{i=1}^{n+1} o_2^{3i-1}d_{j,i}d_{k,i} \right) o_1^{3j}o_1^{3k-2}y_j x_k^\top,$$

One can verify that

$$\widetilde{W} = A + B + C \triangleq aY O_1^2 O_2^1 O_1^1 X^\top + bY O_1^2 O_2^2 O_1^1 X^\top + cY O_1^2 M O_2^3 M^\top O_1^1 X^\top.$$

Then, we will compute the expectation of each term in the following decomposition:

$$\mathcal{L}(V, Q) = \mathbb{E}_{Z_0, W}\left[ \text{tr}\left( \widetilde{W}^\top \widetilde{W} \right) + 2\,\text{tr}\left( W^\top \widetilde{W} \right) + \text{tr}\left( W^\top W \right) \right],$$

Specifically, let $T_1 = \text{tr}\left( MM^\top \right)$, $T_2 = \text{tr}\left( MM^\top MM^\top \right)$, $T_3 = \|M\|_4^4$, $T_4 = \sum_{i=1}^n \|M_{i,:}\|_2^4$, $T_5 = \sum_{j=1}^{n+1} \|M_{:,j}\|_2^4$, we then have

$$\mathbb{E}[\text{tr}\left( A^\top A \right)] = a^2 d^2(np^3 + n(n-1)p^6 + (1+d)np^3),$$

$$\mathbb{E}[\text{tr}\left( B^\top B \right)] = b^2 d^2(np^3 + n(n-1)p^6 + (1+d)np^3),$$

$$\mathbb{E}[\text{tr}\left( C^\top C \right)] = c^2 d^2(p^6 T_1^2 + (1+d)(p^4 - p^6)T_4 + (1+d)(p^5 - p^6)T_5$$
$$+ (1+d)(p^3 - p^4 - p^5 + p^6)T_3 + (p^3 - p^4)T_4 + p^4 T_2 + dp^6 T_2),$$

$$\mathbb{E}[\text{tr}\left( A^\top B \right)] = abd^2(np^4 + n(n-1)p^6 + (1+d)np^4),$$

$$\mathbb{E}[\text{tr}\left( A^\top C \right)] = acd^2((p^4 + (n-1)p^6)T_1 + (1+d)p^4 T_1),$$

$$\mathbb{E}[\text{tr}\left( B^\top C \right)] = bcd^2((p^4 + (n-1)p^6)T_1 + (1+d)p^4 T_1),$$

$$\mathbb{E}[\text{tr}\left( W^\top A \right)] = ad^2 np^3, \quad \mathbb{E}[\text{tr}\left( W^\top B \right)] = bd^2 np^3, \quad \mathbb{E}[\text{tr}\left( W^\top C \right)] = cd^2 p^3 T_1.$$

Summarizing our analysis above, $\min_M \mathcal{L}(V, Q)$ is equivalent to:

$$\min_M \left\{ c_0 + c_1 T_1 + c_2 T_2 + c_3 T_3 + c_4 T_4 + c_5 T_5 + c_6 T_1^2 \right\},$$

where

$$c_0 = 1 + n(2 + d)p^3(a^2 + b^2) + 2np^3(a + b) + 2n(2 + d)p^4 ab + n(n-1)p^6(a + b)^2,$$

$$c_1 = 2(a + b)c(p^4 + (n-1)p^6 + (1+d)p^4) + 2cp^3,$$

$$c_2 = c^2(p^4 + dp^6),$$

$$c_3 = c^2(1+d)(p^3 - p^4 - p^5 + p^6),$$

$$c_4 = c^2((1+d)(p^4 - p^6) + (p^3 - p^4)),$$

$$c_5 = c^2(1+d)(p^5 - p^6),$$

$$c_6 = c^2 p^6.$$

It is easy to verify that $c_2, c_3, c_4, c_5, c_6 \geq 0$. □

## D.7 PROOF OF PROPOSITION 6

**Proposition 6** (Restate). *Let $d_p$ denote the number of non-EOS tokens. Given any $L$-layer, single-head, $d$-dimensional linear-attention transformer with EOS tokens:*

$$\mathsf{TF}\big(Z_0; \{V_l, Q_l, P_l\}_{l \in [L]}\big) = (Z_L)_{:,d_p+1}, \quad (Z_0)_{:,d_p+1} = 0,$$

*where*

$$Z_l \in \mathbb{R}^{d \times (d_p+1)}, \ V_l, Q_l \in \mathbb{R}^{d \times d}, \ P_l \in \mathbb{R}^{(d_p+1) \times (d_p+1)},$$

$$Z_l = Z_{l-1} + V_l Z_{l-1} M(Z_{l-1}^\top Q_l Z_{l-1}^\top + P_l), \quad M = \mathrm{diag}(I_{d_p}, 0).$$

*There exists an $L$-layer, two-head, $2d$-dimensional linear-attention transformer operating without EOS tokens:*

$$\mathsf{TF}\big(\overline{Z}_0; \{\overline{V}_l^h, \overline{Q}_l^h, \overline{P}_l^h\}_{l \in [L], h \in [2]}\big) = (\overline{Z}_L)_{d:2d, d_p},$$

*where*

$$\overline{Z}_l \in \mathbb{R}^{2d \times d_p}, \ \overline{V}_l^h, \overline{Q}_l^h \in \mathbb{R}^{2d \times 2d}, \ \overline{P}_l^h \in \mathbb{R}^{d_p \times d_p},$$

$$\overline{Z}_l = \overline{Z}_{l-1} + \sum_{h=1}^{2} \overline{V}_l^h \overline{Z}_{l-1}(\overline{Z}_{l-1}^\top \overline{Q}_l^h \overline{Z}_{l-1}^\top + \overline{P}_l^h).$$

*Such that for any $Z \in \mathbb{R}^{d \times d_p}$, by letting $Z_0 = [Z \quad 0]$ and $\overline{Z}_0 = \begin{bmatrix} Z \\ 0 \end{bmatrix}$, we have*

$$\mathsf{TF}\big(Z_0; \{V_l, Q_l, P_l\}_{l \in [L]}\big) = \mathsf{TF}\big(\overline{Z}_0; \{\overline{V}_l^h, \overline{Q}_l^h, \overline{P}_l^h\}_{l \in [L], h \in [2]}\big).$$

*Proof.* We construct $\overline{V}_l^h$, $\overline{Q}_l^h$, and $\overline{P}_l^h$ as follows:

$$\overline{V}_l^1 = \begin{bmatrix} V_l & 0 \\ 0 & 0 \end{bmatrix}, \quad \overline{Q}_l^1 = \begin{bmatrix} Q_l & 0 \\ 0 & 0 \end{bmatrix}, \quad \overline{P}_l^1 = (P_l)_{1:d_p, 1:d_p},$$

$$\overline{V}_l^2 = \begin{bmatrix} 0 & 0 \\ V_l & 0 \end{bmatrix}, \quad \overline{Q}_l^2 = \begin{bmatrix} 0 & Q_l \\ 0 & 0 \end{bmatrix}, \quad \overline{P}_l^2 = \begin{bmatrix} 0 & (P_l)_{:,d_p+1} \end{bmatrix}.$$

We will show that for any $l \in [L]$, it satisfies $\overline{Z}_l = \begin{bmatrix} (Z_l)_{:,(1:d_p-1)} & (Z_l)_{:,d_p} \\ 0 & (Z_l)_{:,d_p+1} \end{bmatrix}$. One can verify that it holds trivially for $l = 0$. Then, suppose it holds for some $l = k - 1$, we have

$$\overline{Z}_k = \overline{Z}_{k-1} + \overline{V}_k^1 \overline{Z}_{k-1}(\overline{Z}_{k-1}^\top \overline{Q}_k^1 \overline{Z}_{k-1}^\top + \overline{P}_k^1) + \overline{V}_k^2 \overline{Z}_{k-1}(\overline{Z}_{k-1}^\top \overline{Q}_k^2 \overline{Z}_{k-1}^\top + \overline{P}_k^2)$$

$$= \overline{Z}_{k-1} + \begin{bmatrix} V_k(Z_{k-1})_{:,1:d_p}\left((Z_{k-1})_{:,1:d_p}^\top Q_k(Z_{k-1})_{:,1:d_p} + (P_k)_{1:d_p,1:d_p}\right) \\ 0 \end{bmatrix}$$

$$+ \begin{bmatrix} 0 \\ V_k(Z_{k-1})_{:,1:d_p} \end{bmatrix} \left( \begin{bmatrix} 0 & (Z_{k-1})_{:,1:d_p}^\top Q_k(Z_{k-1})_{:,d_p+1} \end{bmatrix} + \begin{bmatrix} 0 & (P_k)_{:,d_p+1} \end{bmatrix} \right)$$

$$= \overline{Z}_{k-1} + \begin{bmatrix} V_k Z_{k-1} M\big(Z_{k-1}^\top Q_k(Z_{k-1})_{:,1:d_p} + (P_k)_{:,1:d_p}\big) \\ 0 \end{bmatrix}$$

$$+ \begin{bmatrix} 0 & 0 \\ 0 & V_k Z_{k-1} M\big(Z_{k-1}^\top Q_k(Z_{k-1})_{:,d_p+1} + (P_k)_{:,d_p+1}\big) \end{bmatrix}$$

$$= \begin{bmatrix} (Z_k)_{:,1:d_p} \\ 0 \end{bmatrix} + \begin{bmatrix} 0 & 0 \\ 0 & (Z_k)_{:,d_p+1} \end{bmatrix}.$$

The proof is complete. $\qquad\square$

