# OpenReview forum: "Understanding Task Vectors in In-Context Learning: Emergence, Functionality, and Limitations"
_ICLR.cc/2026/Conference — ICLR 2026 Poster_

### Official Review · Reviewer_4skn · 2025-10-28

**Soundness:** 2
**Presentation:** 3
**Contribution:** 2
**Rating:** 4
**Confidence:** 3

**Summary:**

This paper proposes a conjecture that task vectors (TVs) facilitate zero-shot inference by distilling from the original ICL demonstrations a single representative demonstration encoding. To support the conjecture, the paper uses linear-attention models to show that task vectors naturally emerge as a weighted summation of the in-context demonstrations. A key implication of this conjecture is that a single task vector, like a single demonstration, is limited to representing rank-one mappings. To test this limitation, the paper introduces a class of bijection tasks that require higher-rank mappings. Empirical results on practical LLMs validate that the standard TV method consistently fails on these tasks, while full ICL succeeds. Building on this finding, the authors propose an enhanced TaskV-M method that injects multiple task vectors, which successfully improves performance on these complex tasks.

**Strengths:**

- The organization of the paper is clear and easy to follow. Starting with a clear conjecture, the paper goes on to theoretically demonstrate the conjecture with a constructed simplified case.
- The use of a bijection task to identify a weakness of TV methods is sound and interesting.

**Weaknesses:**

- For table 1, while TV performs quite badly on bijection tasks, it seems that ICL is not much better. Given that ICL consistently outperforms TV on all tasks, what I can see most clearly is that the bijection task is much harder than other tasks, but not that "ICL preserves performance in many case".
- Some analysis seems to have been demonstrated by previous papers. For example, the findings of how information flows in ICL in Figure 3 (a,b) has been demonstrated in [1].
- I do not quite understand the implication of Figure 4c. It is difficult to see from Figure 4c that the performance of TV and 1-shot ICL are "parallel".



[1] Wang et al., Label Words are Anchors: An Information Flow Perspective for Understanding In-Context Learning, EMNLP 2023.

**Questions:**

- Theoretically, can the TaskV-M method achieve the same performance as ICL?
- For the triplet tasks, how are the triplet demonstrations delimited? Why does the attention focus on the yi tokens instead of the delimiters?
- I might be a bit confused, but why is that, for the bijection task, in table 1, ICL outperforms TV, but in table 2, ICL underperforms TV?

---

> ### Author Response · Authors · 2025-11-19
>
> We thank Reviewer 4skn for the thoughtful feedback and for recognizing the importance of our contribution to the understanding of ICL. Below, we address the reviewer’s comments and questions in detail:
>
> **Q1. ICL vs. TV on Bijection Tasks**
>
> We agree that bijection tasks are inherently more challenging, resulting in lower absolute accuracies for both ICL and TV. This is further influenced by the relatively weak Llama-7B model and the limited number of in-context examples ($n = 10$). In Tables 8–9 (the revised paper), we evaluate more advanced LLMs (Llama3, Qwen3) on the same suite of tasks with more demonstrations ($n = 20$). Under these conditions, the separation between ICL and TV is much clearer, with ICL accuracies exceeding 60\% in most cases. These results more faithfully reflect our claim that “ICL preserves performance in many cases.”
>
> **Q2. Relation to Prior Information-Flow Analyses**
>
> We agree that Wang's work indeed analyzes information flow via label words and shows that outputs attend to specific prompt tokens. We include Wang's work as one of the references. However, our saliency visualizations in Figure 3 are not intended as a standalone novelty, but rather as empirical validation that the inner mechanism predicted by our theory is present in real LLMs. Since our primary contribution lies in identifying and substantiating the mechanism underlying task vectors, this similarity does not diminish the novelty or significance of our findings.
>
> **Q3. Interpretation of Figure 4(c)**
>
> We agree that the performance curves of TV and 1-shot ICL do not perfectly align in the current plot, and the interpretation in the original text may have been overly precise. A more accurate characterization is that “their performances are highly related,” which aligns with our theoretical perspective that TV approximates 1-shot ICL. We have updated the wording in the revision accordingly.
>
> **Q4. Can TaskV-M Theoretically Match ICL**
>
> In the linear-attention setting, a single task vector induces a rank-one update, whereas TaskV-M with $k$ vectors can implement an update of rank at most $k$. In principle, if the underlying task mapping has rank $\le k$ and the task vectors are appropriately chosen, TaskV-M can match ICL performance in that regime.
>
> However, we do not claim that TaskV-M can generally reproduce arbitrary ICL behavior for complex, nonlinear tasks. Our experiments demonstrate only that increasing the number of task vectors systematically relaxes the rank-one bottleneck and brings performance closer to ICL on higher-rank tasks.
>
> **Q5. Triplet Demonstrations Delimitation**
>
> In our real-world LLM experiments, each triplet demonstration is formatted as ``$x_i \to y_i$’’, where the arrow token serves as the delimiter between input and output (see eq. (13) for the prompt structure). According to our information-flow analysis, the model first aggregates the complete information of each demonstration into a single token, which is then propagated to the final arrow token (as shown in Figure 5). Due to causal attention, $y_i$ is the only token among $(x_i, \to, y_i)$ that can access the full information within a demonstration, which naturally leads to saliency concentrating on $y_i$ rather than the delimiter.
>
> **Q6. Clarification on Table 1 and Table 2**
>
> The ICL results in Table 1 and the Baseline results in Table 2 are not directly comparable, as they rely on different prompt configurations:
>
> * Table 1: ICL uses many-shot prompts ($n = 10$), while task vectors are injected into zero-shot prompts. As expected, task vectors underperform standard ICL because they function as an acceleration mechanism rather than a full replacement.
>
> * Table 2: The Baseline uses few-shot prompts (0–4 demonstrations), matching the setting used for task vector injection. Under these constrained conditions, injecting task vectors improves ICL performance.
>
> We have added further clarifications about this in the revision.

---

### Official Review · Reviewer_t2pe · 2025-10-31

**Soundness:** 4
**Presentation:** 4
**Contribution:** 4
**Rating:** 6
**Confidence:** 4

**Summary:**

This paper looks into how task vectors emerge in ICL. The main idea is that a task vector, formed at a special prompt token, works like a distilled version of the whole demonstration set, basically acting as a single representative example. The authors support this through a theoretical analysis using linear-attention Transformers, showing that such task vectors can only express simple, rank-one mappings. They then back this up with experiments, including on large models like LLaMA-7B, which show that real models behave in similar ways. To deal with the limitation, they try injecting multiple task vectors into the prompt and find that this gives small but consistent gains. Overall, the paper offers a clear explanation for what task vectors are doing and what their limits are.

**Strengths:**

- The paper develops a clear and rigorous theoretical framework that explains the role of task vectors in few-shot learning.
- A major strength lies in the bijection experiments, which directly validate the predicted limitations of single task vectors and demonstrate the practical implications of the theory.
- The authors show that the same task vector behaviors emerge in large pretrained language models, reinforcing the relevance of the conjecture beyond simplified settings.
- The proposed multi-vector strategy builds naturally on the analysis and provides an interpretable method to mitigate expressiveness constraints.
- The clarity of writing, thoughtful experimental design, and theoretical depth make the paper both accessible and impactful within the ICL literature.

**Weaknesses:**

- The work builds on linear attention. I acknowledge that generalizing the proofs to standard Transformers is challenging, but the lack of a formal extension limits the scope.

- The multi vector strategy yields only moderate gains and likely needs further refinement for higher complexity tasks. Most of the tasks tested are short or simple, so it's still unclear how this approach would hold up on more complex reasoning problems.

- In Table 2 the task vector remains strong on bijection tasks, which softens the rank one limitation highlighted by Table 1. The paper should clearly explain why this happens and under which conditions the limitation manifests or is mitigated.

**Questions:**

- Please clarify why the task vector remains strong on bijection tasks in Table 2 despite the failure cases in Table 1. For example specify differences in prompt structure, presence of additional demonstrations, model nonlinearity, or training data effects that could compensate for rank limits. If this discrepancy is convincingly explained and bounded with clear conditions I am inclined to raise the overall score.

- Could the multi vector approach be improved by learning to select or weight vectors adaptively rather than uniform insertion?

- Do you expect the task vector mechanism and its limits to extend to more compositional or long range reasoning tasks, and what evidence would most directly test this?

---

> ### Author Response · Authors · 2025-11-19
>
> We thank Reviewer t2pe for the insightful comments and for acknowledging the significance of our work in advancing the understanding of transformer models. Below, we address the reviewer’s questions in detail:
>
> **Q1. Simplifications in Model Settings**
>
> We agree that our theoretical model differs from real-world LLMs. Our goal is to adopt a minimal, analytically tractable framework that isolates how ICL and task vectors emerge. Similar simplifications are standard in theoretical ICL studies (Von Oswald et al., 2023a; Ahn et al., 2023; Wu et al., 2024). Moreover, our empirical results on full, practical LLMs substantiate the key qualitative predictions of our theory, including the information flow structure and the rank-one limitation.
>
> **Q2. Performance of TaskV-M**
>
> We agree that the current multi-vector strategy yields moderate gains and that extending the evaluation to more complex, long-range reasoning tasks would be valuable. Our objective is to propose a simple, fully interpretable modification that follows directly from the theory: if a single task vector corresponds to a rank-one update, injecting multiple vectors should effectively increase the rank and thereby benefit harder tasks. This is precisely what we observe across the benchmark suite, with the clearest improvements on bijection tasks.
>
> We position our experiments as an initial step and emphasize that applying multi-vector methods to richer reasoning benchmarks and longer contexts represents an important direction for future work.
>
> **Q3. Table 1 and Table 2**
>
> We appreciate the reviewer for highlighting this potential source of confusion. The settings for Table 1 and Table 2 are largely the same, except for the number of demonstrations:
>
> * Table 1: Task vectors are injected into **zero-shot** prompts (TV).
> * Table 2: Task vectors are injected into **few-shot** prompts (TaskV).
>
> Given this difference, it is unsurprising that TaskV achieves higher performance on bijection tasks in Table 2, as the additional few-shot examples compensate for the rank limitation. In contrast, TaskV yields approximately 50\% accuracy on bijection tasks in the zero-shot setting (Table 2), which is consistent with random guessing. We have emphasized this clarification in the revision.
>
> **Q4. Adaptive Weighting and Selection**
>
> We agree that adaptively selecting or weighting task vectors is a promising extension. In this work, we intentionally adopt a simple, training-free scheme to maintain a clear theoretical connection and avoid confounding factors. Meanwhile, methods such as Adaptive Task Vectors (Kang et al., 2025) explore adaptive selection strategies and provide improvements that are orthogonal to our multi-vector approach. We view this as an important direction for future work.
>
> **Q5. More Complex Real-World Tasks**
>
> We agree that extending the analysis to more complex reasoning benchmarks is an exciting direction. Prior works like Adaptive Task Vectors (Kang et al., 2025) demonstrate that the task vector mechanism can indeed be effective on challenging reasoning tasks. However, our current setup focuses on tasks where the task identity must be inferred solely from in-context examples, without explicit instructions. Standard reasoning benchmarks include rich natural-language descriptions, which alter the underlying mechanism (i.e., the task can often be inferred directly from instructions rather than examples). Understanding how rank limitations manifest in such instruction-rich settings is an important and nontrivial extension, which we plan to explore in future work.

---

### Official Review · Reviewer_iUth · 2025-10-31

**Soundness:** 3
**Presentation:** 3
**Contribution:** 3
**Rating:** 6
**Confidence:** 3

**Summary:**

This paper investigates task vectors in in-context learning (ICL), proposing that they act as distilled representative demonstrations. Through analysis of linear-attention models and saliency maps in large language models, the authors show that task vectors emerge from triplet-formatted prompts and encode task information via weighted combinations of examples. They identify a key limitation: task vectors are restricted to rank-one mappings, limiting performance on complex tasks. A multi-vector extension is proposed to address this, showing consistent improvements. The work provides useful insights into the structure and limitations of task vectors.

**Strengths:**

1. The paper presents a well-structured theoretical investigation into the emergence and functionality of task vectors in in-context learning (ICL), centered around the representative demonstration hypothesis and its implications under linear attention assumptions.

2. The qualitative analyses, including weight matrix visualizations and saliency maps, are clearly aligned with the theoretical framework and help to support the proposed mechanisms intuitively.

3. The paper evaluates its hypothesis across a diverse set of synthetic and linguistic tasks (e.g., linear regression, bijection), and the experimental outcomes reflect the predicted rank-one limitations, particularly in settings requiring high-rank mappings.

**Weaknesses:**

1. The theoretical analysis is limited to simplified linear attention models and does not incorporate components such as softmax attention, causal masking, or nonlinearity, which are central to modern transformer architectures.

2. While the appendix refers to prior approaches that leverage task or in-context vectors to improve performance, the empirical evaluation is limited to standard Few-shot ICL, TV, and the authors’ multi-vector variant. Including direct comparisons to these prior methods would help clarify the novelty and effectiveness of the proposed approach.

**Questions:**

1. The paper evaluates models such as GPT-J, Pythia, and Llama-2, but does not report results on more recent architectures like Qwen or the Llama-3 series. Do the rank-one limitations identified in this paper also persist in newer models such as Qwen or Llama-3 family?

2. The empirical evaluation focuses on synthetic and controlled linguistic tasks, leaving it unclear whether the observed rank limitations extend to more complex, real-world applications such as Question-answering or reasoning tasks. Have the authors explored such settings in their analysis?

---

> ### Author Response · Authors · 2025-11-19
>
> We thank Reviewer iUth for the constructive comments and for acknowledging the significance of our theoretical analysis and its insights into understanding task vectors. Below, we address each of the reviewer’s concerns:
>
> **Q1. Simplifications in Model Settings**
>
> We agree that our theoretical model differs from real-world LLMs. However, our goal is to adopt a minimal, analytically tractable framework that isolates the emergence of ICL and task vectors. Similar simplifications are also standard in theoretical ICL studies (Von Oswald et al., 2023a; Ahn et al., 2023; Wu et al., 2024). Moreover, our empirical results on full, practical LLMs support the key qualitative predictions of our theory, including the information flow structure and the rank-one limitation.
>
> **Q2. Comparison to Prior Approaches**
>
> We thank the reviewer for the helpful suggestion. We would like to clarify that the objective of TaskV-M is not to surpass state-of-the-art ICL techniques, but rather to demonstrate the potential of multi-vector injection. The prior approaches discussed in the appendix generally build upon or extend the single-vector injection paradigm and are orthogonal to our multi-vector framework. We chose the standard task vector method (TaskV) for its simplicity and efficiency, as it requires only basic forward passes. In contrast, other approaches introduce additional complexity that would obscure the clean comparison we aim to provide:
>
> * Function vectors (Todd et al., 2024) require computing significance metrics for all attention heads;
> * In-context vectors (Liu et al., 2024) rely on contrastive datasets to extract behavioral directions;
> * State vectors (Li et al., 2024) require external optimization.
>
> Given the limited duration of the discussion phase, we believe that incorporating these methods is better suited for future extensions of our work.
>
> **Q3. Generalizing to More Advanced LLMs**
>
> We appreciate this point and have conducted additional experiments to assess the robustness of our conclusions. As reported in Tables 8–9 in the revised paper, we evaluate more advanced LLMs (Llama3, Qwen3) on the same suite of tasks and observe the same qualitative outcome: **task vectors consistently exhibit the rank-one limitation and fail on bijection tasks.** This consistency across architectures indicates that the failure reflects a fundamental expressiveness constraint rather than model-specific artifacts.
>
> **Q4. More Complex Real-World Tasks**
>
> We agree that extending the analysis to more complex QA or reasoning benchmarks is an interesting direction. Our current setup focuses on tasks for which the task identity must be inferred solely from the in-context examples, without explicit task descriptions in the prompt. In contrast, standard reasoning and QA benchmarks contain rich natural-language instructions, which change the underlying mechanism (i.e., the task can often be inferred directly from instructions rather than examples). Understanding how rank limitations and multi-vector injection behave in such instruction-rich settings is an important and non-trivial extension; we plan to leave this to future work.

---

### Official Review · Reviewer_8Lar · 2025-11-03

**Soundness:** 3
**Presentation:** 3
**Contribution:** 3
**Rating:** 6
**Confidence:** 3

**Summary:**

The paper discussed three formats of in-context leanring: Single, Pairwise, and Triplet.
For Pairwise, the paper provides Theorem 1 to theoretically show that an L-layer linear transformer allocates one layer for embedding concatenation and utilizes the remaining L−1 layers to perform gradient descent.
For Triplet, the paper provides Theorem 2 to theoretically show that task vector naturally emerge from pretraining on triplet format.
These analyses are coupled with experiments in Sec 6.
The paper further show the task vector fails to predict on bijection tasks.
Sec 6 further shows multi-vector injection works better than vanilla

**Strengths:**

(1) The paper considers three ICL formats Single, Pairwise, and Triplet, making the view broad.

(2) The paper provides Theorems 1 and 2 for Pairwise and Triplet and gives clear explanations/intuitions.

(3) The paper identifies a task bijection that the one-shot approach TV cannot solve.

(4) The paper shows that multi-vector injection works better than the original task vector.

**Weaknesses:**

(1) Different from real-world LLM, the considered self-attention is without normalization, with position embedding as a one-hot vector.

(2) For Table 1, the conclusion is not well-supported. Will more advanced LLM lead to different conclusion?

**Questions:**

(1) I wonder whether L134, equation (2), should have "-" instead of "+" for loss.

(2) I did not understand L290-292. I believe the paper is discussing causal attention. Why is it said "as bi-directional attention allows"?

---

> ### Author Response · Authors · 2025-11-19
>
> We thank Reviewer 8Lar for the constructive feedback and for recognizing the breadth and depth of our contributions. Below, we address each of the reviewer’s questions in detail:
>
> **Q1. Simplifications in Linear Attention**
>
> We agree that our theoretical model differs from real-world LLMs. However, our goal is to adopt a minimal, analytically tractable framework that isolates the emergence of ICL and task vectors. Similar simplifications are also standard in theoretical ICL studies (Von Oswald et al., 2023a; Ahn et al., 2023; Wu et al., 2024). Moreover, our empirical results on full, practical LLMs support the key qualitative predictions of our theory, including the information flow structure and the rank-one limitation.
>
> **Q2. Generalizing to More Advanced LLMs**
>
> We appreciate this point and have conducted additional experiments to assess the robustness of our conclusions. As shown in Tables 8–9 in the revised paper, we evaluate more advanced LLMs (Llama3, Qwen3) on the identical suite of tasks and observe the same qualitative outcome: **task vectors consistently exhibit the rank-one limitation and fail on bijection tasks.** This consistency across architectures indicates that the failure arises from a fundamental expressiveness constraint rather than model-specific behavior.
>
> **Q3. $+$/$-$ Sign in Equation (2)**
>
> We clarify that, in our setting, flipping the sign in the loss is equivalent to the reparameterization $W \mapsto -W$ and is therefore without loss of generality. Since our theoretical results depend solely on the structure of the gradient updates, both choices yield identical gradient-descent dynamics up to this trivial reparameterization. We adopt the $+$ sign to simplify the theoretical analysis.
>
> **Q4. Causal vs. Bi-Directional Attention**
>
> We clarify that although our empirical LLM experiments employ causal attention, our theoretical analysis uses bi-directional linear attention for simplicity and more transparent proofs. The sentence at L290–292 refers specifically to the theoretical bi-directional setting. We further direct the reviewer to Section 5, where we discuss the implications of causal attention on our theoretical results.

---

### Author Response · Authors · 2025-11-30
**General Response**

* We thank all reviewers and the area chairs for their careful evaluations and constructive feedback. The reviews recognize the scope of the paper, the clarity of the theoretical results, the identification of the rank-one limitation, and the empirical benefits of multi-vector injection.

* **Realism of the theoretical setting.** Several comments noted that our linear transformer analysis uses simplifying assumptions such as the absence of normalization and one-hot positional embeddings. In the rebuttal and revision, we explicitly position our model as a minimal, analytically tractable framework in line with multiple existing theoretical ICL works, and we emphasize that our empirical studies on practical LLMs qualitatively validate the key predictions of this theory.

* **Generalization to more advanced LLMs.** In response to questions about whether stronger LLMs might behave differently, we added new experiments on modern models (Llama 3 and Qwen 3) on the same task suite. These additional results show the same qualitative pattern. Task vectors systematically exhibit a rank-one limitation and fail on bijection tasks, indicating that the observed phenomenon is fundamental rather than specific to a single model family.

* **Clarification and future works** We revised the manuscript for improved clarity and organization, especially around the different ICL settings used in Tables 1 and 2. We expanded the future work section to incorporate orthogonal improvements for task vectors and more complex reasoning tasks, to make the scope and position of our results more explicit.

* Due to the special situation this year, we respectfully remind the area chairs that **reviewer t2pe explicitly states to raise the score** if the confusion about the ICL settings in Tables 1 and 2 is satisfactorily addressed. Our clarifications directly target this concern, and we believe our response has clearly explained the discrepancy between the two.

---

### Meta-Review · Area_Chair_SeqN · 2026-01-06

**Summary:**

This paper presents a careful theoretical and empirical investigation into the emergence, functionality, and limitations of task vectors in in-context learning. By analyzing linear-attention transformers, the authors provide a clear and principled explanation of how task vectors arise, how they function as representative demonstrations, and why they exhibit fundamental limitations in expressiveness. The paper further validates these insights through empirical analyses on practical LLMs and proposes a natural extension via multi-vector injection. Overall, this is a good and well-executed paper. While several reviewers raised concerns about the focus on simplified linear transformer settings, the rebuttal clearly positions this choice as an appropriate abstraction for gaining theoretical insight. This modeling limitation does not detract from the contribution and does not affect the acceptability of the paper.

**Reviewer Concerns:**

The main concerns raised by reviewers centered on the simplified theoretical setting (linear-attention models), questions about generalization to more advanced and nonlinear LLM architectures, and requests for clarification of scope and implications. These concerns were well addressed in the rebuttal. The authors clarified that the linear setting is intentionally chosen to enable precise theoretical analysis, and they provided convincing empirical evidence and discussion connecting the theory to observed behaviors in real-world LLMs. Additional experiments and explanations sufficiently resolved requests for clarification and strengthened the overall narrative.

**Reviewer Scores:**

Reviewer t2pe’s questions regarding the scope and implications of the theoretical analysis were adequately resolved in the rebuttal, making an increase in score likely. Similarly, it would not be surprising if Reviewer 4skn raised their score from 4, as the newly provided experimental results and clarifications convincingly address their primary concerns. Overall, the discussion supports maintaining or increasing the existing reviewer scores.

---

### Decision · Program_Chairs · 2026-01-26

Accept (Poster)